# RESPIN-S1.0: A read speech corpus of 10000+ hours in dialects of nine Indian Languages

**Saurabh Kumar**[1], **Abhayjeet Singh**[1], **Deekshitha G**[1], **Amartyaveer**[1], **Jesuraj Bandekar**[1], **Savitha Murthy**[1], **Sumit Sharma**[1], **Sandhya Badiger**[1], **Sathvik Udupa**[1], **Amala Nagireddi**[1], **Srinivasa Raghavan K M**[2], **Rohan Saxena**[2], **Jai Nanavati**[2], **Raoul Nanavati**[2], **Janani Sridharan**[2], **Arjun Mehta**[2], **Ashish Khuraishi K S**[2], **Sai Praneeth Reddy Mora**[2], **Prashanthi Venkataramakrishnan**[2], **Gauri Date**[2], **Karthika P**[2], **Prasanta Kumar Ghosh**[1*]

[1]Department of Electrical Engineering, Indian Institute of Science, Bangalore, India
[2]Navana Tech, Mumbai, India
spirelab.ee@iisc.ac.in, prasantg@iisc.ac.in

## Abstract

We introduce **RESPIN-S1.0**, the largest publicly available dialect-rich read-speech corpus for Indian languages, comprising more than 10,000 hours of validated audio across nine major languages: Bengali, Bhojpuri, Chhattisgarhi, Hindi, Kannada, Magahi, Maithili, Marathi, and Telugu. Indian languages exhibit high dialectal variation and are spoken by populations that remain digitally underserved. Existing speech corpora typically represent only standard dialects and lack domain and linguistic diversity. RESPIN-S1.0 addresses this limitation by collecting speech across more than 38 dialects and two high-impact domains: agriculture and finance. Text data were composed by native dialect speakers and validated through a pipeline combining automated and manual checks. Over 200,000 unique sentences were recorded through a crowdsourced mobile platform and categorised into clean, semi-noisy, and noisy subsets based on transcription quality, with the clean portion alone exceeding 10,000 hours. Along with audio and transcriptions, RESPIN provides dialect-aware phonetic lexicons, speaker metadata, and reproducible train, development, and test splits. To benchmark performance, we evaluate multiple ASR models, including TDNN-HMM, E-Branchformer, Whisper, and wav2vec2-based self-supervised models, and find that fine-tuning on RESPIN significantly improves recognition accuracy over pretrained baselines. A subset of RESPIN-S1.0 has already supported community challenges such as the SLT Code Hackathon 2022 and MADASR@ASRU 2023 and 2025, releasing more than 1,200 hours publicly. This resource supports research in dialectal ASR, language identification, and related speech technologies, establishing a comprehensive benchmark for inclusive, dialect-rich ASR in multilingual low-resource settings.

**Dataset:** https://spiredatasets.ee.iisc.ac.in/respincorpus
**Code:** https://github.com/labspire/respin_baselines.git

---

*Corresponding author: prasantg@iisc.ac.in

39th Conference on Neural Information Processing Systems (NeurIPS 2025) Track on Datasets and Benchmarks.

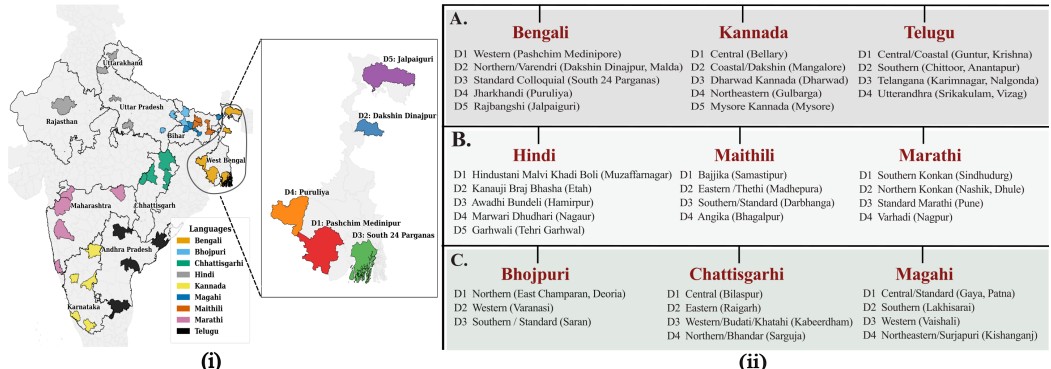

Figure 1: (i) District-level distribution of the nine RESPIN languages across India, based on the 2011 Census (illustrative, not to scale). Each language is shown in a distinct color, with an inset showing dialect-wise representation for Bengali. (ii) Language classification: A refers to scheduled, non-Devanagari; B to scheduled, Devanagari; and C to non-scheduled, Devanagari languages.

# 1   Introduction

India's linguistic landscape, comprising 22 scheduled languages and hundreds of dialects [2], demands inclusive speech technologies. However, the lack of curated and dialect-rich audio–text datasets [1, 2] has limited progress. Although 64% of India's population resides in rural areas and 57.8% belong to agricultural households [3], most ASR research has focused on English or standard language forms [4]. Existing corpora usually represent only standard dialects [5], resulting in degraded performance on regionally diverse speech.

To bridge this gap, **RESPIN-S1.0** presents a large-scale, multi-dialectal, multi-domain read-speech corpus covering nine Indian languages: Bengali (bn), Bhojpuri (bh), Chhattisgarhi (ch), Hindi (hi), Kannada (kn), Magahi (mg), Maithili (mt), Marathi (mr), and Telugu (te). These were selected based on speaker population, socio-economic diversity, and resource availability. Figure 1 shows the district-level language distribution and dialect breakdown. RESPIN is the first public corpus to provide large-scale dialectal data for Bhojpuri, Chhattisgarhi, and Magahi. The pipeline, from text composition to audio validation, was implemented at the dialect level to preserve linguistic integrity and includes manually verified phonetic lexicons following Indian Language Speech Label (ILSL) guidelines [6] and speaker metadata such as pincode, gender, and age group.

To promote reproducibility, RESPIN provides train, development, and test splits, dialect-level metadata, and ASR benchmarks using TDNN-HMM [7], E-Branchformer [8], Whisper [9], and wav2vec2-based SSL models such as IndicWav2Vec2 [10] and SPRING-Data2Vec.[3] Fine-tuning on RESPIN consistently improves ASR performance over models trained on external corpora. The dataset has already supported multilingual ASR challenges including SLT Code Hackathon 2022 [4] and MADASR (ASRU 2023, ASRU 2025) [5], with over 1200 hours released to the community. By capturing India's dialectal diversity, RESPIN advances inclusive voice technologies for underserved linguistic communities across India.

The RESPIN project was supported by the Gates Foundation to promote speech technologies for low-resourced Indian languages and marginalized communities. All data collection followed institutional ethics approval at Indian Institute of Science (IISc) Bangalore, with informed consent, privacy safeguards, and fair participant compensation. Additional details on demographics, compensation, and consent are provided in Appendix K.

---

[2]https://web.archive.org/web/20240914124112/https://censusindia.gov.in/nada/index.php/catalog/42561/download/46187/Language_Atlas_2011.pdf

[3]https://asr.iitm.ac.in/models

[4]https://sites.google.com/view/slt-team

[5]https://sites.google.com/view/respinasrchallenge2025/home

## 2  Background

Dialectal datasets are essential for developing accurate and inclusive speech technologies. They reflect real-world language use, improve recognition across regions, ensure accessibility for marginalized communities, support public service applications, and help preserve linguistic diversity. Continued investment in open, community-driven, and dialect-rich data collection is crucial for building equitable and effective speech systems. As emphasized in [11], treating dialects respectfully is fundamental to supporting underserved populations.

Globally, several initiatives have aimed to capture dialectal variation for inclusive speech technology. In the Arabic-speaking world, CASABLANCA [12] and MADAR [13] provide large-scale, multi-dialect corpora representing regional Arabic. In China, projects such as AISHELL [14] and THCHS-30 [15] focus primarily on Mandarin but lay important groundwork for dialectal research. In Africa, Masakhane ASR [16] and CMU Wilderness [17] expand coverage for underrepresented African languages and dialects through open, community-based initiatives. Collectively, these efforts represent major progress toward bridging the global gap in speech technology resources.

Table 1: Existing Indic Datasets

| Dataset | Languages | Domains | Districts | Hours | Speakers | Sentences | Source |
|---|---|---|---|---|---|---|---|
| INDICVOICES [18] | 13 | 52 | 145 | 7348 | 16237 | 11,00,000+ | Wikipedia, Composed, Spontaneous |
| INDICVOICES-R [19] | 22 | multi | multi | 1704 | 10496 | NA | NA |
| Kathbath [20] | 12 | multi | 203 | 1684 | 1217 | 12,00,000+ | IndicCorp (Web data) |
| Shrutilipi [21] | 12 | multi | NA | 6457 | NA | 33,00,000 | All India Radio |
| NPTEL [22] | 8 | 1 | NA | 857 | NA | NA | Lectures |
| Svarah [23] | 1 | 9 | 65 | 9.6 | 117 | NA | Wikipedia, Prompts, Spontaneous |
| SPRING-INX [24] | 10 | multi | 40+ | 2000 | 7609 | NA | NA |
| SPIRE-SIES [25] | 1 | NA | NA | 193 | 1607 | NA | NA |
| FLEURS [26] | 13 | NA | NA | 156 | 39 | NA | Wikipedia |
| Gram Vaani [27] | 1 | multi | 25 | 1108 | NA | NA | Spontaneous Speech |
| IISc-MILE [28] | 2 | NA | NA | 497 | 1446 | NA | NA |
| MUCS [28] | 3 | 4 | 4 (for Odia) | NA | 310 | 9080 | NA |
| Vāksãñcayah [29] | 1 | 8 | NA | 78 | 27 | 46,000 | Online stories |
| E&NE languages [30] | 4 | NA | multi | 19.75 | NA | NA | NA |
| NISP [31] | 6 | NA | NA | 56.86 | 345 | NA | news, TIMIT |
| CommonVoice [32] | 8 | 4 | NA | 373 | NA | NA | Wikipedia, Composed |
| CMS [33] | 6 | NA | NA | 35 | 243 | NA | Composed |
| IITB-MSC [34] | 1 | 1 | 1 | 109 | 36 | 3000 | Textbooks |
| IndicSpeech [35] | 3 | NA | NA | 24 | 3 | 42,046 | Online news |
| MSR Challenge [36] | 3 | NA | NA | 150 | 1286 | 1,02,397 | NA |
| Google TTS [37] | 1 | NA | NA | 3 | 6 | NA | NA |
| IIITH-ILSC [38] | 23 | NA | NA | 103.5 | 1150 | NA | NA |
| IndicTTS [39] | 13 | 4+ | NA | 389.6 | 26 | NA | Literature, newspapers |
| IIITH-ISD [40] | 7 | NA | NA | 11 | 35 | 1000 | Wikipedia |
| **RESPIN-S1.0** | **9** | **2** | **38+** | **10,416.58** | **18,000+** | **2,09,822** | **Composed** |

*NA* = Information Not Available

Table 1 compares major open-source Indic speech corpora across languages, domains, districts, duration, speaker count, and data sources. While many datasets cover multiple languages and include large audio volumes, most lack dialectal diversity and regional representation. They rely primarily on web content such as Wikipedia, books, or news articles, leading to limited relevance to everyday speech. RESPIN-S1.0 addresses these gaps by focusing on agriculture and banking, two domains central to India's rural and low-literacy communities, and by manually composing 2,09,822 sentences that capture regionally grounded, colloquial usage.

RESPIN-S1.0 introduces several key contributions that distinguish it from existing corpora:

**New language and dialect coverage** Unlike large-scale datasets such as IN-DICVOICES [18] and INDICVOICES-R [19], which emphasize scheduled languages, RESPIN is the first publicly available corpus offering validated data for low-resource, non-scheduled languages including Bhojpuri, Chhattisgarhi, and Magahi. These are often grouped under Hindi but possess distinct linguistic characteristics. With over 10,000 hours of validated audio from more than 18,000 speakers across 38 dialect-rich districts, RESPIN is the most comprehensive dialect-aware resource for Indian languages.

**Domain-specific composition** In contrast to datasets derived from generic sources, RESPIN's text corpus was authored by native dialect speakers for agriculture and finance. This design ensures vocabulary and sentence structures mirror natural communication within these key domains. The dataset enables voice-based digital services in native dialects, improving accessibility and fostering user trust.

**Dialectal integrity across the pipeline** Each stage of corpus creation—from text design to recording and validation—was implemented at the dialect level to preserve authenticity and ensure linguistic consistency across all 38 dialects.

Together, these design choices make RESPIN-S1.0 a distinctive and valuable resource for advancing inclusive, dialect-aware speech technologies in India's linguistically diverse setting.

# 3 Data Collection and Validation Pipeline

RESPIN is the first large-scale Indian speech corpus designed to preserve dialectal integrity throughout the data creation process. As shown in Figure 2, the pipeline includes language and dialect selection, manual text composition, multi-stage validation, and speaker-level audio collection. Unlike corpora built from scraped or generic online content, RESPIN focuses on agriculture and finance, with all text and audio created and validated at the dialect level. Con-

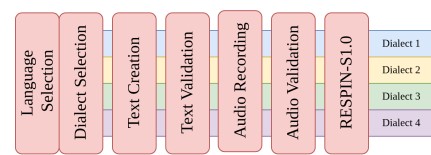

Figure 2: Data creation pipeline maintaining dialectal integrity.

tributors received guidance and updates throughout data collection, remaining well-informed and actively engaged with the project's goals and progress.

Participants were authenticated through a secure WhatsApp-based workflow, completed a click-wrap consent form in the Bolo App, and were briefed on task goals and compensation in their native languages (see Appendix K for onboarding and consent details).

## 3.1 Language and Dialect Selection

According to the Census of India (2011), 50.58M, 16.25M, 12.71M, and 13.58M people speak Bhojpuri, Chhattisgarhi, Magahi, and Maithili, respectively. While Magahi is often misclassified as a dialect of Hindi, it represents a distinct branch of the Indo-Aryan subfamily. To support such large speaker populations, it is essential to develop robust speech resources with rich vocabularies and diverse sentence corpora. RESPIN aims to build an ecosystem of speech recognition resources that empower India's working-class population. Between 2022 and 2023, 45.76% of India's workforce was engaged in agriculture and allied sectors, while finance and banking continue to play a central role in daily communication and access to services. By focusing on these two domains, RESPIN seeks to bridge the gap between under-resourced language communities and accessible, voice-driven technologies.

To support domain-specific sentence creation, a comprehensive list of topics was curated across agriculture and finance to guide sentence composition. These topics were compiled from diverse sources including magazines, websites, academic portals, and Wikipedia's outline articles. Wikipedia's topic trees and linked articles were particularly useful for hierarchical organization. The final list contains around 1500 topics, each associated with relevant reference links. Starting from broad categories such as crop cultivation or digital banking, the list narrows to specific subtopics including sugarcane harvesting techniques, UPI PIN setup, and transaction history checks in mobile apps. This curated topic bank ensured comprehensive and contextually relevant coverage of the target domains.

## 3.2 Text Data Acquisition and Validation

The creation of a dialect-level text corpus formed the foundation of RESPIN. Figure 3 outlines the overall workflow. The process began with onboarding and training dialect experts who curated text with high dialectal specificity, ensuring the inclusion of regional nuances and natural variation. As described earlier, the corpus was designed to capture sentences from agriculture and finance domains, making RESPIN uniquely domain-specific. Native speakers were hired through a multi-stage selection process to compose these sentences. The raw text then passed through a validation pipeline that combined automatic and manual checks to ensure linguistic quality and compliance. Only validated sentences were used for subsequent audio collection.

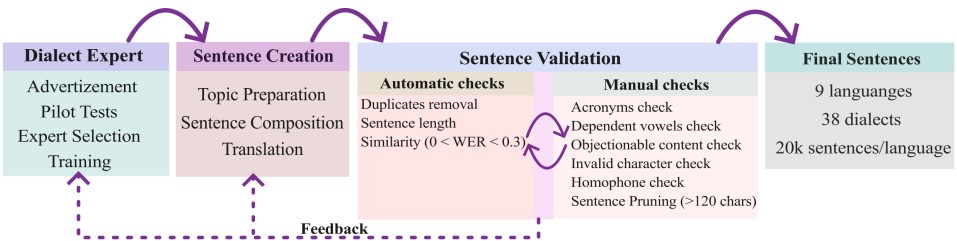

Figure 3: Flowchart showing the RESPIN text data preparation pipeline.

### 3.2.1 Sentence Creation

Large volumes of digital text exist in standard language forms but often lack colloquial and dialectal variation. To address this, RESPIN prioritized sourcing sentences directly from native speakers across districts, ensuring that the text reflects authentic regional expressions and natural communication patterns. Composers were tasked with crafting conversational, domain-specific sentences aligned with assigned topics in agriculture and finance. This enriched linguistic diversity but also introduced challenges, as dialectal variation can change even within small geographic regions. Recognizing this fluidity, RESPIN adopted an inclusive strategy that embraced intra-dialectal variation, yielding a rich and representative dataset.

Sentence composition followed strict guidelines to ensure consistency and usability: limiting sentence length, avoiding sentence-initial pronouns, excluding non-language numerals, restricting punctuation to period, comma, and question mark, avoiding controversial content, adhering to topic relevance, and maintaining consistent acronym formatting. Manual composition, though resource-intensive, produced the highest quality data. Translation from composed sentences was used selectively to address dialectal gaps. The proportion of translated sentences in Bhojpuri, Chhattisgarhi, Hindi, Kannada, Magahi, Maithili, Marathi, and Telugu was 6.65%, 100%, 9.8%, 0.1%, 0.4%, 16.5%, 5.1%, and 5.2%, respectively. Bengali sentences were composed entirely from scratch.

### 3.2.2 Sentence Validation

The composed text corpus underwent a multi-stage validation pipeline involving both automated (AC) and manual checks (MC) by trained validators. As multiple contributors participated in composition, inconsistencies and minor errors were expected. Since each sentence serves as a prompt for crowd-sourced recording, validation ensures that every utterance is accurate, unambiguous, and compatible with the mobile interface. The pipeline structure was consistent across languages with minor adaptations for linguistic differences. Key checks included (1) duplicate removal (AC), (2) invalid character correction (MC), (3) sentence length pruning (MC), (4) acronym standardization (MC), (5) matra correction (MC), (6) word-level edits (MC), (7) similar sentence filtering (MC), (8) homophone disambiguation (MC), and (9) additional language-specific checks (see Appendix B.2.3). Approximately 3.6% of the raw corpus was discarded due to unfixable errors or dialect mismatch. The process followed a version-controlled workflow, where each stage generated a new corpus version for auditing and rollback.

### 3.3 Audio Data Acquisition and Validation

Following text validation, audio data collection was conducted through a dedicated mobile application. Native speakers were prompted to read validated sentences aloud and record them in quiet environments. Each speaker was assigned a maximum of 577 sentences, though some recorded additional sentences to meet dialect-specific targets when others dropped out. To capture intra-dialectal acoustic variation, each sentence was recorded by multiple speakers, typically between 30 and 150. This many-to-one mapping enabled the dataset to represent a range of pronunciation styles, prosodic patterns, and speaking rates within each dialect.

### 3.3.1 Audio Validation Pipeline

The recorded audio underwent a structured validation process combining manual and semi-automated checks. Initially, about 5% of utterances in each dialect were manually audited to verify audio-text alignment. Based on these results, the entire dataset was categorized into three quality slabs: *Clean*, *Semi-noisy*, and *Noisy*, using a semi-automated scoring approach.

The slab categorization reflects the proportion of perfectly matched audio-text pairs. The clean slab contains the highest share of exact alignments, while the noisy slab includes those with the lowest. This design allows downstream ASR tasks to select subsets based on quality and robustness requirements. Complete definitions of slabs and associated thresholds are provided in Appendix E.

This validation framework ensures that the RESPIN audio corpus is high-quality, dialect-specific, and suitable for benchmarking robust ASR systems under realistic multilingual and multi-dialect conditions.

## 4 RESPIN-S1.0 Corpus

### 4.1 Text Data Analysis

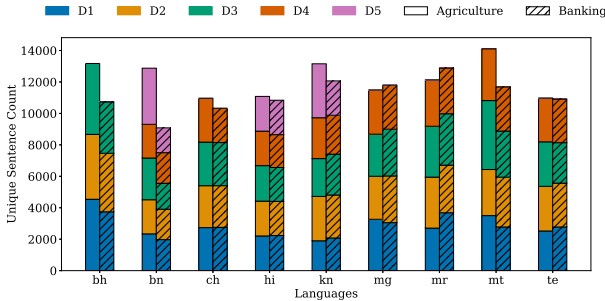

Figure 4: Unique sentence count per dialect, domain and language.

Table 2: Lexicon statistics across languages.

| LID | #chars | #phones | #words |
|-----|--------|---------|--------|
| bn | 64 | 50 | 18571 |
| bh | 71 | 54 | 14105 |
| ch | 68 | 50 | 13230 |
| hi | 72 | 55 | 16571 |
| kn | 66 | 50 | 50822 |
| mg | 72 | 54 | 21711 |
| mt | 72 | 55 | 19336 |
| mr | 68 | 51 | 35709 |
| te | 63 | 48 | 39235 |

Figure 4 presents the distribution of unique sentence counts across dialects, domains (Agriculture and Banking), and languages. Each language includes over 20,000 curated sentences covering 3–5 dialects with representation in both domains. Although perfect balance is constrained by the availability of dialect experts and regional factors, the dataset maintains approximate uniformity across dialect–domain pairs. Slight deviations, such as higher contributions from dialect D5 in `kn` and D3 in `mt`, reflect stronger regional participation or easier contributor access.

Table 2 shows lexicon statistics including unique characters, phonemes, and words per language. Lexicons were generated from the full sentence set (see Appendix C for details). Kannada (`kn`) and Telugu (`te`) exhibit higher word counts (50k and 39k+), indicative of rich morphology. In contrast, Bhojpuri (`bh`) and Chhattisgarhi (`ch`) have smaller vocabularies, possibly due to lower lexical variation. Character counts (63–72) align with script complexity, while phoneme inventories (around 50–55) are consistent with Indo-Aryan and Dravidian phonological systems.

These statistics highlight the linguistic richness and dialectal coverage of the RESPIN text corpus. The balanced representation across dialects and domains, together with detailed lexicons, provides a strong foundation for multilingual and multidialectal ASR, language modeling, and speech-language research.

Table 3: Dialect-wise duration (in hours) across Clean, Semi-noisy, and Noisy subsets for 9 Indian languages.

| Dialect | Type | bh | bn | ch | hi | kn | mg | mr | mt | te |
|---|---|---|---|---|---|---|---|---|---|---|
| D1 | Clean | 351.25 | 206.40 | 344.89 | 205.25 | 237.38 | 340.88 | 312.58 | 195.10 | 348.78 |
| | Semi-noisy | 32.09 | 64.61 | 31.75 | 49.35 | 58.72 | 15.12 | 63.69 | 117.80 | 51.61 |
| | Noisy | 41.43 | 1.31 | 21.07 | 80.67 | 52.81 | 17.60 | 61.15 | 71.01 | 41.96 |
| D2 | Clean | 417.74 | 271.45 | 329.20 | 159.78 | 245.03 | 349.01 | 328.89 | 112.16 | 333.28 |
| | Semi-noisy | 11.25 | 12.97 | 22.37 | 90.78 | 37.07 | 13.94 | 54.39 | 139.60 | 74.12 |
| | Noisy | 5.68 | 0.80 | 12.07 | 88.07 | 38.36 | 13.13 | 49.17 | 180.44 | 33.67 |
| D3 | Clean | 347.97 | 283.17 | 297.63 | 195.93 | 235.92 | 333.33 | 321.62 | 203.16 | 331.65 |
| | Semi-noisy | 62.53 | 22.55 | 77.19 | 70.51 | 55.35 | 26.11 | 60.99 | 164.29 | 58.34 |
| | Noisy | 29.46 | 1.10 | 22.81 | 68.28 | 44.17 | 14.87 | 23.49 | 55.73 | 54.49 |
| D4 | Clean | – | 216.14 | 324.25 | 138.83 | 248.10 | 321.18 | 316.39 | 212.64 | 290.27 |
| | Semi-noisy | – | 62.64 | 67.56 | 116.41 | 34.66 | 57.22 | 156.14 | 88.55 | 38.46 |
| | Noisy | – | 2.13 | 34.17 | 99.35 | 48.41 | 27.14 | 66.13 | 98.11 | 18.87 |
| D5 | Clean | – | 236.08 | – | 245.14 | 228.13 | – | – | – | – |
| | Semi-noisy | – | 27.19 | – | 35.74 | 64.40 | – | – | – | – |
| | Noisy | – | 1.39 | – | 54.49 | 42.48 | – | – | – | – |
| Total | Clean | 1116.96 | 1213.24 | 1295.97 | 944.93 | 1194.56 | 1344.40 | 1279.48 | 723.06 | 1303.98 |
| Total | Semi-noisy | 105.87 | 189.96 | 198.87 | 362.79 | 250.20 | 112.39 | 335.21 | 510.24 | 222.53 |
| Total | Noisy | 76.57 | 6.73 | 90.12 | 390.86 | 226.23 | 72.74 | 199.94 | 405.29 | 148.99 |

## 4.2 Audio Data Analysis

### 4.2.1 Slab-Wise Audio Distribution

Table 3 summarizes dialect-wise audio durations across the Clean, Semi-noisy, and Noisy slabs for all nine languages. The corpus contains over 12,000 hours of read-speech audio spanning more than 20,000 sentences per language. Based on transcription quality and alignment confidence (see Section 3.3.1), audio is grouped into three slabs: *Clean*, *Semi-noisy*, and *Noisy*.

The collection goal was 200 hours of clean data per dialect for languages with five dialects (e.g., Hindi, Bengali, Kannada) and 250 hours per dialect for those with four dialects (e.g., Magahi, Marathi, Telugu). Most dialects met these targets, particularly in Bengali, Chhattisgarhi, Kannada, and Marathi. Some under-resourced dialects (e.g., Hindi D2, D4, and Maithili D2) fell short, requiring higher proportions of semi-noisy and noisy data to ensure sufficient coverage. These shortfalls likely reflect challenges in recruiting fluent readers in specific dialects due to literacy variation, regional accessibility, and dialectal overlap. For instance, Maithili and Hindi show lower clean-slab totals (723.06 and 944.93 hours, respectively) compared to other languages that exceed 1100 hours.

Across the full dataset, the clean slab totals 10,416.58 hours, semi-noisy 2,288.06 hours, and noisy 1,617.47 hours. The inclusion of noisy subsets captures real-world transcription variability and supports ASR training under practical conditions. This stratification balances dialectal coverage with data quality, enabling robust model evaluation across varying acoustic and transcription conditions.

### 4.2.2 Signal-Level Audio Quality

Table 4: Audio statistics per language including low SNR and speaking rate.

| LID | #Files | #Low SNR | %SNR | Wds/Aud | Dur (s) | WPM |
|---|---|---|---|---|---|---|
| bn | 870,793 | 3712 | 0.43 | 9 | 4.18 | 142.00 |
| bh | 866,619 | 4404 | 0.51 | 10 | 3.94 | 159.37 |
| ch | 823,803 | 1605 | 0.19 | 12 | 4.87 | 161.18 |
| hi | 756,886 | 1686 | 0.22 | 11 | 3.81 | 173.91 |
| kn | 744,617 | 1749 | 0.23 | 8 | 4.84 | 110.16 |
| mg | 968,365 | 2981 | 0.31 | 10 | 4.25 | 153.97 |
| mt | 518,504 | 1144 | 0.22 | 10 | 3.87 | 150.73 |
| mr | 1,002,599 | 2055 | 0.20 | 8 | 4.27 | 132.66 |
| te | 895,131 | 3051 | 0.34 | 8 | 4.40 | 117.16 |

**Abbreviations:** LID = Language ID; #Files = No. of audio files; #Low SNR = No. of low-SNR files (SNR < 4 dB); %SNR = Percentage of low-SNR files; Wds/Aud = Avg. words per audio; Dur (s) = Avg. duration in seconds; WPM = Words per minute.

Table 5: Train, development, and test set statistics for each language.

| LID | #Dialects | Train Set | | | | Dev Set | | | | Test Set | | | |
|-----|-----------|-----------|---|---|---|---------|---|---|---|----------|---|---|---|
| | | Dur (h) | #Utts | #Sents | #Spks | Dur (h) | #Utts | #Sents | #Spks | Dur (h) | #Utts | #Sents | #Spks |
| bh | 3 | 142.98 | 95280 | 19056 | 1445 | 2.14 | 1500 | 575 | 60 | 3.10 | 2220 | 694 | 120 |
| bn | 5 | 142.96 | 85800 | 17160 | 1280 | 2.27 | 1500 | 494 | 100 | 3.26 | 2174 | 648 | 200 |
| ch | 4 | 175.22 | 85800 | 17160 | 1586 | 2.40 | 1413 | 511 | 80 | 3.85 | 2234 | 695 | 160 |
| hi | 5 | 128.47 | 85800 | 17160 | 2172 | 2.21 | 1539 | 722 | 100 | 3.30 | 2288 | 853 | 201 |
| kn | 5 | 164.83 | 85800 | 17160 | 1859 | 2.37 | 1430 | 518 | 100 | 3.61 | 2161 | 663 | 200 |
| mg | 4 | 157.77 | 95280 | 19056 | 1493 | 2.10 | 1431 | 494 | 80 | 3.17 | 2193 | 640 | 160 |
| mt | 4 | 159.32 | 95280 | 19056 | 1913 | 2.06 | 1409 | 693 | 80 | 3.33 | 2172 | 993 | 160 |
| mr | 4 | 140.49 | 95280 | 19056 | 2305 | 1.98 | 1386 | 509 | 80 | 3.04 | 2170 | 711 | 160 |
| te | 4 | 155.89 | 95280 | 19056 | 1848 | 2.30 | 1438 | 500 | 80 | 3.37 | 2226 | 652 | 160 |

**LID**: Language ID, **#Dialects**: number of dialects, **Dur**: duration in hours, **#Utts**: number of utterances, **#Sents**: number of unique sentences, **#Spks**: number of speakers.

Table 4 presents signal-level quality metrics for clean-slab data, including the number and proportion of low-SNR files, average words per audio, average duration, and speaking rate (words per minute). Each audio was trimmed using forced-alignment timestamps to remove leading and trailing silence or prompts. SNR was computed using the pre-trained FB-Denoiser [41], with 4 dB chosen empirically as the threshold for low-SNR classification. Speaking rate was calculated as the ratio of transcript word count to aligned duration.

Although contributors were instructed to record in quiet environments, the crowdsourced nature of data collection introduced acoustic diversity. The corpus includes 10,416 hours of clean, 2,288 hours of semi-noisy, and 1,617 hours of noisy audio, with fewer than 1% of clean files classified as low-SNR.

### 4.2.3 Speaker Metadata Validation

Speaker metadata quality was assessed using two validation checks: (1) intra-speaker and (2) inter-speaker consistency. The intra-speaker check identified discrepancies within a single speaker's recordings, while the inter-speaker check detected potential overlaps between recordings assigned to different speaker IDs. To address these issues, we developed a bucketization algorithm validated on unseen data (see Appendix F). The algorithm successfully resolved 99.28% of intra-speaker inconsistencies and 52.91% of inter-speaker mismatches, providing a reliable measure of speaker identity consistency. After this validation, speakers without any discrepancies were selected for the development and test sets, ensuring no overlap across train, dev, and test splits (see Appendix G for more details on train-dev-test splits).

## 5 Benchmarking ASR Performance

### 5.1 Datasets

To enable reproducible research and fair comparison, we release standardized train, development, and test splits for all nine languages. Table 5 summarizes duration, utterances, unique sentences, and speakers. Each language contains 3–5 dialects and roughly 130–175 hours of training audio with 85k–95k utterances. The dev and test sets contain 2–4 hours each and up to 2.2k utterances from 60–200 speakers. The train set reported in Table 5 is the *small* balanced subset of the *clean* corpus used for all experiments in this paper. For mt_D2, where clean audio was insufficient, a small portion of semi-noisy audio was included. Additional variants are provided in Appendix G.

All dev and test sets are drawn from the *uncontaminated* speaker bucket (Section 4.2.3), ensuring no speaker overlap with training and preserving dialectal balance and sentence diversity across splits.

### 5.2 Models

We evaluate a range of ASR systems: (i) traditional models trained from scratch on RESPIN subsets, (ii) multilingual and SSL models pretrained on external data and used without RESPIN fine-tuning, and (iii) the same pretrained models fine-tuned on RESPIN. Concretely, we use TDNN-HMM (Kaldi) and an E-Branchformer CTC/attention system (ES-

Table 6: CER and WER (%) for different models across languages. **Pretrained models** refer to models fine-tuned on publicly available data other than RESPIN. **Traditional models** are trained from scratch on RESPIN. **Fine-tuned models** are pretrained SSL or Whisper models further fine-tuned on a subset of RESPIN. For SeamlessM4T-v2-Large, **bh**, **ch**, and **mg**, and for the pretrained SSL models, **bh**, **ch**, **mg**, and **mt** are evaluated using Hindi-tuned models.

| Model | CER (%) | | | | | | | | | | WER (%) | | | | | | | | | |
|---|---|---|---|---|---|---|---|---|---|---|---|---|---|---|---|---|---|---|---|---|
| | bh | bn | ch | hi | kn | mg | mr | mt | te | avg | bh | bn | ch | hi | kn | mg | mr | mt | te | avg |
| **Pretrained Models (fine-tuned on non-RESPIN public data)** | | | | | | | | | | | | | | | | | | | | |
| SeamlessM4T-v2-Large | 29.09 | 17.54 | 33.20 | 15.34 | 18.91 | 30.07 | 14.44 | 27.15 | 14.33 | 22.23 | 56.77 | 45.56 | 71.86 | 25.43 | 55.38 | 56.49 | 42.09 | 66.64 | 46.11 | 51.81 |
| IndicW2V | 17.08 | 14.27 | 22.77 | 11.02 | 10.37 | 19.64 | 15.09 | 23.30 | 8.61 | 15.80 | 51.61 | 42.83 | 65.98 | 28.34 | 42.37 | 54.32 | 53.91 | 66.10 | 37.82 | 49.25 |
| SPRING-W2V2 | 15.10 | 12.50 | 20.81 | 8.80 | 11.43 | 16.35 | 7.56 | 20.12 | 6.97 | 13.29 | 41.32 | 25.93 | 55.42 | 22.99 | 44.35 | 42.09 | 34.15 | 53.69 | 36.32 | 39.58 |
| SPRING-Data2Vec-AQC | 15.02 | 11.94 | 21.26 | 7.20 | 10.78 | 15.81 | 7.49 | 19.91 | 6.53 | 12.88 | 42.35 | 23.69 | 56.17 | 20.93 | 42.79 | 42.47 | 33.40 | 53.65 | 33.98 | 38.83 |
| **Traditional Models (trained from scratch on RESPIN subset)** | | | | | | | | | | | | | | | | | | | | |
| TDNN-HMM | 5.67 | 5.22 | 4.45 | 3.25 | 4.88 | 7.69 | 3.30 | 6.53 | 3.94 | 4.99 | 17.57 | 16.87 | 12.69 | 8.72 | 23.01 | 22.33 | 13.40 | 20.13 | 20.81 | 17.28 |
| E-Branchformer | 4.95 | 4.33 | 3.63 | 3.52 | 4.62 | 6.68 | 3.19 | 5.75 | 3.97 | 4.52 | 15.21 | 14.96 | 10.59 | 9.94 | 24.50 | 20.38 | 14.48 | 17.95 | 21.64 | 16.63 |
| **Fine-tuned Models (fine-tuned on RESPIN subset)** | | | | | | | | | | | | | | | | | | | | |
| Whisper-Tiny | 9.62 | 11.60 | 7.13 | 9.69 | 12.62 | 13.98 | 9.15 | 10.73 | 11.43 | 10.66 | 27.45 | 32.51 | 20.81 | 21.71 | 48.54 | 36.40 | 30.93 | 31.96 | 41.61 | 32.44 |
| Whisper-Base | 7.15 | 7.69 | 5.36 | 5.80 | 8.10 | 10.44 | 6.23 | 7.51 | 7.51 | 7.31 | 22.51 | 24.71 | 16.67 | 15.19 | 36.52 | 30.54 | 24.28 | 24.80 | 32.99 | 25.36 |
| Whisper-Small | 7.90 | 5.46 | 3.85 | 4.16 | 6.00 | 7.46 | 3.93 | 5.94 | 6.54 | 5.69 | 19.02 | 18.91 | 12.36 | 11.78 | 29.66 | 23.94 | 16.95 | 20.28 | 27.82 | 20.08 |
| IndicW2V | 4.42 | 4.28 | 3.24 | 3.16 | 4.68 | 6.02 | 3.19 | 5.19 | 4.54 | 4.30 | 16.07 | 16.65 | 11.36 | 10.47 | 24.86 | 21.51 | 15.13 | 19.19 | 24.03 | 17.69 |
| SPRING-W2V2 | 3.92 | 3.86 | 2.99 | 2.37 | 4.30 | 5.20 | 2.49 | 4.37 | 3.85 | 3.71 | 14.61 | 15.12 | 10.74 | 8.22 | 23.90 | 19.40 | 12.75 | 16.64 | 21.92 | 15.92 |
| SPRING-Data2Vec-AQC | 3.95 | 3.63 | 2.84 | 2.27 | 4.11 | 4.98 | 2.38 | 4.30 | 3.72 | 3.58 | 14.84 | 14.15 | 10.25 | 7.91 | 23.13 | 18.50 | 12.28 | 16.41 | 21.17 | 15.40 |

Pnet), Whisper models (Tiny, Base, Small), IndicWav2Vec, and two SPRING SSL models (Wav2Vec2 and Data2Vec-AQC).

## 5.3 Experimental Setup

All experiments were run on a single NVIDIA RTX 3090 GPU (24 GB).

**Whisper** We fine-tune Tiny (39M), Base (74M), and Small (244M) variants using Hugging Face checkpoints with the Trainer API and early stopping on dev WER. Decoding conditions include the language ID.

**Fairseq SSL** We fine-tune IndicWav2Vec[6], SPRING-Wav2Vec2, and SPRING-Data2Vec-AQC[7] on RESPIN.

**ESPnet** We train an `e_branchformer` encoder (8 blocks, 256 hidden), CTC/attention criterion, Adam optimizer, SpecAugment, AMP, and early stopping on dev CER.

**Kaldi** We train TDNN-HMM using the chain recipe with 40-dim MFCCs, i-vectors, speed and volume perturbation, and a trigram LM trained on RESPIN transcripts.

All training recipes and checkpoints are available at `https://github.com/labspire/respin_baselines`.

## 5.4 Results and discussion

Table 6 reports CER and WER for nine languages. The results highlight the value of dialect-aware supervision.

**Pretrained models without RESPIN supervision** Models trained only on external data, such as SeamlessM4T-v2-Large, IndicWav2Vec (pretrained), and SPRING-Wav2Vec2 (pretrained), show high error rates for several languages, especially those with strong dialectal variation such as Bhojpuri and Chhattisgarhi. This gap reflects domain and dialect mismatch.

**Training from scratch on RESPIN** Traditional systems trained solely on RESPIN subsets outperform the above. E-Branchformer achieves an average WER of 16.63%, underscoring the benefit of dialect-specific supervision even without large-scale pretraining.

**Whisper fine-tuning** Fine-tuned Whisper models improve over their pretrained counterparts but generally remain behind scratch-trained E-Branchformer, indicating limited adaptation to dialectal nuances.

**Fine-tuned SSL models** SSL models fine-tuned on RESPIN perform best overall. SPRING-Data2Vec-AQC attains the lowest average WER (15.40%), and SPRING-

---

[6] `https://github.com/AI4Bharat/IndicWav2Vec`

[7] `https://asr.iitm.ac.in/models`

Wav2Vec2 is consistently strong, showing that SSL pretraining combined with dialect-aware fine-tuning is effective for multi-dialect ASR.

**Generalization to public test sets** We also evaluate on CommonVoice, FLEURS, Gram-Vaani, IndicTTS, Kathbath, and MUCS for `bn`, `hi`, `kn`, `mr`, and `te`. Pretrained models are slightly stronger on these non-domain-specific sets, yet RESPIN-fine-tuned SSL models remain competitive. Full results are provided in Appendix I.

## 6 Applications, Impact, and Limitations

RESPIN-S1.0 has been actively used in community challenges and research benchmarks. Over the past two years, subsets of the corpus have supported multiple workshops, challenges, and research efforts. A Bengali and Bhojpuri subset was used in the SLT Code Hackathon 2022 to build dialect-aware ASR systems. The first Multi-Dialect ASR Challenge (MADASR) was organized at ASRU 2023 [42, 43] using RESPIN data for Bengali and Bhojpuri. The ongoing MADASR 2.0 Challenge at ASRU 2025 expands this to 1,200 hours across eight languages (`bh`, `bn`, `ch`, `kn`, `mg`, `mr`, `mt`, `te`), enabling large-scale benchmarking of dialect-aware ASR systems. RESPIN has also been used for dialect identification across eight Indian languages [44, 45]. Beyond ASR, the corpus facilitates research in language and dialect identification (LID/DID), unsupervised speech translation, and other multilingual speech-language processing tasks. Its focus on agriculture and finance provides valuable coverage of socially relevant domains, particularly for underrepresented Indian languages.

Despite its scope, RESPIN-S1.0 has certain limitations. The current release includes only read speech, whereas spontaneous and conversational data are more reflective of real-world communication. Its domain coverage is limited to agriculture and finance, and future expansions into healthcare, education, and governance would enhance applicability. Finally, the reliance on literate native speakers with smartphone access may underrepresent marginalized communities. Nonetheless, RESPIN establishes a strong foundation for inclusive, dialect-rich ASR development in India, and future releases will expand linguistic coverage and include spontaneous speech to address these limitations.

## 7 Conclusion and Future Work

In this work, we introduced **RESPIN-S1.0**, a large-scale, dialect-rich speech corpus spanning nine Indian languages and two socially relevant domains—agriculture and finance. By integrating dialectal, phonetic, and demographic diversity at scale, RESPIN establishes a unified benchmark for automatic speech recognition (ASR) and related speech-language processing tasks in low-resource, multilingual settings. The corpus is accompanied by standardized train–development–test splits, dialect-aware lexicons, detailed metadata, and multiple ASR baselines to enable transparent and reproducible research. Beyond improving ASR performance across dialects, RESPIN-S1.0 provides a foundation for systematic research in dialect identification, multilingual speech translation, and cross-domain adaptation. Future releases will expand coverage to additional domains such as healthcare, education, and governance, and incorporate spontaneous and conversational speech. The dataset will also include new dialects and languages and will introduce open benchmark suites for dialectal ASR and DID evaluation. We further plan to explore integration with large multilingual and self-supervised models to advance inclusive speech technologies for Indian languages. Through open data, transparent benchmarks, and continued community collaboration, RESPIN aims to accelerate equitable speech technology development across India's diverse linguistic landscape.

## Acknowledgements

This work was supported by the Gates Foundation. We thank the numerous NGOs, volunteers, and contributors who participated in data collection, validation, and community outreach, both online and offline. Their collaboration and commitment were instrumental in realizing RESPIN-S1.0.

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

# Appendices

## A   Language and Dialect Selection Process

This appendix describes the systematic procedure used to select representative dialects for nine Indian languages (Bhojpuri, Magahi, Maithili, Bengali, Kannada, Chhattisgarhi, Telugu, Marathi, and Hindi). Our four-step approach targeted, for each language, the identification of 3–5 dialects that jointly cover 70–90% of native speakers while ensuring linguistic diversity. Selections were validated by expert linguists to confirm coverage and representativeness of the chosen dialects and districts.

**Step 1: Literature survey**   We compiled major dialects per language and mapped their geographic distribution, identifying the core districts where each dialect is spoken.

**Step 2: Dialect selection**   We chose dialects that (i) together cover 70–90% of native speakers and (ii) are distinctive in structure/lexicon/phonology while being sufficiently resourced for collection.

**Step 3: District selection**   For each dialect, we prioritized districts reflecting the standard/local norm of the dialect, minimizing overlap with other dialect regions to avoid duplicate speakers, and considering operational feasibility.

**Step 4: Expert validation**   Language experts/linguists reviewed and validated the dialect–district choices for coverage and representativeness.

Based on the dialect selection criteria mentioned above, Table 7 lists the selected dialects along with their core districts and the rationale for inclusion, illustrating how each contributes to the overall dialectal diversity of the corpus. Key considerations during the finalisation of dialects are summarised below:

- **Bhojpuri:** Nagpuri was excluded as it is now recognized as a separate language.
- **Magahi:** The Eastern Magahi cluster was excluded due to internal variation, while Maithili-mixed Magahi ("Angika") was reclassified under Maithili.
- **Maithili:** Bajjika and Angika were retained, though both could also be classified as separate languages.
- **Bengali:** Eastern and south-eastern dialects were excluded since they are primarily spoken in Bangladesh.
- **Chhattisgarhi:** The Rakshahun (Southern) dialect was excluded due to its smaller speaker population.
- **Marathi:** Zadi Boli was excluded due to operational challenges in data collection.
- **Hindi:** Owing to the large number of Hindi dialects (50+ according to the 2011 Census), accent-based variation was prioritized over fine-grained dialectal distinctions to ensure coverage of 70–90% of Hindi speakers.

## B   Text Data Preparation and Validation Pipeline

The text data collection was carried out in two distinct phases. In **Phase 1**, a minimum of 5,000 sentences per language were collected, consisting primarily of interrogative sentences. In **Phase 2**, an additional ~15,000 sentences were gathered, resulting in at least 20,000 sentences per language. These sentences were uniformly distributed across dialects and domains—*agriculture* and *finance*—and included all major sentence types, with a particular focus on maximizing dialectal coverage.

Table 7: Language-wise Dialect and Districts Selected

| Language | Dialect | Core Districts | Justification |
|---|---|---|---|
| Bhojpuri | Southern Standard | Saran | Representative of standard variety |
| | Northern | East Champaran, Deoria | Captures sub-varieties in Bihar and UP |
| | Western | Varanasi | Major urban center for Western dialect |
| Magahi | Central Variety | Gaya, Jahanabad | Representative of standard Magahi |
| | Southern | Jamui, Lakhisaray, Nawada | Distinct from central variety |
| | Western | Vaishali | Shows Bhojpuri influence |
| | NE (Surjapuri) | Kishanganj, Purnia, Katihar | Distinct northern variety |
| Maithili | Standard (Sotipura) | Darbhanga, Madhubani | Representative of standard variety |
| | Bajjika | Samastipur, Saharsa | Distinct morphological features |
| | Eastern (Thēthi) | Araria, Madhepura | Eastern variety with distinct features |
| | Angika | Bhagalpur | Originally classified under Magahi |
| Bengali | Western | Purba/Paschim Medinipur | Shows Odia influence |
| | Varendri/Pundra | Malda (Core), Dakshin Dinajpur | Northern variety |
| | Rajbangsi | Jalpaiguri, Cooch Behar | Distinct northern dialect |
| | Jharkhandi | Purulia, Bankura | Variety spoken in Jharkhand region |
| | Standard | Kolkata, Nadia/Hooghly | Standard variety |
| Kannada | Hyderabad | Bellary | Formerly classified as Central & Hyderabad Karnataka |
| | Mangalore | Dakshin Kannada (Mangalore) | Coastal variety |
| | Dharwad | Dharwad, Uttar Kannada | North Western variety |
| | NE | Gulbarga | Shows strong Urdu influence |
| | Mysore | Mysore Rural, Mandya | Southern standard variety |
| Chhattisgarhi | Kedri (Central) | Bilaspur, Durg | Central standard variety |
| | Utti (Eastern) | Raigarh | Eastern variety |
| | Budati/Khatahi | Kabirdham, Balaghat | Western variety |
| | Bhandar | Sarguja | Northern variety |
| Telugu | Mid-Coastal | Guntur, Krishna | Central variety |
| | Rayalseema | Chittoor, Anantpur | Southern variety |
| | Telangana | Karimnagar, Nalgonda | Northern variety |
| | Utterandhra | Vishakapattanam, Srikakulam | Eastern variety |
| Marathi | S Konkan | Sindhudurga | Coastal south |
| | N Konkan | Dhule, Nashik | Coastal north |
| | Varhadi | Nagpur Rural | Eastern variety |
| | Standard | Pune Rural | Standard variety |
| Hindi | Hindustani+Malvi | Muzaffarnagar | Phonological similarities |
| | Kannauji+Braj | Etah | Transitional district with speakers of both dialects |
| | Awadhi+Bundeli | Hamirpur | Transitional district with speakers of both dialects |
| | Marwari+Dhundhari | Nagaur | Phonological similarities |
| | Garhwali | Tehri Garhwal | Distinct variety requiring separate collection |

## B.1 Phase 1: Initial Collection and Validation

### B.1.1 Domain Identification and Topic Mapping

To ensure domain relevance, data were systematically collected from the agriculture and finance sectors following a structured topic-selection process. The finalized topics informed the creation of domain-specific Google Forms, each containing a standardized set of questions in the nine target languages. This process involved comprehensive market research to identify existing agricultural and financial service providers, cataloging the features and data structures of current applications, analyzing products relevant to the Indian market, and evaluating mobile and web applications designed for low-literacy users. The final step included assessing interaction topics and advisory systems to identify potential real-world deployment scenarios for conversational agents derived from the collected data.

Lexical resources were developed in parallel through keyword mining, identification of topic clusters within the Indian agricultural and financial ecosystems, and creation of semantic categorization frameworks to organize the collected terminology. These frameworks guided the design of the Google Forms. Table 8 and Table 9 list the subtopics finalized for the agriculture and finance domains, respectively.

### B.1.2 Text Collection from Domain Experts

Dialect-specific domain experts contributed text data through standardized Google Forms. Each submission was reviewed by the Validation Team before being approved for the voice-collection phase. Figure 5 shows a sample form for the subtopic "Climate and Weather" in Hindi.

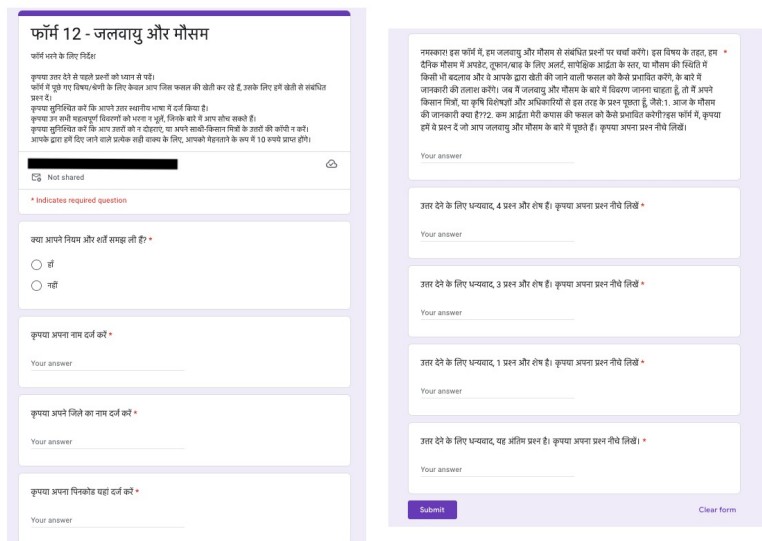

Figure 5: Sample Google Form used for Phase 1 text data collection.

### B.1.3 Text Validation and Sanitization

All text submissions underwent a rigorous multi-stage validation pipeline. Only linguistically verified entries advanced to the voice-collection stage.

**Step 1: Preliminary Linguistic Validation.** Submissions were first screened for linguistic quality. Those not meeting the required standards were rejected with feedback. Accepted entries were passed to the detailed sentence validation and sanitization process. The authenticity check performed by language resource managers ensured that sentences were domain-relevant, vocabulary-rich, and conversationally natural, representing realistic expert–user communication. A parallel linguistic diversity check by computational linguists

Table 8: Finalised Agriculture Domain sub-topics

| Sl. No. | Category Name | #Questions | Keywords | Example Questions |
|---|---|---|---|---|
| 1 | Crop Names and Crop seasons | 5 | Jowar, Bajra, rabi season and kharif season | 1. Can I plant rapeseed in Rabi season?
2. What months fall under the Late Kharif season? |
| 2 | Seeds, Seed Names, Varieties and Hybrids | 5 | Mysore Mallige, NMS2, Chinna Ponnu | 1. What is the best high yield variety in Basmati Rice?
2. Is there a drought resistant Ragi variety? |
| 3 | Soil name/Soil type | 5 | Salt Affected soil, Alkali Soil, Saline soil | 1. How do I treat alkaline soil?
2. Can I plant legume in red soil? |
| 4 | Farming methods | 5 | Hydroponics, Organic farming, Precision Farming, Irrigation, Shifting cultivation | 1. How do I grow vegetables in low cost organic farming?
2. What is the advantage of crop rotation? |
| 5 | Crop growth stage | 5 | Sprouting, Sapling, Flowering, Fruiting, Ripening | 1. How do I make my marigold blossom faster?
2. How do I prevent cross-pollination? |
| 6 | Nutrient Management | 5 | Vermicompost, organic soil preparation, Soil Nutrients, Manure, Fertilisers | 1. What can I do to increase humus content in my soil?
2. What is the ratio of NPK to use for 1 acre farmland? |
| 7 | Irrigation systems | 5 | Micro Irrigation Systems & Parts, Drip Lines, Laterals, Filters, Sprinkler Irrigation Systems & Parts, Drip Irrigation Kits, Rain Irrigation system | 1. Is sprinkler irrigation good for tomatoes?
2. What motor can I use for watering the paddy? |
| 8 | Pest Infestations | 5 | Pest names, pest symptoms | 1. What are the leaf curl symptoms?
2. What are the symptoms of fungal disease in my chilli crop? |
| 9 | Pest Management | 5 | Fungicides, Organic Fertilizer, Insecticide, Plant Growth Stimulator, Virucide, Bactericide, Nematicide | 1. What are the different types of pheromone traps used in cotton for pest control?
2. What are the preventive measures for bollworm? |
| 10 | Tools and Equipment | 5 | Tools and equipments available in the market, price of equipments, advice on using the best tool for cultivation, instruction on use. | 1. What is a good tractor to use for cutting maize crops?
2. Which is the best weeder for chilli and where it is available? |
| 11 | Weight and Measurements | 5 | Weight and measurement keywords used commonly by farmers. | 1. What is the labour charge for unloading one MT (Metric ton) of chillies?
2. How many kilograms of urea to be applied per acre of chilli cultivation? |
| 12 | Climate and Weather | 5 | Climate prediction for a region, daily weather, weather alerts (eg: cyclone, thunderstorm), Humidity levels | 1. What is the weather information today?
2. How will low humidity impact my cotton crop? |
| 13 | Financial aid and Schemes for farmers | 5 | Names of schemes/policies available to farmers, applying for financial aid, eligibility | 1. Is a small farmer like me eligible for crop insurance? How to get it?
2. Which government scheme will cover the cost of buying my machines? |
| 14 | Post-Harvesting Techniques | 5 | Processing of produce, storage, transportation, harvest processing | 1. How do I ensure that my produce contains no moisture?
2. How long can I store my cotton produce in the warehouse before it starts to degrade? |

Table 9: Finalised Financial Services Domain sub-topics

| Sl. No. | Category Name | #Questions | Keywords | Example Questions |
|---|---|---|---|---|
| 1 | Balance Checking / Manage Accounts | 6 | Balance Check, view transaction history, mini statement, view and download a/c statement for a specific date range, view a/c summary (customer ID, MMID, IFSC, branch name etc), view all a/cs (savings a/c, loan a/c) | 1. What is my balance? 2. Can you tell me the account number for my zero-balance account? |
| 2 | Fund Transfer | 6 | inter-banks or intra-bank transfer/beneficiary, enter IFSC/MMID for transfer to non-beneficiary, IMPS/NEFT/RTGS mode of transfer, view and delete beneficiaries, view daily payment limit, register for net banking, get login ID & password | 1. How can I send money to my sister from another city? 2. What is the meaning of RTGS? |
| 3 | Loan Eligibility | 6 | Checking Credit score/CIBIL, Credit assessment for the poor, Suret. EMI projection | 1. How can I check if I am eligible to get an agricultural loan? 2. Will I get a loan without collateral? |
| 4 | Loan Information and Loan Application | 6 | apply/request for gold loan/personal loan/ agriculture loan/ home/two-wheeler/car loan online, get online loan application | 1. Can I get a loan for installing solar devices in my farm? 2. Can I get a loan for sending my child to school |
| 5 | Loan Repayment | 6 | View EMI Calendar,make monthly installment/repay loan, payment methods and view loan details, view loan a/c statement, view outstanding loans | 1. What date do I need to repay my loan for this month? 2. How much is my EMI amount? |
| 6 | Savings a/c opening & KYC updation | 6 | instantly open digital salary and savings a/c, select a/c type -regular savings/basic savings/women's savings/senior citizen's savings a/c, track application status, update KYC with Aadhar during a/c opening, convert existing a/c to salary a/c | 1. Can I open an account in the bank without a PAN card? 2. Do you need address proof for bank account activation? |
| 7 | Debit/Credit Cards | 6 | reset debit card PIN, block debit card, hotlist debit card, set transaction limit, enable/disable debit card, activate cardless cash facility to withdraw cash from ATMs without debit card apply for credit card, view customer care number provided on the app to digitally apply for credit card , view total credit utilized, view total due, view due date, pay your due bills, view virtual card, block and replace card | 1. I forgot my ATM pin, What to do? 2. Can I apply for new debit card? |
| 8 | Deposits & Investments | 6 | select FD/RD/tax saving schemes to apply, Open new FD/RD/tax saver deposits, view & redeem deposit, view interest rates, calculate deposit interest, apply for mutual fund, download deposit certificates, convert RD to FD on maturity, download deposit slips, Gold Bonds | 1. Can I install in an RD for better interest? 2. What is the minimum closing period in FD? |
| 9 | Utility bills & Payments | 6 | Pay cable TV/DTH/broadband/electricity /FASTag/Gas/landline/water bill, recharge mobile postpaid & prepaid, pay taxes | 1. I want to top-up my mobile 2. Can I recharge my D2H connection? |
| 10 | Insurance | 6 | E-insurance a/c opening, view plans- accident cover/health/travel/home insurance plans/life insurance, pay periodical premium | 1. What are the benefits of having insurance? 2. Will my family benefit by me having health insurance? |
| 11 | Social sector schemes | 6 | Government's PMJJBY scheme, know more about the scheme, open NPS a/c, submit pensioner life certificate, enroll for Atal pension yojana/senior citizen savings scheme | 1. What is the procedure to apply to PMJJBY scheme? 2. Am I eligible for PMKY? |
| 12 | UPI Services | 6 | Register with mob no, add bank a/c details- select bank/enter OTP for SMS verification/enter ATM card details, create UPI PIN, transfer fund to own a/c, contacts using the app, a/c numbers, pay by scanning QR code | 1. What is UPI ID? 2. Are online transactions safe? |
| 13 | Alternate / Non-Banking Financial Services | 6 | NBFC, Microfinance banks, SME loans, Small Purpose loans - Sanitation loans, water loans, Festival loans etc delimited to rural residents | 1. Can I get a loan for starting my tailoring business? 2. Can a single woman like me get a loan? |

verified variation in terminology, dialectal coverage across regional and socio-geographic varieties, demographic balance in contributors (age, literacy, gender), and documentation of sociolinguistic patterns relevant to ASR development.

**Step 2: Systematic Validation Protocol.** Before validation began, linguistic consultants (LCs) received detailed orientation on the data-collection methodology, form structure, domain-specific terminology, and expected response formats. During initial review, validators identified common issues such as non-interrogative responses, script inconsistencies, repeated submissions, and transliteration errors. For each Google Form, LCs then conducted systematic verification to produce grammatically correct regional-language sentences paired with English translations. This included verifying question formats, removing off-domain or duplicate entries, checking dialect authenticity, and normalizing grammar, spelling, and orthography. Ambiguous cases were flagged for expert resolution.

Quality assurance followed through cross-verification of validator corrections, comparison between manual and automated translations, documentation of recurring error patterns to improve the pipeline, and compensation based on validated submissions.

**Step 3: Text Sanitization and Corpus Standardization.** After validation, the corpus underwent further cleaning and normalization. Texts were organized by domain, dialect, and language, and assigned to dialect-specific consultants for standardization. The process ensured consistent punctuation, capitalization, and formatting; normalized numerical expressions, abbreviations, and units; and re-categorized cross-domain sentences under the most appropriate subtopics. Transliteration management was implemented to maintain consistency for non-native terms, separate inflectional morphology from borrowed words, and document multiple transliteration variants for ambiguous cases.

Phonological completeness was ensured by comparing the monophone, diphone, and triphone distributions against language reference data. Expert linguists verified these distributions and proposed augmentations where coverage was insufficient. Additional sentences were then composed to fill phonological and lexical gaps, ensuring balanced representation across dialects. The resulting corpus thus achieved uniform linguistic quality, phonological coverage, and dialectal diversity across all languages.

Table 10 presents two Hindi examples from the Marwari region within the agricultural domain, as reviewed and corrected by the Validation Team.

Table 10: Illustration of Text data validation

| DE Name | Pin Code | District | Feature Category | Sentence provided by DE | Corrected sentence in Hindi by LC | Translation provided by LC |
|---|---|---|---|---|---|---|
| XXXX | 341319 | Nagaur | 1 - Crop Names & Seasons | अगेती रबी मौसम के अंतर्गत कौन से महीने आते हैं? | अगेती रबी मौसम के अंतर्गत कौनसे महीने आते हैं? | Which months fall under early rabi season? |
| XXXX | 341319 | Nagaur | 2 - Seeds, Varieties & Hybrids | मक्का की सबसे अच्छी उपज वाली किस्म कोनसी है? | मक्का की सबसे अच्छी उपज वाली किस्म कौनसी है? | Best yielding variety of maize? |

We programmed scripts to automatically tag issues, helping the Validation Team with their preliminary screening. Table11 shows an example for a single sentence to illustrate this process. In the actual validation workflow, the items generated for each field in every sentence are presented as columns in a Google Sheet. In the example provided in the table, the script tagged the following:

1. Special symbol: "?"

2. Abbreviation: "HP"

3. Numeric translation: "आठ" (meaning "eight")

4. Letter transliteration: "के" (this could be the English letter "K" or the Hindi postposition "के")

The data underwent multiple rounds of automatic issue tagging, followed by manual corrections by language consultants and a subsequent multi-stage review by the Validation Team.

Table 11: Illustration of automatic tagging of text for text data validation

| Field Type | Example Sentence 1 | Example Sentence 2 |
|---|---|---|
| LC Sentences | अगेती रबी मौसम के अंतर्गत कौनसे महीने आते हैं ? | आठ H.P की मोटर चलाने के लिए बिजली का कौनसा कनेक्शन लेना होगा ? |
| Transliteration | ageetii rabii mousam kee atargat koun see mahiinee aatee hei | aaṭh eecapii kii moṭar calaanee kee liee bijalii kaa kounasaa kaneekśan leenaa hogaa |
| Translations | Which months come under the early rabi season? | Which connectivity of electricity we should use to run eight H.P motor? |
| Special Symbols | ['?'] | [':', '?'] |
| Acronyms | [] | [] |
| Roman Numerals | [] | [] |
| Alphanumerics | [] | [] |
| Numerics | [] | [] |
| English Words | [] | [] |
| Abbreviations | [] | [H.P] |
| Letters | [] | [] |
| Letter Transliteration | ['के'] | ['के'] |
| Numeric Translation | [] | ['आठ'] |
| Non Whitespaces | [] | [] |

## B.2 Phase 2 — Corpus Expansion and Dialect Balancing

### B.2.1 Domain and Topic Coverage

Phase 2 focused on large-scale corpus expansion and dialectal balancing through a more diverse and comprehensive data collection approach. The process began with curating an extensive list of topics across the agriculture and finance domains. This initiative was designed to assist contributors—especially those unfamiliar with specific subjects—by offering structured prompts and reference material. The goal was to ensure exhaustive domain coverage, leaving no subtopic unexplored.

Topic compilation was performed manually using a variety of sources, including magazines, Wikipedia, specialized websites, and relevant academic and research literature. Magazines sourced from online archives, educational institutions, and research organizations enriched topic breadth, while the Wikipedia "Outline" pages provided structured hierarchies of subtopics and relevant reference links. The final list comprised approximately 1,500 topics, each linked to corresponding online references for contextual understanding. Compared to Phase 1, this represented nearly a 100-fold increase in topic diversity, enabling far greater linguistic and contextual variety. The process began with broad domain segmentation (e.g., agriculture and finance) and progressed toward finer granularity—from crop-specific topics such as sugarcane cultivation to technological and financial subtopics like UPI PIN setup or transaction history retrieval in digital banking applications.

### B.2.2 Sentence Preparation and Composition

The primary source of data in Phase 2 was the manual composition of sentences by trained language experts referencing the curated topic lists. In cases where the target of approximately 15,000 sentences per language was not achieved, supplementary strategies were adopted. These included web scraping for high-resource languages, translation from other dialects within the same language, and cross-language translation where appropriate.

Digital text resources in standard written forms, while voluminous, often lacked colloquial authenticity or dialectal variation. To address this, the collection strategy intentionally prioritized contributions from native speakers of specific districts. Engaging these individuals allowed the corpus to capture locally grounded expressions and speech patterns, thus ensuring linguistic authenticity and alignment with real-world spoken communication. This approach not only improved representativeness but also deepened the understanding of regional linguistic and cultural diversity.

Given the curated topics, sentence composers were instructed to generate conversational and colloquial examples tailored for Automatic Speech Recognition (ASR) model development. Managing dialectal variability was challenging, as linguistic differences could vary significantly even within a 5–10 km radius. However, this diversity was embraced as an advantage rather than a limitation. By accommodating regional variation and the absence of standardized orthography in several dialects, the corpus design intentionally reflected the natural heterogeneity of Indian speech. The resulting data thus embodies both linguistic flexibility and broad coverage.

To maintain uniformity and usability, contributors followed a standardized set of rules during composition:

1. **Character Length Limit:** Sentences should not exceed 120 characters to maintain readability and consistency.

2. **Avoid Pronoun Start:** Sentences should not begin with pronouns to ensure contextual coherence, as each sentence is treated independently.

3. **Numerical Representation:** All numbers must be written in words to enhance readability and prevent ambiguity.

4. **Special Characters:** Only full stops (.), commas (,), and question marks (?) are permitted; other special characters are disallowed for uniform formatting.

5. **Avoid Controversial Statements:** Content must remain neutral and non-political to ensure objectivity.

6. **Topic Adherence:** Each sentence must remain relevant to the assigned topic.

7. **Domain Balance:** Equal representation of agriculture and finance topics must be maintained across the dataset.

8. **Acronym Formatting:** Acronyms should follow a consistent format, such as A.T.M., ensuring clarity and uniform representation.

Adhering to these guidelines ensured a cohesive, balanced, and linguistically standardized text corpus suitable for ASR training and dialectal analysis.

**Translation Strategy.** To achieve the target of 15,000 sentences per language, a portion of the corpus was created via translation by expert linguists. Source sentences were drawn either from the same language's standard dialect or from another linguistically related language. These translations preserved semantic meaning while incorporating dialect-specific vocabulary and stylistic features, enhancing intra- and inter-language diversity.

### B.2.3 Text Corpus Validation

After sentence composition, the entire dataset underwent a comprehensive validation pipeline consisting of multiple automated and manual checks performed by language validators. Since sentences were produced by several contributors, inevitable inconsistencies and typographical deviations were corrected to ensure the text's suitability as stimuli for the

crowd-sourced speech recording interface. Sentences required to be accurate, unambiguous, coherent, and compatible with the recording application, making this validation pipeline a critical stage of quality control.

The pipeline architecture remained consistent across languages, with minor language-specific adaptations. Validation proceeded through successive rounds, each producing a versioned release of the dataset. This approach allowed traceability and rollbacks when required. Independent checks were executed within a single version, while dependent checks were performed sequentially.

The major categories of validation checks are summarized below:

1. **Duplicate Sentence Removal (Automatic):** A pairwise Word Error Rate (WER) analysis was applied to the raw corpus to identify and remove duplicate sentences, reducing redundancy before manual review.

2. **Invalid Character Check and Correction (Manual):** Non-printable and redundant whitespace characters were eliminated. Validators reviewed the character inventory of the corpus and manually corrected sentences containing non-language symbols (excluding allowed punctuation marks: comma, full stop, and question mark). The process was iterated until only valid characters remained.

3. **Sentence Length Pruning (Manual):** Due to recording-interface constraints, sentences exceeding 90 characters were either pruned or rejected by validators to maintain compatibility with UI display limits.

4. **Acronym Standardization (Manual):** Acronyms were required to follow the "A.B.C." format. Tokens containing full stops were extracted and validated to confirm correct acronym usage. Non-standard acronyms, including transliterated English terms, were flagged and reformatted.

5. **Invalid Matra Check and Correction (Manual):** Words containing redundant or incorrect matra usage—such as consecutive matras or visually overlapping diacritics—were flagged and manually corrected to ensure orthographic accuracy.

6. **Interchangeable Character Correction (Manual):** Validators referenced a curated list of commonly confused characters. Words containing such letters were manually reviewed for potential spelling errors and corrected as needed.

7. **Similar Sentence Check (Manual):** Near-duplicate sentence pairs with $0 < \text{WER} < 0.3$ were identified and manually reviewed. Validators retained, corrected, or removed variants depending on linguistic relevance.

8. **Homophone Check (Manual):** Using phonetic transcriptions provided by Navana Tech, phonetic WER was computed to detect homophones. Validators examined flagged pairs for potential spelling or pronunciation inconsistencies and corrected them.

9. **Language-Specific Checks (Manual):** Additional validations were implemented for particular languages to handle unique orthographic, script-level, or dialectal issues. Details of these custom checks are available in the language-specific corpus documentation.

Following validation, each version of the corpus was archived to maintain a complete record of revisions. Table 12 provides statistics for the Phase 2 text corpus, including the total number of sentences by domain (agriculture and finance), their method of creation (composition or translation), and the number and distribution of dialect experts involved. Table 13 presents additional dialect-level statistics, including total samples, Phase 1 contributions, vocabulary sizes, and average sentence lengths.

## C   Phonetic Lexicon Construction

To generate the pronunciation lexicon, we developed a deterministic, rule-based grapheme-to-phoneme (G2P) conversion pipeline following the sound label set creation process defined in the Indian Language Speech-sound Label set (ILSL) [6]. This tagset-based process ensures

Table 12: Summary of Sentence (Phase 2) Statistics and their Sources (Composed or Translated) by Language and Region

| Language | Dialect ID | Total Sentences | # Agri Sentences | # Finance Sentences | # Composed Agri | # Composed Finance | # Translated Agri | # Translated Finance | #Contributors (#Pincodes) |
|---|---|---|---|---|---|---|---|---|---|
| BENGALI | D1 | 3177 | 1794 | 1383 | 1794 | 1383 | 0 | 0 | 2 (1) |
| | D2 | 2908 | 1607 | 1301 | 1607 | 1301 | 0 | 0 | 2 (2) |
| | D3 | 3094 | 2002 | 1092 | 2002 | 1092 | 0 | 0 | 5 (4) |
| | D4 | 2991 | 1580 | 1411 | 1580 | 1411 | 0 | 0 | 1 (1) |
| | D5 | 4052 | 1001 | 3051 | 1001 | 3051 | 0 | 0 | 1 (1) |
| BHOJPURI | D1(NBH-EC) | 1883 | 1505 | 378 | 1505 | 378 | 0 | 0 | 2 (2) |
| | D1(NBH-DE) | 4425 | 2002 | 2423 | 2002 | 2423 | 0 | 0 | 1 (0) |
| | D2 | 5785 | 3050 | 2735 | 3050 | 1854 | 0 | 881 (Hindi) | 5 (3) |
| | D3 | 5385 | 3096 | 2289 | 3096 | 1024 | 0 | 1265 (Hindi) | 3 (3) |
| CHATTISGARHI | D1 | 4003 | 2007 | 1996 | 2007 | 1996 | 0 | 0 | 3 (2) |
| | D2 | 3981 | 1991 | 1990 | 1991 | 1990 | 0 | 0 | 1 (1) |
| | D3 | 3980 | 1998 | 1982 | 1998 | 1982 | 0 | 0 | 2 (2) |
| | D4 | 3492 | 2002 | 1490 | 2002 | 1490 | 0 | 0 | 2 (2) |
| Hindi | STD HN | 16131 | 8187 | 7944 | 8187 | 7944 | 0 | 0 | x (y) |
| KANNADA | D1 | 2839 | 1348 | 1491 | 966 | 977 | 382 (Std Kannada) | 514 (Std Kannada) | 4 (3) |
| | D2 | 4300 | 2122 | 2178 | 2122 | 2178 | 0 | 0 | 4 (2) |
| | D3 | 3692 | 1740 | 1952 | 1740 | 1952 | 0 | 0 | 4 (3) |
| | D4 | 3894 | 2002 | 1892 | 2002 | 1892 | 0 | 0 | 4 (4) |
| | D5 | 4466 | 2798 | 1668 | 2798 | 1668 | 0 | 0 | 3 (2) |
| MAGAHI | D1 | 4644 | 2326 | 2318 | 0 | 0 | 2326 (Hindi) | 2318 (Hindi) | 3 (3) |
| | D2 | 4164 | 1993 | 2171 | 1745 | 1589 | 248 (Hindi) | 582 (Hindi) | 4 (5) |
| | D3 | 4217 | 1941 | 2276 | 398 | 0 | 1543 (Hindi) | 2276 (Hindi) | 3 (5) |
| | D4 | 3914 | 1926 | 1988 | 160 | 0 | 1766 (Hindi) | 1988 (Hindi) | 6 (2) |
| MAITHILI | D1 | 4682 | 2616 | 2066 | 2019 | 1448 | 597 (Hindi) | 618 (Hindi) | 5 (5) |
| | D2 | 4596 | 2165 | 2431 | 1818 | 2431 | 347 (Hindi) | 0 | 1 (4) |
| | D3 | 5850 | 3661 | 2189 | 3661 | 2189 | 0 | 0 | 2 (3) |
| | D4 | 4633 | 2580 | 2053 | 1866 | 1149 | 714 (Hindi) | 904 (Hindi) | 6 (3) |
| MARATHI | D1 | 4929 | 1993 | 2936 | 0 | 0 | 1993 | 978 (Std Marathi) + 1958 (Hindi) | 6 (2) |
| | D2 (NASHIK) | 2591 | 1410 | 1181 | 0 | 0 | 1410 | 1181 | 4 (3) |
| | D2 (DHULE) | 2244 | 1119 | 1125 | 0 | 0 | 1119 | 1125 | 4 (3) |
| | D3 | 5010 | 2504 | 2506 | 0 | 0 | 2504 | 2506 | 3 (2) |
| | D4 | 4403 | 2219 | 2184 | 0 | 0 | 2219 | 2184 | 4 (7) |
| TELUGU | D1 | 3867 | 1788 | 2079 | 1788 | 2079 | 0 | 0 | 2 (2) |
| | D2 | 4046 | 2026 | 2020 | 2026 | 2020 | 0 | 0 | 2 (2) |
| | D3 | 3954 | 2103 | 1851 | 2103 | 1851 | 0 | 0 | 3 (3) |
| | D4 | 4071 | 2040 | 2031 | 2040 | 2031 | 0 | 0 | 8 (7) |

consistent transcription of consonants, vowels, diacritics, and prosodic markers across all Indic languages.

## C.1  1. Text Processing

1. Remove punctuation and extraneous symbols from the input text.
2. Normalize graphemes by mapping each Unicode character to a reduced "base grapheme" set aligned with ILSL's consonant and vowel categories. This normalization step removes visual variants while preserving phonetic distinctions.

## C.2  2. G2P Conversion Pipeline

1. **Basic Conversion Rules:**
   1.1. Each grapheme or grapheme pair is converted into its corresponding ILSL sound label.
   1.2. *Examples:*
   1.2.1. A consonant plus vowel, such as  + , becomes "k + aa".

Table 13: Text Corpus Summary

| LANGUAGE | DIALECT ID | # Total Samples | # Phase 1 Samples | TotalWords | Vocab Size | AvgWordLen |
|---|---|---|---|---|---|---|
| BENGALI | D1 | 4250 | 1073 | 39557 | 6493 | 9.31 (2.57) |
| | D2 | 4043 | 1135 | 38281 | 5642 | 9.47 (2.45) |
| | D3 | 4237 | 1179 | 41368 | 6259 | 9.76 (2.7) |
| | D4 | 4026 | 1035 | 38308 | 7230 | 9.52 (2.28) |
| | D5 | 5088 | 1036 | 51936 | 9474 | 10.21 (2.68) |
| BHOJPURI | D1 | 8222 | 1914 | 86327 | 7327 | 10.5 (3.21) |
| | D2 | 7786 | 2001 | 77473 | 6972 | 9.95 (3.31) |
| | D3 | 7719 | 2334 | 82709 | 7786 | 10.71 (3.45) |
| CHHATTISGARHI | D1 | 5433 | 1430 | 80522 | 6044 | 14.82 (5.35) |
| | D2 | 5251 | 1270 | 77429 | 5365 | 14.75 (5.06) |
| | D3 | 5453 | 1473 | 84760 | 5933 | 15.54 (6.71) |
| | D4 | 4901 | 1404 | 70777 | 6398 | 14.44 (5.19) |
| HINDI | D1 | 1149 | 1149 | 9857 | 1545 | 8.58 (3.35) |
| | D2 | 1101 | 1101 | 9275 | 1124 | 8.42 (2.7) |
| | D3 | 1121 | 1121 | 11295 | 1282 | 10.08 (2.93) |
| | D4 | 994 | 994 | 9947 | 1215 | 10.01 (3.02) |
| | D5 | 1109 | 1109 | 13407 | 1512 | 12.09 (4.46) |
| | STD HN | 16131 | 0 | 196828 | 16006 | 12.2 (3.15) |
| KANNADA | D1 | 3978 | 1139 | 34641 | 12788 | 8.71 (2.78) |
| | D2 | 5562 | 1262 | 51861 | 14223 | 9.32 (2.55) |
| | D3 | 4997 | 1305 | 42571 | 14874 | 8.52 (3.01) |
| | D4 | 5074 | 1180 | 50578 | 12345 | 9.97 (2.54) |
| | D5 | 5552 | 1086 | 51537 | 17316 | 9.28 (2.63) |
| MAGAHI | D1 | 6256 | 1612 | 69565 | 7657 | 11.12 (3.69) |
| | D2 | 5644 | 1480 | 66168 | 7278 | 11.72 (3.04) |
| | D3 | 5589 | 1372 | 62744 | 8067 | 11.23 (3.36) |
| | D4 | 5541 | 1627 | 50273 | 9560 | 9.07 (2.77) |
| MAITHILI | D1 | 6216 | 1534 | 68370 | 9258 | 11.0 (3.05) |
| | D2 | 6038 | 1442 | 71409 | 6970 | 11.83 (3.14) |
| | D3 | 7252 | 1402 | 72353 | 6064 | 9.98 (2.31) |
| | D4 | 6047 | 1414 | 69415 | 7614 | 11.48 (3.34) |
| MARATHI | D1 | 6390 | 1461 | 57623 | 14647 | 9.02 (2.86) |
| | D2 | 6271 | 1436 | 56452 | 14894 | 9.0 (2.7) |
| | D3 | 6442 | 1432 | 50122 | 10660 | 7.78 (2.39) |
| | D4 | 5851 | 1448 | 52573 | 12129 | 8.99 (2.62) |
| TELUGU | D1 | 5301 | 1434 | 47645 | 13364 | 8.99 (2.61) |
| | D2 | 5640 | 1594 | 48382 | 14587 | 8.58 (2.28) |
| | D3 | 5409 | 1455 | 49228 | 14588 | 9.1 (2.77) |
| | D4 | 5572 | 1501 | 44766 | 15133 | 8.03 (2.3) |

1.2.2. A consonant with a halant () drops its default vowel and joins the next consonant, e.g.,  +  → "k + sh".

2. **Special Pronunciation Patterns:**

2.1. *Schwa deletion:* Removes the unstressed " " sound where it is not pronounced, which is common in Indo-Aryan languages.

2.2. *Nasalization and gemination:* Correctly handle *anusvāra* (nasal sounds) and doubled consonants to preserve pronunciation accuracy.

## C.3   3. Rule Ordering and Language Overrides

1. General phonological rules that apply across all Indic scripts are implemented first.

2. Language-specific overrides are applied subsequently to handle exceptions such as irregular spellings or orthographic variations, ensuring the overall G2P conversion remains efficient and scalable.

This rule-based framework enables consistent phonetic lexicon generation across multiple scripts and dialects while allowing flexibility for language-specific adjustments. The resulting lexicons were validated manually for a subset of entries to ensure alignment with native pronunciation norms.

## D   Audio Recording and Quality Validation

### D.1   Participant Assignment and Recording Workflow

Voice data providers are referred to as **Voice Participants (VPs)**. Before contributing, they are screened by **Preliminary Audio Validators**, who compare recordings against text prompts to check for audio quality and reading accuracy. Participants failing to meet quality standards are removed early from the process.

The validation workflow parallels that of text validation, with a key distinction: **Language Consultants (LCs)** undergo rigorous screening, interviews, and specialized training. They are provided with custom audio validation tools and integrated into a continuous training and feedback loop before handling audio validation tasks.

**Voice Participant Criteria**

- **Geographic Authenticity**: Must reside in the district or village where the target dialect is natively spoken.
- **Literacy**: Must be able to read and speak the dialect.
- **Smartphone Proficiency**: Able to navigate the collection app and record sentences.
- **Demographic Compliance**: Must be over 18 years old and meet predefined age-gender quotas.

**Preliminary Audio Validator Criteria**   Validators assess submission quality and influence payment decisions. Each validator must possess:

- Native-level dialectal fluency
- Training in phonetic error detection
- Strong text-audio alignment skills
- Proficiency with smartphones
- High concentration and consistency
- Excellent auditory skills
- Familiarity with regional accent variations

### D.2   Validation and Payment Pipeline

**System Components**

1. WhatsApp-based participant authentication
2. Mobile application (*Bolo App*) for guided speech recordings
3. Manual and hybrid validation pipelines
4. Cloud-based backend for task management and payments

### D.2.1   Participant Authentication via WhatsApp

A dedicated WhatsApp bot, **Bolo Code Bot**, served as the authentication interface for registering and onboarding voice participants (VPs). Prior to participation, phone numbers were submitted by regional partners and stored in a secure database. Each verified participant received a unique 16-digit access code via WhatsApp, which was required to log in to

the Bolo App. Unregistered users attempting to access the app received an error message and were redirected to the respective project coordinator.

The bot supported multilingual instructions to ensure accessibility across linguistic backgrounds. Once authenticated, participants received onboarding material including installation links, demo videos, and recording guidelines. This pre-screening ensured that only vetted users satisfying dialectal and demographic criteria could contribute recordings, thereby enhancing data quality and project security.

### D.2.2 Data Collection Process

The voice collection pipeline was executed through regional partners responsible for recruiting, training, and supervising contributors. Once authenticated via WhatsApp, each participant accessed the *Bolo App* for guided recording. Submitted utterances were automatically uploaded to a centralized cloud backend, where they were processed through validation pipelines for technical quality and transcript alignment. Participant compensation was computed based on the count of validated recordings.

Each language corpus targeted approximately 1152 hours of audio distributed across five categories. These comprised 556 hours of phonetically balanced sentences (**Phase-2**), 556 hours of domain-specific content (**Phase-1**), and 10 hours each of shared agricultural and banking-domain prompts recorded by all speakers. An additional 22 hours were allocated to spontaneous prompts designed to elicit natural prosody and conversational style. Table 14 summarizes the per-language and per-dialect task allocation. For instance, languages with five dialects contributed ∼111 hours per dialect per phase, whereas those with three dialects contributed ∼185 hours.

Table 14: Dialect-wise task distribution per language.

| Task Type | Total (h) | 3 Dialects | 4 Dialects | 5 Dialects |
|---|---|---|---|---|
| Phase-2 (Phonetic) | 556 | 185.33 | 139.00 | 111.20 |
| Phase-1 (Domain) | 556 | 185.33 | 139.00 | 111.20 |
| Common Agri | 10 | 3.33 | 2.50 | 2.00 |
| Common Bank | 10 | 3.33 | 2.50 | 2.00 |
| Spontaneous | 22 | 7.33 | 5.50 | 4.40 |

Each participant was assigned 279 Phase-2 and 278 Phase-1 sentences, plus 5 common agricultural and 5 common banking prompts shared across all speakers to maximize speaker diversity. For spontaneous speech, each speaker responded to 10 open-ended prompts (e.g., "Describe a festival in your area," "What local dishes do you usually eat?"). While intended to elicit natural speech, some participants read the questions aloud; such instances were excluded from the public release.

### D.3 Bolo App Workflow

**Authentication and Profile Setup** Participants authenticated into the *Bolo App* via:

1. OTP-based mobile verification,
2. Entry of the 16-digit access code issued by the WhatsApp bot, and
3. Acceptance of the privacy policy.

They then completed a brief profile setup collecting optional photo, gender, year of birth, and pincode (for dialect verification). These steps ensured demographic balance and traceability without compromising anonymity.

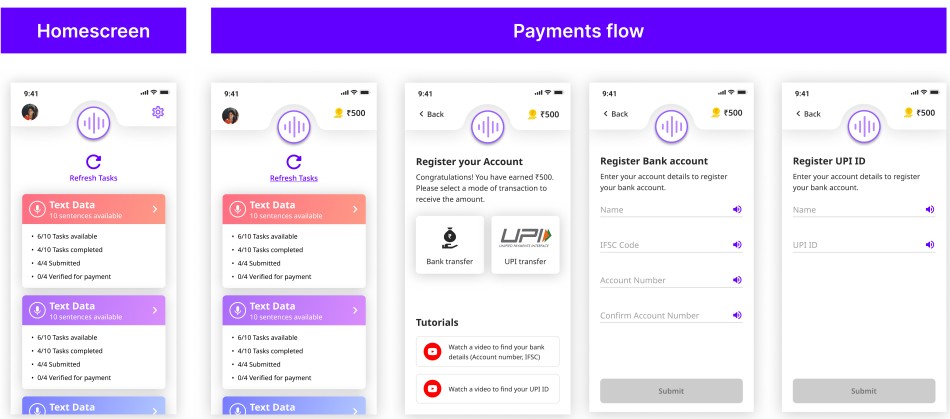

Figure 6: **(a)** NT Bolo App homescreen and task interface. The homescreen displays the number of available, completed, and verified speech tasks for each domain. Participants can refresh assigned tasks, monitor submission progress, and track validation status directly within the app. Each task card represents a text-prompt set, color-coded by completion state, enabling efficient management of recordings across multiple phases and domains.

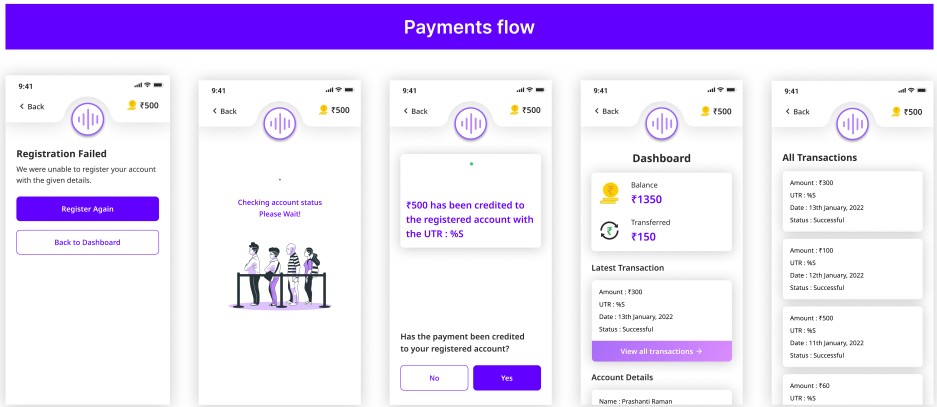

Figure 7: **(b)** Payment and verification workflow in the NT Bolo App. Screens illustrate the full post-recording process: account registration via bank or UPI, automatic verification of account validity, credit confirmation with a unique transaction reference (UTR), and display of balance and transaction history on the contributor dashboard. This integrated payment interface ensured transparent, traceable compensation and streamlined validation feedback to participants.

**Recording Interface**  The recording screen presented categorized sentence lists with waveform visualization and playback controls to ensure recording quality before submission. Tasks were color-coded by status (available, submitted, or verified), and progress summaries encouraged participant completion.

**Audio Validation and Payment**  All recordings were subjected to a hybrid validation pipeline comprising both manual and automated review stages. Manual validation was performed on a 5 % random sample of each dataset and checked for:

- Accuracy of audio–text alignment,
- Absence of excessive background noise, and
- Sufficient speech volume and clarity.

The automated stage used a neural acoustic model to compute phoneme- and word-level match scores, producing a quality score per utterance. Recordings were categorized into three slabs:

- **Clean**: ∼5 % mismatch,
- **Semi-noisy**: ∼12 % mismatch,
- **Noisy**: ≥15 % mismatch.

Languages such as Bengali and Bhojpuri exhibited slightly higher baseline mismatch rates due to orthographic complexity. Slab scoring, validation logic, and pseudocode are detailed in Appendices E.1–E.3.

Following successful validation, participant payments were automatically computed and disbursed through either direct bank transfer or UPI. Figures 6–7 summarize the end-to-end participant workflow, from recording to verified payment, implemented within the NT Bolo ecosystem.

# E  Audio Quality Categorization and Thresholding

This section describes the comprehensive process used to categorize utterances into three quality slabs: **Clean**, **Seminoisy**, and **Noisy**. The slab assignment is driven by automatic scoring models and verified through manual validation across multiple stages.

## E.1  Score Computation for Slab Assignment

To begin, consider a target dialect $Z$ and its corresponding validation model $V_{Z,O}$, where $O$ indicates the model iteration.

Let $D = [U, W, P]$ denote the dataset for dialect $Z$:

- $U = [u_1, u_2, ..., u_N]$: the set of $N$ utterances.
- $W = [W_1, ..., W_N]$: word-level sentence prompts, where each $W_i = [W_{i1}, ..., W_{iJ}]$.
- $P = [P_1, ..., P_N]$: phone-level transcriptions. Each $P_i = [P_{i1}, ..., P_{iK}]$ is derived from $W_i$ using a pronunciation lexicon.

For each utterance $u_i$:

a. Extract acoustic feature sequence $F_i = [F_{i0}, ..., F_{iT}]$.
b. Pass $(W_i, P_i, F_i)$ into $V_{Z,O}$ for alignment.
c. The model produces alignment-based scores:

$$S_{wi} = [S_{wi1}, ..., S_{wiJ}], \quad S_{pi} = [S_{pi1}, ..., S_{piK}]$$

representing word- and phone-level pronunciation quality.

d. Apply a statistical aggregation function $Y$ to get the utterance score:

$$SO_i = Y[S_{wi}, S_{pi}]$$

### E.2 Manual Validation Workflow

Manual validation ensures the reliability of score-based slab thresholds. The process is led by a multi-tiered team comprising Language Resource Managers (LRMs), Senior LRMs, and Dialect-Specific Language Consultants (LCs).

**1. Bin Selection and Task Delegation**

- The LRM selects $L$ score bins (width 0.02) and randomly chooses $M$ utterances from each bin.
- The Senior LRM reviews and approves the bin set.
- The validated bins are then assigned to LCs for manual annotation.

**2. Manual Annotation by Language Consultants**

- LCs receive the validation model ID, bin range, and $M$ utterances per bin.
- Each sample is labeled as `MATCH` or `MISMATCH`:
    - `MATCH`: Audio is correct or has minor phonetic variations.
    - `MISMATCH`: Audio is incorrect, has insertions/deletions, or meaning-altering substitutions.
- Difficult cases are resolved through discussions with LRMs or scientists.

**3. Validation Portal Process**

LCs perform validation using a structured interface:

   i. Login and model selection (VZO).
  ii. Input score interval and number of samples ($M$).
 iii. Review each utterance with playback and transcription.
 iv. Submit final MATCH/MISMATCH label.

**4. Review Mechanism**

LRMs and Senior LRMs:

- Access all LC decisions with filters.
- Overrule incorrect labels with justifications.
- Coordinate feedback loops for quality control.

**Note:**

- LCs are native dialect speakers.
- LRMs and Senior LRMs are experienced linguists or domain experts.

### E.3 Slab Generation Pseudocode

The slab generation involves multiple stages of automatic scoring and manual validation:

1. **Initialization:**
    - Dialect $Z$ has $N$ utterances scored using $V_{Z,O}$ as described earlier.
    - All $SO_i$ scores are ranked. The range covering 90%+ of data is binned into $L$ intervals of width 0.02.
2. **Voice Validation Dataset Creation:**
    - Sample $M$ utterances from each bin.
    - Manually annotate these using the LC process.

3. **Analysis-I:**
   - For each bin $l$, compute $P_l = \%$ of MATCH labels.
   - Identify thresholds:

$$SO_{\text{clean}} : P_l > 0.95, \quad SO_{\text{seminoisy}} : P_l > 0.88$$

4. **Additional Validation (Optional):**
   - Validate $K$ bins before/after identified thresholds.
   - Update scores, rerun Analysis-I.

5. **Analysis-II with Refined Model:**
   - Compute updated scores $SN_i$ for all $N$ utterances using refined model $V_{Z,N}$.
   - Repeat binning, annotation, and thresholding steps to refine:

$$SN_{\text{clean}}, \quad SN_{\text{seminoisy}}$$

6. **Final Slab Assignment:**
   - Label each utterance:

$$\text{Clean: } SN_i > SN_{\text{clean}}, \quad \text{Seminoisy: } SN_{\text{seminoisy}} < SN_i \leq SN_{\text{clean}},$$

$$\text{Noisy: } SN_i \leq SN_{\text{seminoisy}}$$

In summary, the slab generation process combines automatic scoring with expert manual validation to ensure that speech quality annotations are both reliable and reproducible. This categorization is crucial for downstream training, evaluation, and dataset release pipelines.

# F   Speaker Metadata Validation and Consistency Checks

In crowdsourced speech data collection frameworks, accurate speaker metadata is crucial but difficult to guarantee. Among various metadata types, speaker identity plays a pivotal role in training robust models and enabling tasks like speaker adaptation and verification. Errors in speaker IDs broadly fall into two categories:

**Intra-Speaker ID Errors:** A single speaker is assigned multiple distinct speaker IDs.

**Inter-Speaker ID Errors:** Multiple speakers are incorrectly assigned the same speaker ID.

Figure 8 illustrates both these error types using simplified speaker ID relationships.

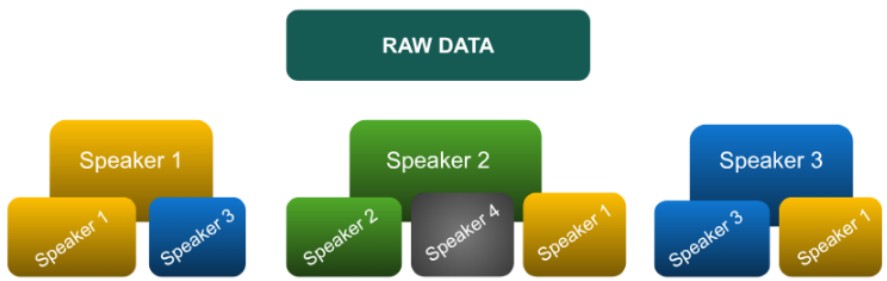

Figure 8: Illustration of intra-speaker and inter-speaker ID inconsistencies

## F.1   Speaker Meta-Data Error Prevalence and Degree

To assess the prevalence of speaker ID inconsistencies, we extracted speaker embeddings using the pretrained SpeechBrain TDNN model[8]. For intra-speaker ID analysis, cosine similarity was computed across all utterances with the same speaker label. For inter-speaker

---

[8]https://huggingface.co/speechbrain/spkrec-xvect-voxceleb

ID errors, similarity was measured between utterances from different speaker IDs within the same district.

The distributions of cosine similarities for intra- and inter-speaker pairs in Bengali are shown in Figure 9. Based on manual inspection across cosine similarity intervals, we empirically identified a threshold of 0.92. Utterance pairs with cosine similarity below 0.90 are likely to be affected by intra-speaker error, while inter-speaker error is suspected for similarity values above 0.92.

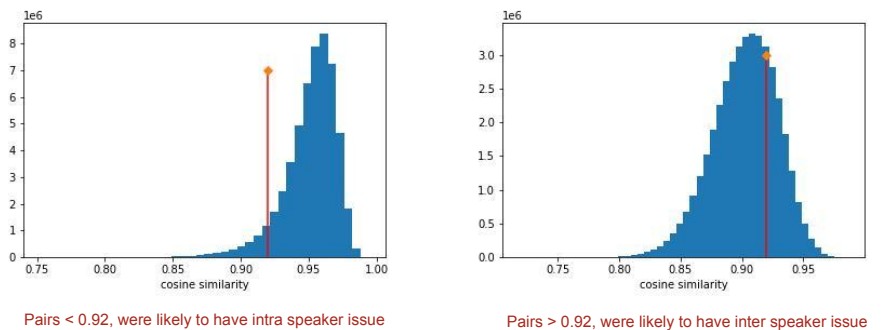

Figure 9: Cosine similarity distributions for intra- and inter-speaker embeddings in Bengali

## F.2 Speaker Meta-Information Recovery via Multi-Stage Clustering

To correct speaker labeling inconsistencies, we propose a multi-stage clustering framework consisting of calibrated threshold estimation, intra- and inter-speaker clustering, postprocessing, and validation. An overview is shown in Figure 10.

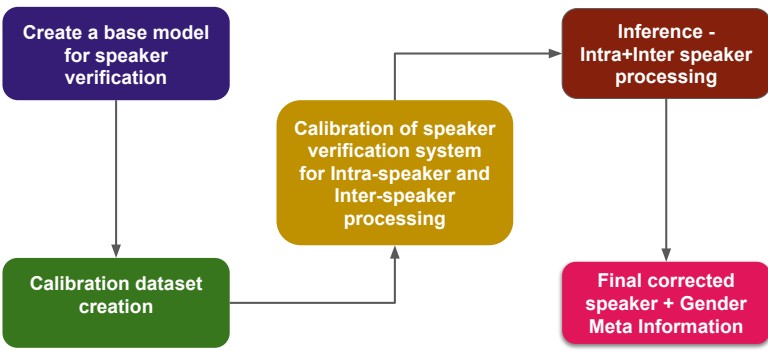

Figure 10: Overview of the speaker clustering and correction (bucketization) pipeline

### F.2.1 Dialect-Specific Calibration

We computed dialect-specific similarity thresholds using an Equal Error Rate (EER)-based calibration method. For each dialect, we curated a speaker verification set with 500 speakers (4 samples each) and 50 speakers (20 samples each), yielding 20k trials equally divided into genuine and impostor pairs. The similarity threshold at EER was adopted for clustering.

Speakers for calibration were selected to maximize demographic diversity (gender, age). Manual validation ensured reliability of these calibration samples.

### F.2.2 Intra-Speaker Clustering Methodology

Given a potentially inconsistent speaker label, we clustered the associated utterances using a similarity threshold:

1. A random utterance is chosen as the seed.

2. Utterances exceeding the threshold are clustered.

3. A centroid is computed for the cluster.

4. The cluster is refined using the centroid similarity.

5. Clustered utterances are removed; process repeats.

To address over-fragmentation, post-clustering merges were performed if cross-cluster centroid and utterance similarities both exceeded the threshold.

A confidence scoring mechanism was implemented using centroid similarity, intra-cluster consistency, and post-validation cross-similarity scores. Low-confidence utterances were filtered to improve cluster purity.

Gender metadata was also integrated to flag inconsistencies. Clusters with mixed-gender predictions were reviewed or split.

### F.2.3 Inter-Speaker Clustering Framework

To merge clusters representing the same speaker across speaker labels:

1. Compute centroid similarities between different speaker clusters.

2. If similar, sample utterances from both clusters and compute pairwise similarities.

3. Merge clusters only if both conditions are satisfied.

A hierarchical verification protocol was adopted to ensure reliability:

1. Initial inter-speaker cluster proposals

2. Re-run intra-clustering on merged sets

3. Filter by confidence score

4. Manual review of borderline cases

### F.3 Sampling Uncontaminated Speakers for Dev and Test Sets

Following speaker clustering and verification, we compiled a list of *uncontaminated speakers—* those not involved in any intra- or inter-speaker ID errors. This uncontaminated speaker pool formed the basis for dev and test set sampling, ensuring no overlap with the training data. This separation is critical to maintain dataset partition integrity and enables reliable benchmarking.

### F.4 Testing the Bucketization Algorithm

To evaluate the effectiveness of our clustering-based bucketization, we designed a synthetic blind test corpus. This test set was constructed by combining samples from standard and internal Indian speech corpora with ground-truth speaker labels. New synthetic speaker labels were introduced to simulate known intra- and inter-speaker ID errors.

Figure 11 outlines the creation of this test corpus.

Figure 12 shows the application of the bucketization algorithm to recover correct speaker IDs.

**Intra-Speaker Evaluation:** Alignment score is used to measure the purity of predicted speaker clusters.

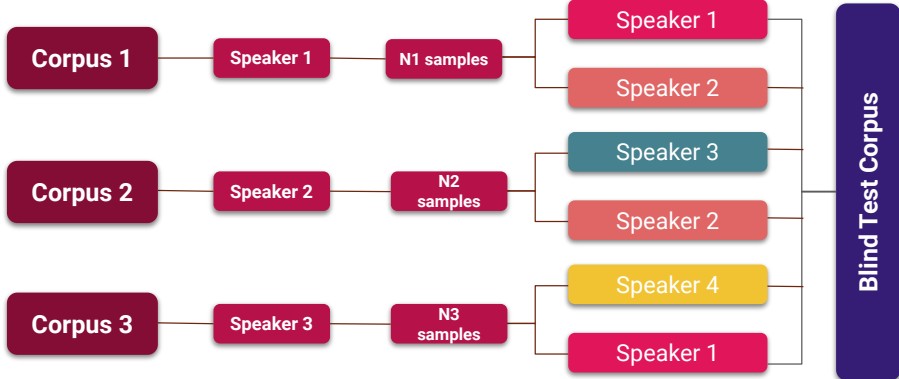

Figure 11: Creating the blind test corpus by simulating intra and inter speaker label errors

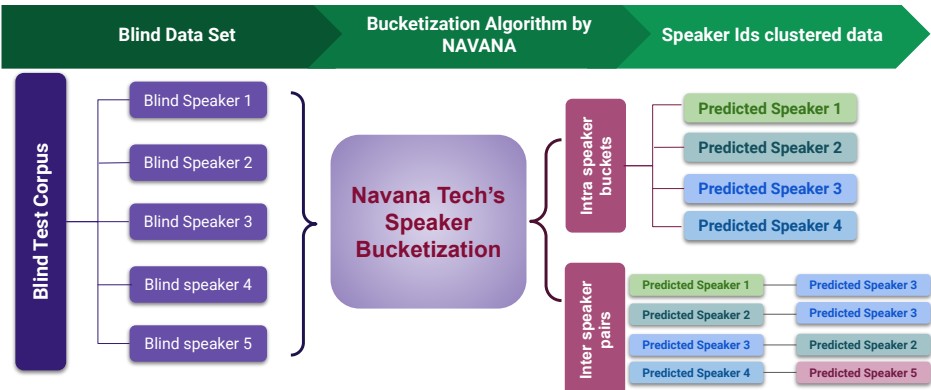

Figure 12: Inference of bucketization algorithm on blind test set

---

**Algorithm 1** Calculating Alignment Score

---

**for** *each bucket in Blind Test Corpus* **do**

    Create a confusion matrix (cm) between predicted and ground-truth speakers  Initialize alignment score to 0

    **while** *cm is not empty* **do**

        Find the maximum value in cm

        Delete the row and column corresponding to the maximum value

        Increment alignment score by the maximum value

    Normalize the alignment score

---

**Intra-Speaker Evaluation:** To evaluate the ability of the bucketization algorithm to resolve intra-speaker ID errors, we compute an *alignment score* for each speaker bucket. This metric quantifies how well the predicted speaker clusters align with the ground-truth speaker labels.

Figure 13 illustrates the evaluation workflow for intra-speaker analysis. The alignment score is computed using the confusion matrix between predicted and actual speaker labels, as formalized in Algorithm 1. Higher alignment scores indicate better cluster purity and minimal fragmentation.

Figure 15 (left) shows a sample confusion matrix for one speaker cluster, where the maximum alignment path is highlighted. The algorithm achieved an average alignment score of **99.28%**, confirming that intra-speaker identity inconsistencies were almost completely resolved.

**Inter-Speaker Evaluation:** For inter-speaker errors, where multiple labels are used for the same speaker, the algorithm proposes pairs of speaker clusters to merge. We evaluate this using *Intersection over Union (IoU)* between the sets of ground-truth speaker IDs represented in each proposed cluster pair.

The evaluation process is illustrated in Figure 14, while Figure 15 (right) depicts how IoU is calculated between the overlapping speaker identity sets. The algorithm achieved an average inter-speaker IoU of **52.91%**, suggesting moderate success in merging duplicated speaker identities. The lower IoU relative to the alignment score indicates that resolving inter-speaker ID inconsistencies remains more challenging.

**Final Evaluation Summary:** Overall performance is summarized in Figure 16, which visualizes intra and inter evaluation metrics across the test corpus. Although intra-speaker resolution was highly accurate, approximately **10% of utterances** in the blind test corpus remained unassignable due to confidence score thresholds or conflicting metadata.

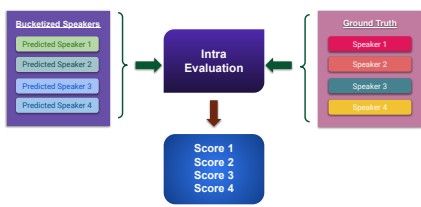

Figure 13: Blind Test Intra-Speaker Evaluation

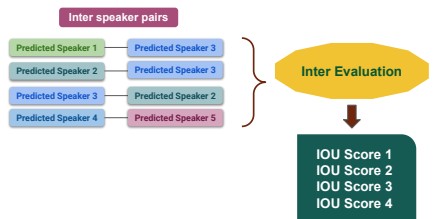

Figure 14: Blind Test Inter-Speaker Evaluation

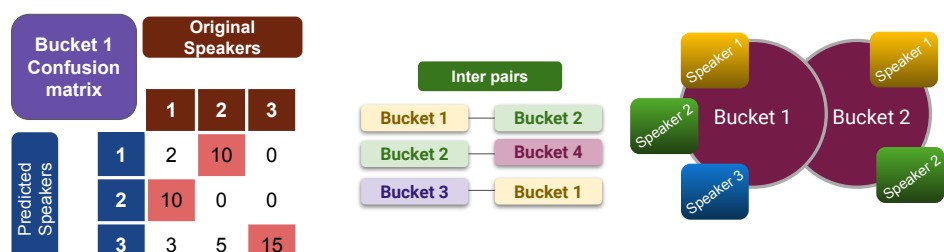

Figure 15: Evaluation metrics: (left) Confusion matrix alignment for intra-speaker analysis; (right) IoU for inter-speaker cluster overlap

# G   Train–Dev–Test Partitioning Strategy

## G.1   Dev and Test Sets Creation

Creating the dev and test sets from the dataset $D_Z$ of Dialect Z involves careful selection of speakers and utterances, ensuring diversity and avoiding speaker contamination between sets. The dataset $D_Z$ consists of audio files categorized into three slabs: $A_c$ (Clean), $A_{sn}$ (Semi-noisy), and $A_n$ (Noisy), as described in Appendix E. Each utterance belongs to one of these slabs, and no speaker or text ID is repeated across dev, test, and train splits.

Since the dev and test sets are relatively small compared to the training set, and need to be balanced across dialects, domains, and genders, we prioritize their creation. Balance is maintained wherever possible in the number of utterances across domains (Agriculture and Banking), task types (Question and Statement), and speaker gender.

The pseudocode below outlines the initialization and constraints applied to construct these sets:

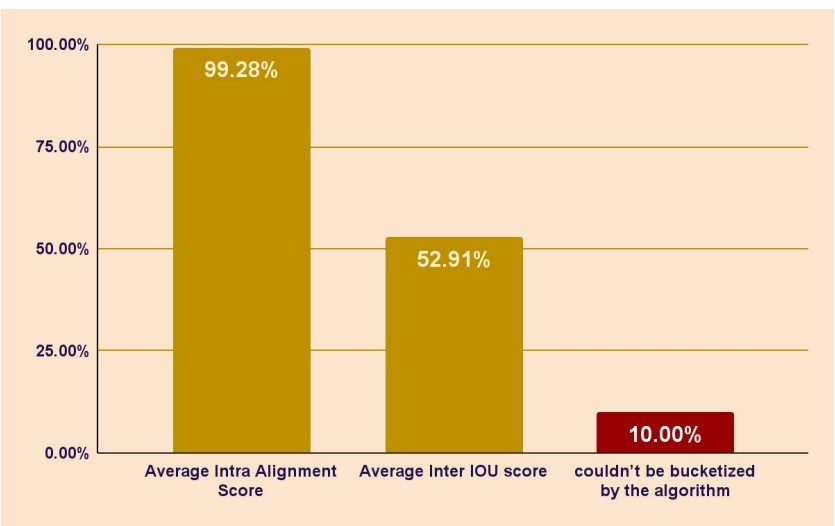

Figure 16: Bucketization performance results across the test corpus

---

**Algorithm 2** Test Set and Dev Set Creation

---

1: **Initialize:**
2: Define $A$ as the total number of audio files in $D_Z$.
3: Define subsets $A_c$, $A_{sn}$, and $A_n$ corresponding to the three slabs (see Appendix E for slab definitions).
4: Define $S$ as the total number of speakers, and $S_{uncont}$ as the set of uncontaminated speaker IDs (see Appendix F for contamination criteria).
5: Let max_rep represent the maximum allowed repetitions of any text ID across the test and dev sets.
6: **Diversity Constraints:**
7: Maintain balance across task (Question, Statement), domain (Banking, Agriculture), and gender (male, female).
8: Ensure minimal loss of utterances during test and dev set creation.

---

### G.1.1 Test Set Creation

The test set is constructed by filtering the list of uncontaminated speakers and selecting utterances from the clean slab $A_c$. Gender balance is enforced by sampling from both male and female speaker pools, denoted $S_{ms}$ and $S_{fs}$. For each selected speaker, exactly one utterance is chosen for each of the four combinations of task and domain.

To avoid repeated evaluation content, utterances with overused text IDs are discarded after reaching the `max_rep` limit. A final diversity analysis ensures the test set $A_{test}$ is balanced. The process is summarised below:

**Algorithm 3 Test Set Creation:**

1: Filter speakers based on $S_{uncont}$.
2: Select $A_c$ from $S_{uncont}$ for test set creation.
3: Form subsets $S_{ms}$ and $S_{fs}$, ensuring balance in tasks and domains.
4: **for** each selected speaker per gender **do**
5:    Select a single $A_c$ utterance for each of the following:
6:       Task: Question, Domain: Agriculture
7:       Task: Statement, Domain: Agriculture
8:       Task: Question, Domain: Banking
9:       Task: Statement, Domain: Banking
10:    Delete all utterances with text IDs selected `max_rep` times.
11: **end for**
12: Perform diversity analysis. If constraints are met, freeze the test set $A_{test}$.

### G.1.2  Dev Set Creation

The development set is generated using a similar strategy to the test set, with the added constraint that neither speakers nor text IDs from the test set are reused. As before, gender and task-domain balance is ensured through careful sampling. The finalized dev set $A_{dev}$ is frozen once diversity constraints are satisfied.

**Algorithm 4 Dev Set Creation:**

1: Filter speakers based on $S_{uncont}$, excluding those from $A_{test}$.
2: Select $A_{slab1}$ from $S_{uncont}$ for dev set creation.
3: Form subsets $S_{ms}$ and $S_{fs}$, ensuring balance in tasks and domains.
4: **for** each selected speaker per gender **do**
5:    Select a single $A_c$ utterance for each of the following:
6:       Task: Question, Domain: Agriculture
7:       Task: Statement, Domain: Agriculture
8:       Task: Question, Domain: Banking
9:       Task: Statement, Domain: Banking
10:    Delete all utterances with text IDs selected `max_rep` times.
11: **end for**
12: Perform diversity analysis. If constraints are met, freeze the dev set $A_{dev}$.

### G.2  Training Set Creation

The training set $A_{train}$ is created by filtering out any utterances associated with speakers or text IDs from the test and dev sets. Diversity analysis is then performed to ensure the distribution of sentences, domains, and gender is consistent with the full dataset $D_Z$. The final training set is frozen after confirming that diversity constraints are met.

1: **Training Set Creation:**
2: Filter out utterances belonging to any speaker or text ID from $A_{test}$ and $A_{dev}$.
3: Perform diversity analysis to verify that the training set $A_{train}$ reflects the diversity of $D_Z$.
4: Freeze the training set $A_{train}$.

### G.3  Handling High Loss Dialects

Some dialects, such as Mt_D2, Hi_D2, and Hi_D4, suffer from low availability of clean and semi-noisy samples or fail to meet the diversity constraints due to skewed distributions. For such cases, we expand speaker selection to include both slabs $A_{slab1}$ and $A_{slab0.5}$. This ensures sufficient representation while preserving uncontaminated speaker quality.

---

1: **Sampling for High Loss Dialects:**
2: Follow the same steps as for the test and dev sets, but include both $A_{slab1}$ and $A_{slab0.5}$ for speaker selection.

---

## G.4 Balanced Sampling for Small Train Subsets

To support broader usage and evaluation of the RESPIN corpus, we release a compact version of the training set for each of the nine languages, named `train_<lang>_small`. This subset is constructed using a principled balanced sampling strategy that ensures fair representation across dialects and domains while maintaining sufficient speaker and utterance diversity.

The first step in this process is to perform balanced sampling of text IDs from the available training data. The goal is to distribute the sampling quota equally across all dialects ($\mathcal{D}_{\mathrm{did}}$), and then within each dialect, further divide it among its constituent domains ($\mathcal{D}_{\mathrm{dom}}$). To ensure meaningful inclusion, only those text IDs with at least five utterances are considered valid for sampling. If a particular domain lacks enough valid text IDs to meet its quota, the deficit is compensated by drawing additional IDs from other domains within the same dialect. This ensures the target size is met without sacrificing dialectal balance.

The detailed logic is formalized in Algorithm 5, which outputs a set of sampled text IDs, $\mathcal{T}_{\mathrm{sampled}}$, balanced across dialect-domain pairs. These sampled IDs serve as the foundation for constructing the final training subset.

Once the text IDs are sampled, the next step is to select corresponding utterances. For each sampled text ID, we aim to include five utterances from distinct speakers. To avoid speaker imbalance, speakers with the lowest frequency of participation in the training set are prioritized.

Algorithm 6 outlines the procedure to filter utterances, compute speaker frequencies, and perform speaker-aware sampling. This guarantees both text diversity and speaker variation in the final subset.

---

**Algorithm 5** Balanced Sampling of Text IDs Across Dialects and Domains

**Input:**
- $\mathcal{T}$: Set of all text IDs
- $\mathcal{D}_{\mathrm{did}}$: Set of dialect IDs
- $\mathcal{D}_{\mathrm{dom}}$: Set of domain IDs
- $N_{\mathrm{total}}$: Total number of text IDs to sample

**Output:** $\mathcal{T}_{\mathrm{sampled}}$: Balanced sampled text IDs

1: **Filter Valid Text IDs:**
$$\mathcal{T}_{\mathrm{valid}} = \{t \in \mathcal{T} \mid |\mathrm{utterances}(t)| \geq 5\}$$

2: **Set Sampling Targets:**
$$N_{\mathrm{dialect}} = \frac{N_{\mathrm{total}}}{|\mathcal{D}_{\mathrm{did}}|} \qquad N_{\mathrm{target}}(d, o) = \frac{N_{\mathrm{dialect}}}{|\{o \in \mathcal{D}_{\mathrm{dom}} \mid d \text{ fixed}\}|}$$

3: **Sample Text IDs:** For each $(d, o) \in \mathcal{D}_{\mathrm{did}} \times \mathcal{D}_{\mathrm{dom}}$:

$$\mathcal{T}_{\mathrm{sampled}}(d, o) = \begin{cases} \text{Compensate from other domains,} & \text{if } |\mathrm{available}(d, o)| < N_{\mathrm{target}}(d, o) \\ \text{Sample within domain,} & \text{otherwise} \end{cases}$$

Append $\mathcal{T}_{\mathrm{sampled}}(d, o)$ to $\mathcal{T}_{\mathrm{sampled}}$

4: **Save Results:**
$$\text{Save } \mathcal{T}_{\mathrm{sampled}} \text{ to output files}$$

---

**Algorithm 6** Sample 5 Utterances for Each Text ID

**Input:**

- $\mathcal{T}_{\text{sampled}}$: Set of sampled text IDs from the previous algorithm
- $\mathcal{U}$: Set of all utterances with associated text and speaker IDs
- $\mathcal{U}_{\text{train}}$: Set of utterances in the training set

**Output:**

- $\mathcal{S}_{\text{sampled}}$: Table of speaker IDs sampled for each text ID
- $\mathcal{U}_{\text{sampled}}$: Table of utterance IDs sampled for each text ID

1: **Filter Utterances:**

$$\mathcal{U}_{\text{filtered}} = \{u \in \mathcal{U} \mid \text{tid}(u) \in \mathcal{T}_{\text{sampled}} \text{ and } u \in \mathcal{U}_{\text{train}}\}$$

2: **Compute Speaker Frequencies:**

$$f_s(s) = \text{Count of occurrences of each speaker ID } s \text{ in } \mathcal{U}_{\text{filtered}}$$

3: **Sample Speaker IDs:** For each $t \in \mathcal{T}_{\text{sampled}}$:

$$\mathcal{S}_t = \text{Select 5 speakers with the lowest } f_s(s) \text{ for } t$$

4: **Sample Utterances:** For each $t \in \mathcal{T}_{\text{sampled}}$:

$$\mathcal{U}_t = \text{Select utterances associated with } \mathcal{S}_t$$

5: **Save Results:**

$$\mathcal{S}_{\text{sampled}} = \{(t, s_1, s_2, \dots, s_5) \mid t \in \mathcal{T}_{\text{sampled}}, s_i \in \mathcal{S}_t\}$$
$$\mathcal{U}_{\text{sampled}} = \{(t, u_1, u_2, \dots, u_5) \mid t \in \mathcal{T}_{\text{sampled}}, u_i \in \mathcal{U}_t\}$$

### G.5 Training Set Statistics and Insights

Table 15: Language-wise statistics for Clean, Semi-noisy, and Noisy train sets including duration, utterances, unique sentences, and speaker counts.

| LID | #Dialects | Clean Train Set | | | | Semi-noisy Train Set | | | | Noisy Train Set | | | |
|---|---|---|---|---|---|---|---|---|---|---|---|---|---|
| | | Dur (h) | #Utts | #Sents | #Spks | Dur (h) | #Utts | #Sents | #Spks | Dur (h) | #Utts | #Sents | #Spks |
| bh | 3 | 889.61 | 694738 | 21967 | 1851 | 91.83 | 58179 | 16493 | 1505 | 68.53 | 43543 | 15256 | 1392 |
| bn | 5 | 894.60 | 645791 | 20187 | 1791 | 162.84 | 92208 | 17766 | 1479 | 5.77 | 6587 | 5393 | 805 |
| ch | 4 | 1009.09 | 640649 | 19400 | 2082 | 161.28 | 78942 | 17334 | 2048 | 73.96 | 36486 | 14456 | 2024 |
| hi | 5 | 712.61 | 574544 | 19345 | 2305 | 273.27 | 181710 | 19180 | 2304 | 304.55 | 186275 | 19286 | 2314 |
| kn | 5 | 888.72 | 554398 | 23294 | 1931 | 202.42 | 104324 | 22004 | 1919 | 191.18 | 97806 | 21918 | 1935 |
| mg | 4 | 1066.30 | 769679 | 21500 | 2078 | 91.30 | 53414 | 16361 | 2037 | 60.18 | 36206 | 14943 | 2025 |
| mr | 4 | 1026.06 | 809934 | 23069 | 2644 | 285.46 | 191098 | 21669 | 2629 | 174.16 | 109910 | 20306 | 2634 |
| mt | 4 | 552.16 | 392739 | 23108 | 1917 | 400.04 | 237484 | 23154 | 1916 | 314.21 | 174313 | 22609 | 1934 |
| te | 4 | 1021.16 | 702883 | 19978 | 2287 | 186.10 | 111471 | 19370 | 2273 | 127.66 | 73390 | 17914 | 2258 |
| **Total** | **38** | **8060.31** | **5785355** | **191848** | **18886** | **1854.54** | **1108830** | **173331** | **18110** | **1320.20** | **764516** | **152081** | **17325** |

**LID**: Language ID, **#Dialects**: number of dialects, **Dur**: duration in hours, **#Utts**: number of utterances, **#Sents**: number of unique sentences, **#Spks**: number of speakers.

Table 15 provides a comprehensive summary of language-wise training set statistics across the Clean, Semi-noisy, and Noisy slabs in the RESPIN corpus. For each language, the table presents the number of dialects, total duration in hours, number of utterances, number of unique sentences, and number of speakers for each slab.

The Clean Train Set comprises high-quality, curated audio data and represents the largest portion of the training set, totaling 8060.31 hours and over 5.78 million utterances. Lan-

guages like Marathi (mr), Magahi (mg), and Chhattisgarhi (ch) contribute significantly to the clean set, with over 1000 hours each. The number of speakers per language ranges from approximately 1800 to over 2600, ensuring high speaker diversity.

The Semi-noisy Train Set introduces moderate background noise and variability, offering a middle ground between clean and highly degraded conditions. It adds approximately 1854.54 hours and 1.1 million utterances to the training data. Hindi (hi), Maithili (mt), and Marathi (mr) are the top contributors in terms of duration. Speaker coverage remains uniformly distributed, preserving balance across languages.

The Noisy Train Set is characterized by challenging acoustic conditions and contains 1320.20 hours and 764,516 utterances. Despite filtering for quality, a sizable portion of the data remains usable. Notably, languages such as Hindi and Maithili still provide substantial noisy data, while Bengali (bn) contributes significantly less due to stringent filtering, offering only 5.77 hours.

Across all three slabs combined, the RESPIN training corpus comprises approximately 11,235 hours of audio spanning over 7.96 million utterances, 517,260 unique sentences, and 54,321 speakers (non-unique across slabs). This extensive and diverse training set enables the development of robust speech models capable of generalizing across noise conditions, dialectal variations, and speaker demographics.

## H    Audio Data Analysis

The distributions of SNR values, durations and speaking rates for RESPIN audio data are specified in sections H.1 and H.2, respectively

### H.1    SNR-Based Audio Quality Analysis

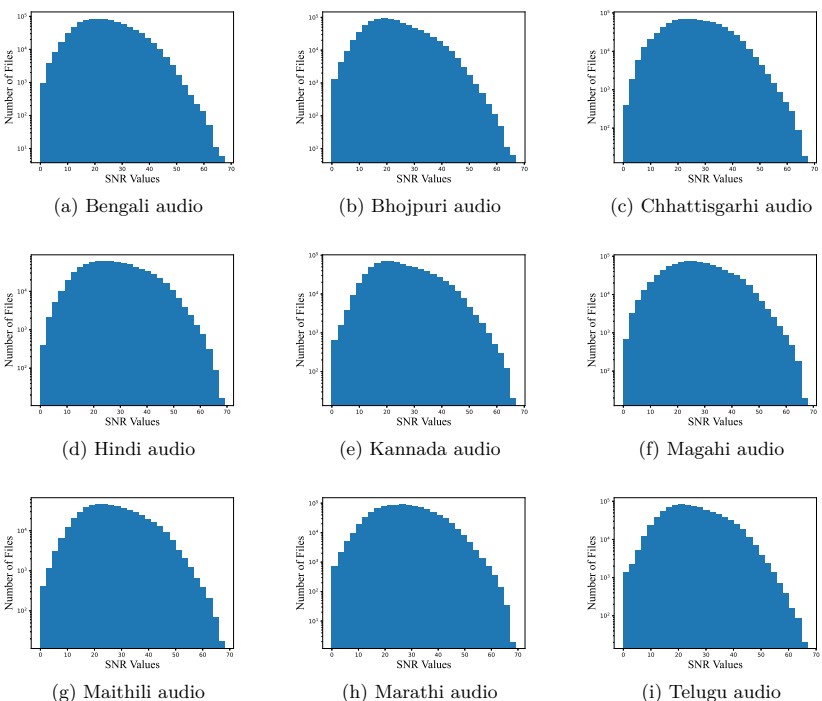

Figure 17: Histograms of SNR values for slab *Clean*

We use Signal-to-Noise Ratio (SNR) as a proxy to quantify audio quality across the RESPIN dataset. SNR measures the ratio of speech signal energy to background noise energy, ex-

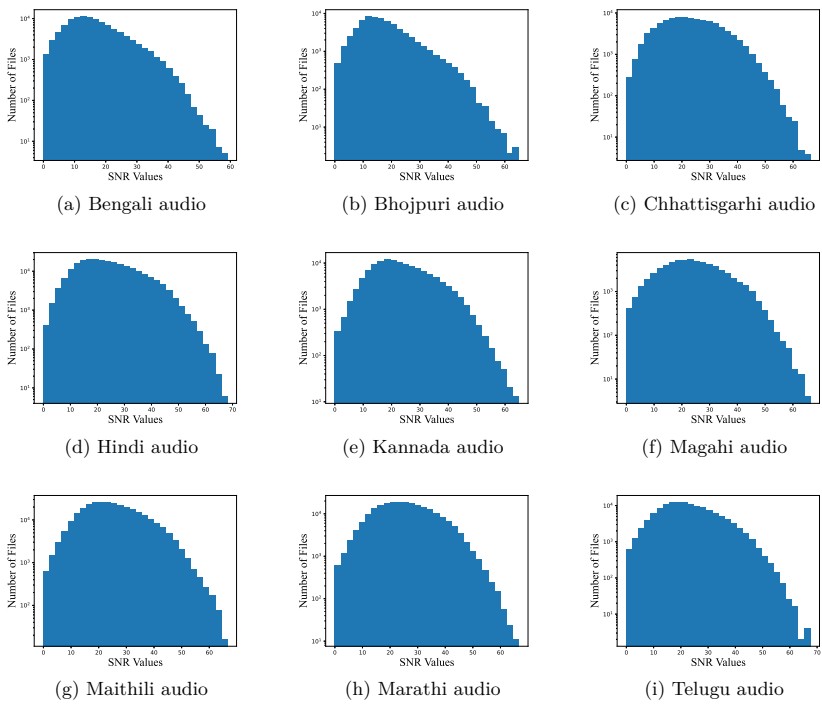

Figure 18: Histograms of SNR values for slab *Semi-noisy*

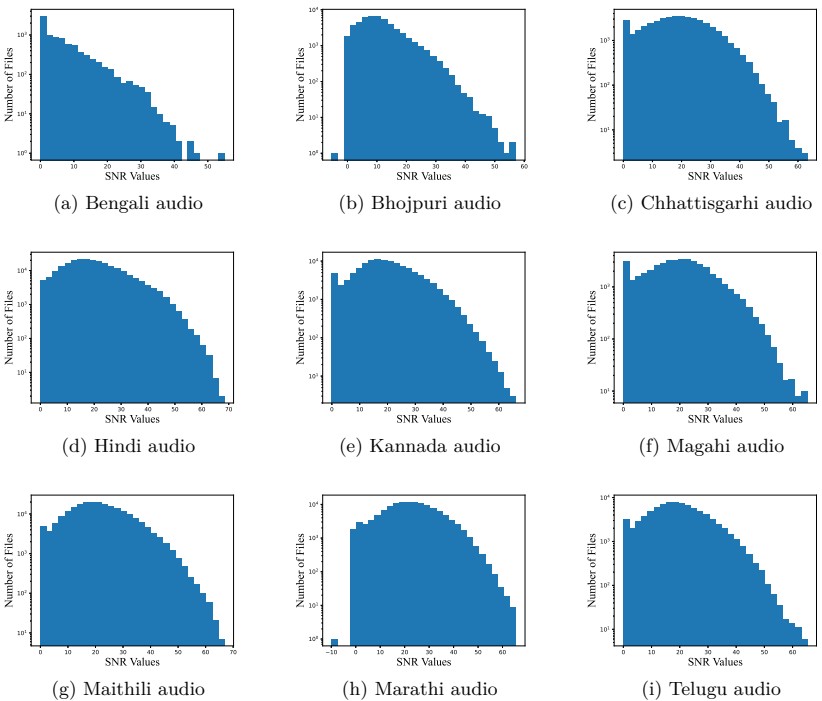

Figure 19: Histograms of SNR values for slab *Noisy*

Table 16: Language-wise audio statistics in the Semi-noisy slab of RESPIN. The table shows the number of files, count and percentage of utterances with SNR < 4 dB, and speaking rate features.

| LID | bn | bh | ch | hi | kn | mg | mt | mr | te |
|---|---|---|---|---|---|---|---|---|---|
| **# Files** | 107,920 | 66,966 | 98,095 | 238,370 | 129,533 | 65,546 | 304,410 | 223,938 | 133,286 |
| **# Low SNR** | 4,358 | 1,607 | 844 | 1,453 | 877 | 989 | 1,706 | 1,445 | 1,509 |
| **% Low SNR** | 4.04 | 2.40 | 0.86 | 0.61 | 0.68 | 1.51 | 0.56 | 0.65 | 1.13 |
| **Wds/Aud** | 9.28 | 10.11 | 14.91 | 11.59 | 9.12 | 10.72 | 11.19 | 9.04 | 9.02 |
| **Dur (s)** | 5.38 | 4.91 | 6.45 | 4.75 | 5.95 | 5.30 | 5.22 | 4.64 | 5.15 |
| **WPM** | 118.90 | 134.19 | 149.81 | 157.58 | 100.05 | 136.48 | 137.63 | 124.74 | 112.55 |

**Abbreviations:** LID = Language ID; # Files = Number of audio files; # Low SNR = Number of utterances with SNR < 4 dB; % Low SNR = Proportion of low-SNR utterances; Wds/Aud = Average words per utterance; Dur (s) = Average utterance duration in seconds; WPM = Words per minute.

Table 17: Language-wise audio statistics in the Noisy slab of RESPIN, showing number of files, count and percentage of utterances with SNR < 4 dB.

| LID | bn | bh | ch | hi | kn | mg | mt | mr | te |
|---|---|---|---|---|---|---|---|---|---|
| **# Files** | 10,413 | 50,799 | 47,657 | 238,990 | 118,999 | 46,119 | 227,340 | 128,562 | 88,378 |
| **# Low SNR** | 4,170 | 7,436 | 4,102 | 10,130 | 6,774 | 4,198 | 7,620 | 6,077 | 4,859 |
| **% Low SNR** | 40.05 | 14.64 | 8.61 | 4.24 | 5.69 | 9.10 | 3.35 | 4.73 | 5.50 |

**Abbreviations:** LID = Language ID; # Files = Number of audio files; # Low SNR = Number of utterances with SNR < 4 dB; % Low SNR = Proportion of low-SNR utterances.

pressed in decibels (dB). A higher SNR indicates a cleaner, less noisy recording, making it a useful metric for characterizing the quality of speech data.

**SNR Computation Methodology.** As shown in Figure 20, we compute the SNR of each utterance using the pretrained `FB-Denoiser`[9] model, which is based on the DEMUCS architecture for speech enhancement. Given an original audio sample $x_m$, we generate its denoised counterpart $y_m$, and compute the estimated noise as $n_m = x_m - y_m$. To ensure that the SNR reflects only the spoken content, non-speech segments at the start and end of the audio are trimmed using forced alignment timestamps generated by a Kaldi TDNN-HMM model trained on slab Clean.

The SNR value is calculated using the standard formula:

$$\text{SNR} = 10 \log_{10}\left(\frac{X}{N}\right) \tag{1}$$

where $X$ is the average signal power, and $N$ is the average noise power. These are computed as:

$$X = \frac{1}{m}\sum_{i=1}^{m}(x_i)^2 \tag{2}$$

$$N = \frac{1}{m}\sum_{i=1}^{m}(n_i)^2 \tag{3}$$

---

[9]`https://github.com/facebookresearch/denoiser`

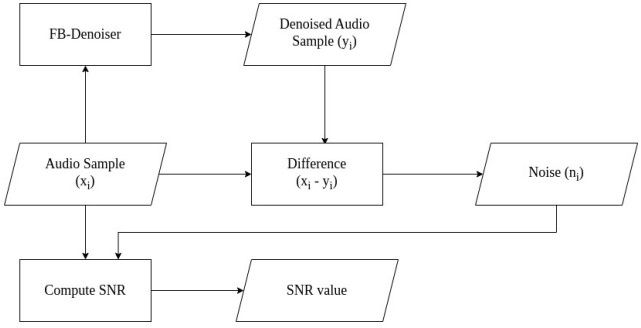

Figure 20: Calculation of SNR values using FB-Denoiser

**SNR Distributions Across Slabs:** Figures 17, 18, and 19 visualize the distributions of SNR values for all nine languages across the Clean, Semi-noisy, and Noisy slabs, respectively. These histograms clearly show how signal quality varies across slabs.

In the Clean slab (Figure 17), SNR values are tightly concentrated between 25 and 40 dB, with minimal low-SNR outliers. Languages like Hindi, Kannada, and Maithili demonstrate particularly clean profiles with sharp histogram peaks and minimal long tails.

In the Semi-noisy slab (Figure 18), the SNR distributions are noticeably wider, and the peaks shift leftward. Table 16 reports that the percentage of utterances with SNR below 4 dB is still relatively low—ranging from 0.56% for Maithili to 4.04% for Bengali. Hindi and Maithili continue to exhibit stable audio quality, while Bengali, Bhojpuri, and Magahi show modest increases in low-SNR samples.

In the Noisy slab (Figure 19), the degradation becomes more evident. Bengali has the most substantial drop in quality, with 40.05% of its utterances falling below 4 dB, as shown in Table 17. Bhojpuri, Chhattisgarhi, and Magahi also show elevated noise levels. In contrast, Hindi and Maithili retain a relatively small proportion of low-SNR files even in the Noisy slab, suggesting better recording conditions or noise resilience during collection.

SNR variation is not only slab-dependent but also language-specific. Bengali consistently exhibits more low-SNR recordings, while Maithili and Hindi maintain relatively clean audio across all slabs. This difference may stem from differences in recording environments, devices used, or speaker demographics. Additionally, the long-tailed nature of the Noisy distributions reflects the presence of a small but significant number of extremely noisy files that may require additional filtering in downstream tasks.

**Summary** These SNR-based analyses validate the design of the slab-based data curation pipeline. The progressive leftward shift of SNR distributions from Clean to Semi-noisy to Noisy slabs confirms the intended stratification of audio quality. This tiered structure makes RESPIN suitable for benchmarking ASR models under varying levels of noise and enables controlled experimentation for noise-robust speech recognition in diverse Indian language dialects.

## H.2 Speech Duration and Speaking Rate Analysis

We analyze two important prosodic features of the RESPIN audio corpus—utterance duration and speaking rate (in words per minute, WPM)—across the Clean and Semi-noisy slabs. These metrics help assess consistency in recording conditions and variability in natural speech across languages.

**Utterance Duration:** Utterance duration is computed from forced alignment outputs obtained using a Kaldi TDNN-HMM model trained on Clean audio. We define the duration as the time interval between the start of the first word and the end of the last word, ignoring leading/trailing silences. Figures 23 and 24 show the distribution of utterance durations for each language in the Clean and Semi-noisy slabs, respectively.

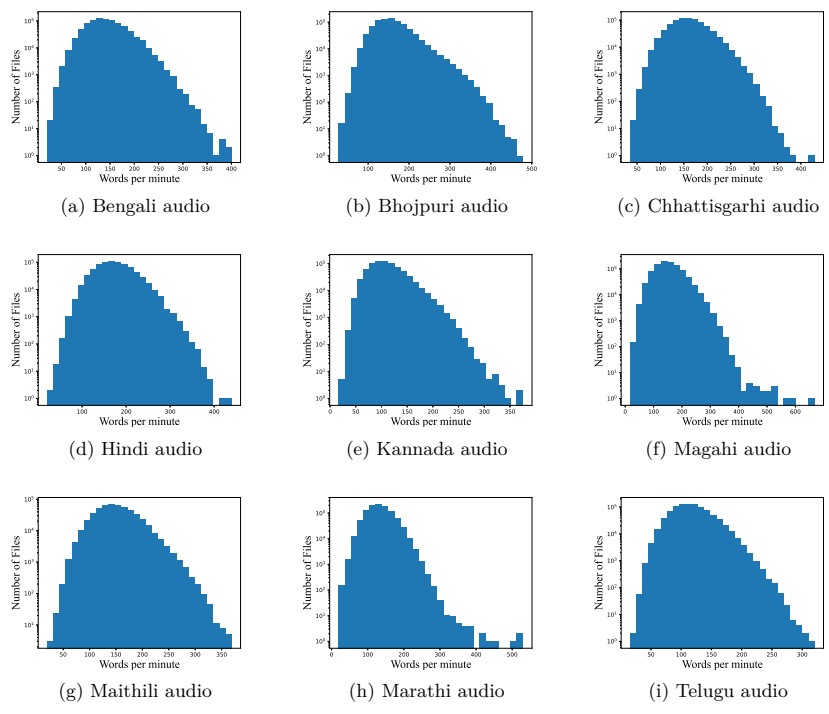

Figure 21: Histograms of WPM values for slab Clean

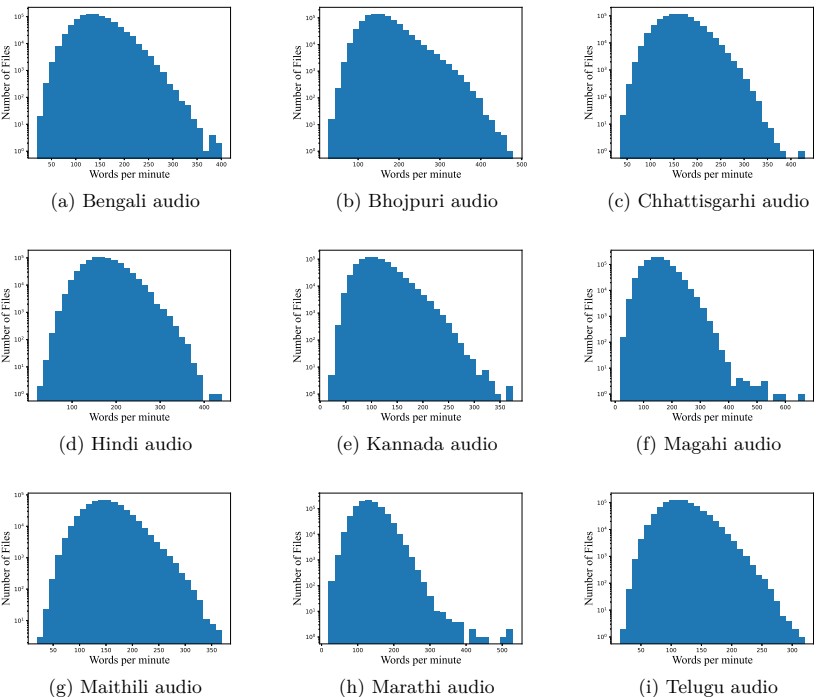

Figure 22: Histograms of WPM values for slab Semi-noisy

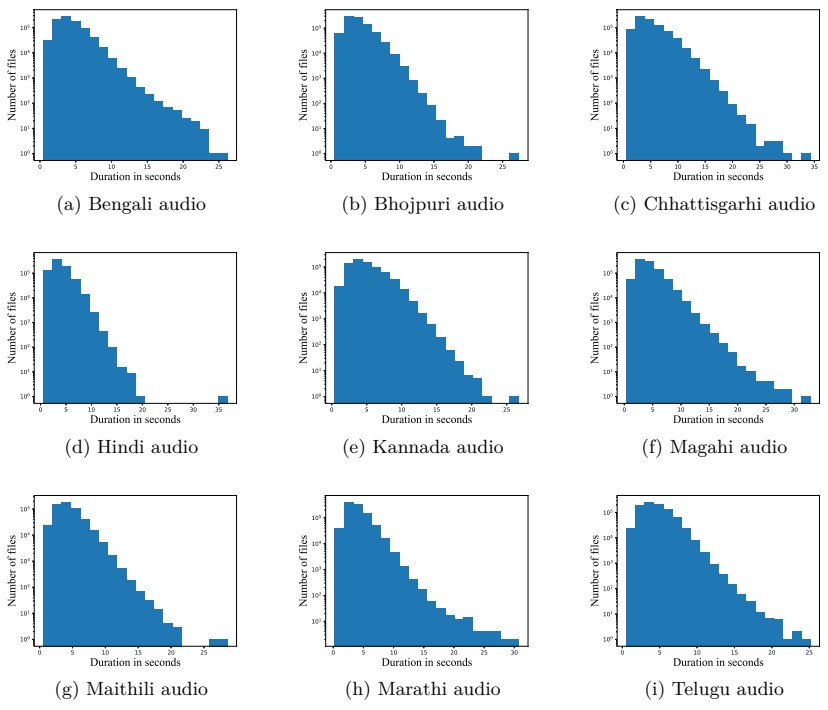

Figure 23: Histograms of audio durations for slab Clean

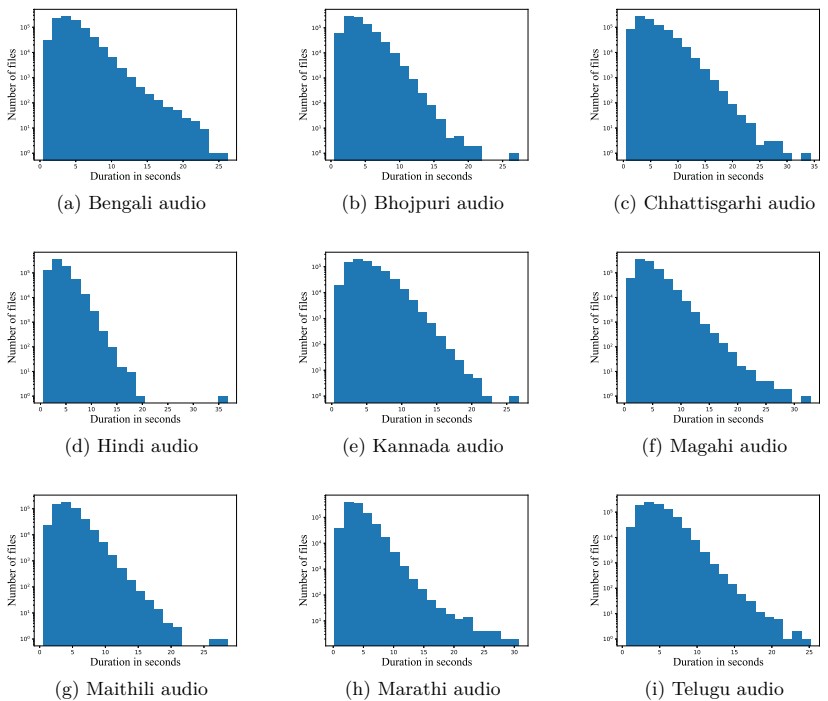

Figure 24: Histograms of audio durations for slab Semi-noisy

In the Clean slab, most languages exhibit tightly peaked distributions around 4–6 seconds with minimal long tails, indicating consistent control during scripted recording. In contrast, the Semi-noisy slab shows greater variability and longer tails, especially for Kannada and Maithili, which display more spread-out patterns. These differences reflect the less constrained and more spontaneous nature of recordings in Semi-noisy conditions.

**Speaking Rate:** Speaking rate is defined as the number of words spoken per minute in each utterance. We compute this by dividing the number of tokenized words by the utterance duration (in seconds) and scaling appropriately. Figures 21 and 22 present the WPM histograms for Clean and Semi-noisy slabs.

In the Clean slab, the distributions are largely Gaussian-shaped and fall between 110–150 WPM. Languages such as Hindi, Maithili, and Chhattisgarhi exhibit higher average rates (above 140 WPM), while Kannada and Bengali cluster toward the lower end. The Semi-noisy slab, however, reveals increased spread and irregularities in speaking rate. Notably, Bengali and Kannada show low WPM outliers (<50 WPM), whereas Hindi retains a relatively consistent and faster rate. These trends are quantitatively supported by Table 16, which shows Hindi reaching 157.58 WPM, and Kannada at 100.05 WPM.

Together, these observations highlight how the Clean slab offers tightly controlled speech characteristics, while the Semi-noisy slab introduces useful variability for training robust ASR models suited to real-world speech.

# I  Benchmarking ASR performance

Tables 18 and 19 present a detailed comparison of baseline ASR models evaluated on the RESPIN test set. The experiments span Fairseq-based, ESPnet-based, Whisper-based, and Kaldi-based systems, offering a broad perspective across different modeling frameworks. These results offer insights into how different modeling paradigms perform when confronted with the dialectal, phonetic, and structural diversity embedded in RESPIN.

The results show that models fine-tuned on RESPIN consistently outperform those trained or fine-tuned solely on external corpora. This confirms that RESPIN's dialect-rich coverage is essential for achieving reasonable performance across all nine languages. In particular, the differences in WER between fine-tuned and non-fine-tuned versions of the same model (e.g., SPRING-Data2Vec-AQC) illustrate how dialect-specific supervision significantly impacts accuracy.

Among the baselines, SPRING-Data2Vec-AQC (fine-tuned on RESPIN subset) performs best for almost all languages and dialects; E-Branchformer model trained from scratch on RESPIN performs competitively across many languages, especially those with high data coverage in the clean subset. Its performance in languages such as Hindi, Marathi and Telugu suggests that well-curated, supervised data can serve as a strong foundation even in the absence of large-scale pretraining.

Whisper models demonstrate robustness across languages, but tend to underperform compared to other RESPIN-finetuned models. In some languages, such as Chhattisgarhi and Magahi, the gap between Whisper and fine-tuned SSL baselines widens, indicating that Whisper's general-purpose training may not adequately capture the nuances of dialectal variation.

The detailed language-wise breakdown also reveals non-uniform trends in CER and WER. For instance, Dravidian languages like Kannada and Telugu often yield lower CERs but relatively higher WERs, which may reflect challenges in tokenization or word boundary segmentation. On the other hand, languages like Hindi and Bhojpuri tend to show tighter CER–WER alignment.

Variability across dialects is also evident, with certain dialects (e.g., D2, D4 of Hindi, D4 of Bengali, D3, D4 of Kannada) consistently resulting in higher error rates. This highlights the importance of dialect-level metadata in training and evaluation, which RESPIN-S1.0 explicitly encodes.

Table 18: Dialect-wise and overall CER and WER (%) for different fairseq-based models across languages. **Pretrained models** refer to models fine-tuned on publicly available data other than RESPIN. **Fine-tuned models** are pretrained SSL models further fine-tuned on a subset of RESPIN. For pretrained SSL models, **bh**, **ch**, **mg**, and **mt** are evaluated using Hindi-tuned models.

| Model | Dialect | CER (%) | | | | | | | | | WER (%) | | | | | | | | |
| --- | --- | --- | --- | --- | --- | --- | --- | --- | --- | --- | --- | --- | --- | --- | --- | --- | --- | --- | --- |
| | | bh | bn | ch | hi | kn | mg | mr | mt | te | bh | bn | ch | hi | kn | mg | mr | mt | te |
| **pre-trained IndicW2V2 (fine-tuned on non-RESPIN public data)** | | | | | | | | | | | | | | | | | | | |
| | D1 | 17.69 | 15.01 | 23.17 | 9.70 | 8.32 | 22.14 | 16.86 | 24.44 | 8.94 | 52.69 | 45.45 | 65.38 | 25.75 | 38.89 | 55.66 | 63.59 | 68.47 | 37.63 |
| | D2 | 15.76 | 12.64 | 22.27 | 13.35 | 5.00 | 17.11 | 15.32 | 23.08 | 8.47 | 48.54 | 38.38 | 65.45 | 34.32 | 21.92 | 48.95 | 52.81 | 65.38 | 38.93 |
| | D3 | 17.65 | 10.89 | 22.94 | 9.44 | 16.36 | 19.01 | 12.54 | 23.86 | 8.17 | 53.28 | 33.25 | 67.51 | 25.07 | 59.02 | 52.82 | 44.87 | 67.12 | 36.25 |
| | D4 | - | 17.09 | 22.75 | 12.54 | 17.96 | 20.99 | 15.50 | 21.84 | 8.90 | - | 52.33 | 65.70 | 31.49 | 65.06 | 62.68 | 53.68 | 63.58 | 38.64 |
| | D5 | - | 16.08 | - | 8.48 | 5.66 | - | - | - | - | - | 45.88 | - | 21.99 | 27.27 | - | - | - | - |
| | Overall | 17.08 | 14.27 | 22.77 | 11.02 | 10.37 | 19.64 | 15.09 | 23.30 | 8.61 | 51.61 | 42.83 | 65.98 | 28.34 | 42.37 | 54.32 | 53.91 | 66.10 | 37.82 |
| **pre-trained SPRING-W2V2 (fine-tuned on non-RESPIN public data)** | | | | | | | | | | | | | | | | | | | |
| | D1 | 15.75 | 11.81 | 20.70 | 8.68 | 9.50 | 18.30 | 9.38 | 22.11 | 6.57 | 42.38 | 26.03 | 54.07 | 23.66 | 39.48 | 44.48 | 40.43 | 57.98 | 33.05 |
| | D2 | 15.08 | 12.85 | 20.23 | 9.76 | 5.34 | 13.78 | 7.47 | 17.25 | 6.93 | 40.64 | 25.22 | 53.62 | 24.87 | 27.99 | 34.24 | 35.15 | 49.38 | 37.42 |
| | D3 | 14.48 | 11.39 | 20.88 | 8.23 | 16.79 | 15.74 | 5.43 | 20.12 | 6.80 | 40.86 | 21.82 | 55.89 | 20.98 | 56.79 | 41.67 | 25.78 | 54.19 | 35.31 |
| | D4 | - | 14.44 | 21.40 | 8.68 | 20.91 | 18.46 | 7.81 | 21.93 | 7.67 | - | 30.30 | 57.91 | 24.11 | 64.73 | 51.46 | 34.68 | 54.41 | 39.69 |
| | D5 | - | 12.03 | - | 8.33 | 6.07 | - | - | - | - | - | 26.65 | - | 20.55 | 32.71 | - | - | - | - |
| | Overall | 15.10 | 12.50 | 20.81 | 8.80 | 11.43 | 16.35 | 7.56 | 20.12 | 6.97 | 41.32 | 25.93 | 55.42 | 22.99 | 44.35 | 42.09 | 34.15 | 53.69 | 36.32 |
| **pre-trained SPRING-Data2Vec-AQC (fine-tuned on non-RESPIN public data)** | | | | | | | | | | | | | | | | | | | |
| | D1 | 15.83 | 11.17 | 21.16 | 6.31 | 8.73 | 17.14 | 9.60 | 20.77 | 5.90 | 43.67 | 23.99 | 55.35 | 20.74 | 37.37 | 44.33 | 40.64 | 56.27 | 31.01 |
| | D2 | 14.88 | 12.59 | 20.78 | 8.62 | 4.63 | 13.43 | 7.54 | 18.10 | 6.60 | 42.00 | 24.18 | 54.73 | 22.85 | 25.90 | 34.69 | 34.34 | 50.42 | 34.90 |
| | D3 | 14.38 | 11.36 | 21.39 | 6.09 | 15.07 | 15.49 | 5.22 | 20.73 | 6.43 | 41.37 | 21.55 | 56.26 | 18.86 | 55.92 | 42.68 | 24.85 | 55.95 | 32.50 |
| | D4 | - | 13.78 | 21.71 | 7.01 | 21.15 | 18.02 | 7.43 | 20.59 | 7.30 | - | 27.00 | 58.19 | 21.43 | 64.04 | 51.60 | 33.13 | 52.91 | 37.75 |
| | D5 | - | 10.80 | - | 7.25 | 5.57 | - | - | - | - | - | 22.08 | - | 19.89 | 30.65 | - | - | - | - |
| | Overall | 15.02 | 11.94 | 21.26 | 7.20 | 10.78 | 15.81 | 7.49 | 19.91 | 6.53 | 42.35 | 23.69 | 56.17 | 20.93 | 42.79 | 42.47 | 33.40 | 53.65 | 33.98 |
| **IndicW2V2 (fine-tuned on RESPIN subset)** | | | | | | | | | | | | | | | | | | | |
| | D1 | 4.45 | 4.09 | 3.13 | 2.59 | 4.15 | 6.76 | 4.54 | 5.7 | 4.78 | 15.89 | 15.57 | 11.13 | 9.14 | 23.46 | 23.90 | 21.61 | 21.43 | 23.55 |
| | D2 | 4.22 | 3.55 | 2.96 | 4.23 | 2.12 | 5.52 | 3.08 | 5.49 | 4.07 | 15.64 | 15.02 | 10.40 | 12.93 | 12.08 | 17.99 | 16.19 | 19.26 | 22.87 |
| | D3 | 4.57 | 3.01 | 2.88 | 1.92 | 7.05 | 5.55 | 2.11 | 4.4 | 4.41 | 16.63 | 11.29 | 10.56 | 7.29 | 35.98 | 21.49 | 8.66 | 17.55 | 23.82 |
| | D4 | - | 6.1 | 3.89 | 4.01 | 8.03 | 6.43 | 2.92 | 5.07 | 4.91 | - | 22.53 | 13.16 | 13.20 | 37.29 | 23.76 | 13.51 | 18.35 | 25.93 |
| | D5 | - | 4.75 | - | 2.18 | 2.68 | - | - | - | - | - | 19.13 | - | 7.77 | 15.98 | - | - | - | - |
| | Overall | 4.42 | 4.28 | 3.24 | 3.16 | 4.68 | 6.02 | 3.19 | 5.19 | 4.54 | 16.07 | 16.65 | 11.36 | 10.47 | 24.86 | 21.51 | 15.13 | 19.19 | 24.03 |
| **SPRING-W2V2 (fine-tuned on RESPIN subset)** | | | | | | | | | | | | | | | | | | | |
| | D1 | 3.94 | 3.45 | 2.75 | 2.04 | 3.75 | 5.77 | 3.62 | 4.80 | 3.81 | 14.81 | 13.22 | 9.56 | 7.61 | 21.89 | 20.87 | 18.02 | 18.12 | 20.05 |
| | D2 | 3.76 | 3.05 | 2.73 | 3.17 | 1.75 | 4.71 | 2.48 | 4.57 | 3.70 | 14.02 | 13.89 | 9.85 | 10.28 | 10.96 | 15.97 | 14.10 | 16.85 | 21.61 |
| | D3 | 4.04 | 2.82 | 2.75 | 1.49 | 6.75 | 4.94 | 1.39 | 3.74 | 3.70 | 14.93 | 9.83 | 10.47 | 5.60 | 35.82 | 19.81 | 6.33 | 15.19 | 21.51 |
| | D4 | - | 5.63 | 3.64 | 2.90 | 7.60 | 5.53 | 2.41 | 4.32 | 4.23 | - | 20.89 | 12.86 | 9.60 | 36.81 | 22.17 | 12.05 | 16.25 | 24.62 |
| | D5 | - | 4.40 | - | 1.69 | 2.33 | - | - | - | - | - | 17.94 | - | 6.69 | 14.55 | - | - | - | - |
| | Overall | 3.92 | 3.86 | 2.99 | 2.37 | 4.30 | 5.20 | 2.49 | 4.37 | 3.85 | 14.61 | 15.12 | 10.74 | 8.22 | 23.90 | 19.40 | 12.75 | 16.64 | 21.92 |
| **SPRING-Data2Vec-AQC (fine-tuned on RESPIN subset)** | | | | | | | | | | | | | | | | | | | |
| | D1 | 3.95 | 3.48 | 2.68 | 2.03 | 3.59 | 5.48 | 3.73 | 4.95 | 3.60 | 14.79 | 13.27 | 9.72 | 7.77 | 19.95 | 20.01 | 18.56 | 19.05 | 19.27 |
| | D2 | 3.81 | 2.74 | 2.55 | 2.90 | 1.71 | 4.41 | 2.32 | 4.46 | 3.61 | 14.34 | 12.34 | 9.20 | 9.13 | 11.04 | 15.26 | 13.25 | 16.22 | 21.48 |
| | D3 | 4.07 | 2.48 | 2.54 | 1.67 | 6.40 | 5.00 | 1.35 | 3.58 | 3.61 | 15.33 | 8.87 | 9.51 | 6.19 | 35.02 | 19.56 | 6.43 | 14.50 | 20.42 |
| | D4 | - | 5.14 | 3.49 | 2.49 | 7.22 | 5.16 | 2.04 | 4.17 | 4.10 | - | 19.33 | 12.37 | 8.68 | 35.73 | 20.09 | 10.36 | 15.78 | 23.64 |
| | D5 | - | 4.37 | - | 1.87 | 2.26 | - | - | - | - | - | 17.22 | - | 7.05 | 14.42 | - | - | - | - |
| | Overall | 3.95 | 3.63 | 2.84 | 2.27 | 4.11 | 4.98 | 2.38 | 4.30 | 3.72 | 14.84 | 14.15 | 10.25 | 7.91 | 23.13 | 18.50 | 12.28 | 16.41 | 21.17 |

Table 19: Dialect-wise and overall CER and WER (%) for different Whisper and traditional models across languages. **Fine-tuned models** are Whisper models further fine-tuned on a subset of RESPIN. **Traditional models** are trained from scratch on RESPIN.

| Model | Dialect | CER (%) | | | | | | | | | WER (%) | | | | | | | | |
|---|---|---|---|---|---|---|---|---|---|---|---|---|---|---|---|---|---|---|---|
| | | bh | bn | ch | hi | kn | mg | mr | mt | te | bh | bn | ch | hi | kn | mg | mr | mt | te |
| **Whisper-Tiny (fine-tuned on RESPIN subset)** | | | | | | | | | | | | | | | | | | | |
| | D1 | 9.45 | 11.72 | 6.81 | 7.27 | 12.38 | 16.8 | 10.8 | 11.69 | 13.2 | 26.84 | 32.65 | 19.22 | 19.43 | 46.98 | 40.19 | 37.21 | 34.8 | 44.05 |
| | D2 | 9.2 | 10.45 | 6.7 | 14.03 | 8.57 | 11.53 | 9.39 | 11.44 | 9.15 | 26.8 | 29.85 | 20.22 | 25.45 | 35.34 | 30.49 | 32.8 | 33.16 | 37.85 |
| | D3 | 10.14 | 9.1 | 6.44 | 6.85 | 17.14 | 13.11 | 6.74 | 8.88 | 10.29 | 28.62 | 26.04 | 18.82 | 17.37 | 61.88 | 35.51 | 22.16 | 28.6 | 40.28 |
| | D4 | - | 14.39 | 8.4 | 11.38 | 17.74 | 15.13 | 9.5 | 10.67 | 13.21 | - | 39.01 | 24.49 | 26.84 | 61.02 | 41.64 | 30.91 | 30.72 | 44.42 |
| | D5 | - | 12.53 | - | 6.26 | 8.43 | - | - | - | - | - | 35.55 | - | 16.2 | 38.16 | - | - | - | - |
| | Overall | 9.62 | 11.6 | 7.13 | 9.69 | 12.62 | 13.98 | 9.15 | 10.73 | 11.43 | 27.45 | 32.51 | 20.81 | 21.71 | 48.54 | 36.4 | 30.93 | 31.96 | 41.61 |
| **Whisper-Base (fine-tuned on RESPIN subset)** | | | | | | | | | | | | | | | | | | | |
| | D1 | 7.22 | 7.34 | 5.21 | 4.4 | 7.35 | 11.41 | 8.2 | 8.17 | 8.71 | 22.3 | 24.35 | 15.8 | 12.62 | 34.45 | 32.84 | 31.66 | 27.55 | 34.44 |
| | D2 | 6.73 | 7.05 | 4.81 | 7.9 | 4.68 | 8.94 | 6.51 | 8.02 | 6.52 | 22.11 | 22.52 | 15.48 | 19.27 | 23.38 | 26.05 | 26.28 | 25.09 | 30.88 |
| | D3 | 7.46 | 5.38 | 4.77 | 4.18 | 11.67 | 9.92 | 4.21 | 6.35 | 7.16 | 23.07 | 17.94 | 15.16 | 11.23 | 48.4 | 30.01 | 15.92 | 21.93 | 31.45 |
| | D4 | - | 10.19 | 6.48 | 7.24 | 12.69 | 12.04 | 5.84 | 7.35 | 7.62 | - | 30.92 | 19.89 | 18.45 | 49.88 | 35.06 | 22.59 | 24.38 | 35.38 |
| | D5 | - | 8.62 | - | 3.72 | 5.03 | - | - | - | - | - | 28.27 | - | 11.48 | 26.93 | - | - | - | - |
| | Overall | 7.15 | 7.69 | 5.36 | 5.8 | 8.1 | 10.44 | 6.23 | 7.51 | 7.51 | 22.51 | 24.71 | 16.67 | 15.19 | 36.52 | 30.54 | 24.28 | 24.8 | 32.99 |
| **Whisper-Small (fine-tuned on RESPIN subset)** | | | | | | | | | | | | | | | | | | | |
| | D1 | 8.43 | 4.96 | 3.86 | 3.28 | 5.52 | 7.99 | 5.6 | 6.97 | 6.24 | 19.44 | 17.93 | 11.81 | 10.1 | 27.73 | 24.5 | 23.93 | 23.79 | 27.16 |
| | D2 | 8.8 | 4.38 | 3.52 | 5.39 | 2.79 | 6.55 | 3.8 | 6.34 | 4.74 | 18.19 | 16.18 | 11.78 | 14.44 | 15.41 | 20.43 | 18.31 | 20.6 | 25.91 |
| | D3 | 6.61 | 3.77 | 3.51 | 2.87 | 9.09 | 7.19 | 2.57 | 4.72 | 8.81 | 19.34 | 14.24 | 11.32 | 8.8 | 41.89 | 24.18 | 9.66 | 17.02 | 28.1 |
| | D4 | - | 7.92 | 4.43 | 5.53 | 9.86 | 8.47 | 3.64 | 5.57 | 6.17 | - | 25.63 | 14.29 | 14.49 | 43.46 | 28.21 | 15.29 | 19.43 | 30.14 |
| | D5 | - | 6.34 | - | 2.58 | 3.57 | - | - | - | - | - | 20.87 | - | 8.95 | 20.39 | - | - | - | - |
| | Overall | 7.9 | 5.46 | 3.85 | 4.16 | 6 | 7.46 | 3.93 | 5.94 | 6.54 | 19.02 | 18.91 | 12.36 | 11.78 | 29.66 | 23.94 | 16.95 | 20.28 | 27.82 |
| **E-Branchformer (trained from scratch on RESPIN subset)** | | | | | | | | | | | | | | | | | | | |
| | D1 | 4.97 | 3.90 | 3.60 | 2.98 | 4.14 | 7.44 | 4.84 | 6.18 | 4.51 | 14.91 | 13.90 | 9.98 | 8.71 | 22.90 | 21.89 | 21.05 | 19.64 | 21.58 |
| | D2 | 4.87 | 3.86 | 3.10 | 4.62 | 1.85 | 5.82 | 3.01 | 5.90 | 3.19 | 14.82 | 13.86 | 9.78 | 12.18 | 11.64 | 16.55 | 16.07 | 17.79 | 19.72 |
| | D3 | 5.00 | 2.53 | 3.37 | 2.25 | 7.43 | 6.16 | 1.86 | 4.82 | 3.80 | 15.86 | 9.72 | 10.35 | 7.36 | 36.96 | 21.10 | 7.69 | 15.90 | 20.78 |
| | D4 | - | 6.56 | 4.37 | 4.45 | 8.44 | 7.61 | 2.92 | 6.05 | 4.40 | - | 20.67 | 12.10 | 11.99 | 37.06 | 23.28 | 12.51 | 18.41 | 24.62 |
| | D5 | - | 4.90 | - | 2.41 | 2.11 | - | - | | - | - | 16.94 | - | 7.79 | 14.52 | - | - | - | - |
| | Overall | 4.95 | 4.33 | 3.63 | 3.52 | 4.62 | 6.68 | 3.19 | 5.75 | 3.97 | 15.21 | 14.96 | 10.59 | 9.94 | 24.50 | 20.38 | 14.48 | 17.95 | 21.64 |
| **TDNN-HMM** (trained from scratch on RESPIN subset) | | | | | | | | | | | | | | | | | | | |
| | D1 | 5.80 | 5.04 | 4.25 | 2.57 | 4.04 | 8.87 | 4.52 | 6.90 | 4.35 | 18.02 | 15.22 | 11.88 | 7.56 | 20.71 | 24.30 | 17.94 | 21.52 | 19.72 |
| | D2 | 5.55 | 4.51 | 3.63 | 3.95 | 2.33 | 7.07 | 3.22 | 7.18 | 3.17 | 17.36 | 15.26 | 11.19 | 9.55 | 12.84 | 19.04 | 13.92 | 20.80 | 19.79 |
| | D3 | 5.66 | 3.46 | 3.80 | 2.31 | 7.42 | 7.05 | 2.44 | 5.47 | 4.10 | 17.31 | 11.71 | 11.41 | 6.24 | 33.34 | 22.91 | 8.97 | 18.24 | 21.47 |
| | D4 | - | 8.48 | 5.92 | 4.21 | 9.22 | 7.92 | 2.92 | 6.39 | 4.12 | - | 26.33 | 15.94 | 10.94 | 35.63 | 23.99 | 12.37 | 19.66 | 22.26 |
| | D5 | - | 4.80 | - | 2.40 | 2.18 | - | - | - | - | - | 16.26 | - | 7.89 | 12.57 | - | - | - | - |
| | Overall | 5.67 | 5.22 | 4.45 | 3.25 | 4.88 | 7.69 | 3.30 | 6.53 | 3.94 | 17.57 | 16.87 | 12.69 | 8.72 | 23.01 | 22.33 | 13.40 | 20.13 | 20.81 |

Table 20: Average public-test CER/WER (%) across all languages. **Pretrained** = models fine-tuned on public data other than RESPIN; **Traditional** = trained from scratch on RESPIN; **Fine-tuned** = pretrained SSL/Whisper further fine-tuned on a RESPIN subset.

| Model | CER (%) | | | | | | | | WER (%) | | | | | | | |
|---|---|---|---|---|---|---|---|---|---|---|---|---|---|---|---|---|
| | CV | FL | GV | IT | KB | KBN | MUCS | Avg | CV | FL | GV | IT | KB | KBN | MUCS | Avg |
| **Pretrained Models (fine-tuned on non-RESPIN public data)** | | | | | | | | | | | | | | | | |
| SeamlessM4T-v2-Large (PT) | **7.50** | 9.68 | 34.75 | 18.34 | 9.56 | 10.84 | 17.86 | 15.51 | **26.27** | 29.12 | 48.58 | 48.12 | 31.50 | 33.00 | 42.61 | 37.03 |
| IndicW2V2 (PT) | 13.13 | 14.07 | 21.28 | 16.09 | 8.17 | 10.58 | **5.32** | 12.66 | 42.55 | 36.82 | 48.70 | 44.20 | 32.36 | 36.90 | **21.53** | 37.58 |
| SPRING-W2V2 (PT) | 10.17 | 9.52 | 14.48 | **6.92** | 7.63 | 8.71 | 7.81 | 9.32 | 31.53 | 28.66 | 32.49 | **26.36** | 27.23 | 29.82 | 30.44 | 29.51 |
| SPRING-Data2Vec-AQC (PT) | 8.88 | **8.94** | 13.47 | 7.03 | **6.91** | **7.27** | 7.31 | **8.55** | 27.88 | **25.92** | 29.64 | 27.14 | **24.18** | **25.21** | 28.29 | **26.89** |
| **Traditional Models (trained from scratch on RESPIN subset)** | | | | | | | | | | | | | | | | |
| TDNN-HMM | 26.19 | 19.70 | 42.11 | 19.46 | 19.77 | 24.10 | 20.83 | 24.59 | 59.09 | 52.42 | 61.55 | 66.87 | 56.52 | 59.79 | 50.26 | 58.07 |
| E-Branchformer | 19.86 | 14.75 | 35.74 | 11.67 | 12.93 | 15.81 | 14.10 | 17.84 | 52.80 | 42.24 | 58.97 | 47.31 | 42.73 | 47.15 | 42.54 | 47.68 |
| **Fine-tuned Models (fine-tuned on RESPIN subset)** | | | | | | | | | | | | | | | | |
| Whisper-Tiny | 30.90 | 34.38 | 49.16 | 28.13 | 22.63 | 27.58 | 24.43 | 31.03 | 70.12 | 65.90 | 74.84 | 69.12 | 62.45 | 68.67 | 60.32 | 67.35 |
| Whisper-Base | 23.90 | 20.39 | 43.14 | 18.46 | 17.40 | 20.47 | 18.86 | 23.23 | 61.23 | 52.29 | 68.18 | 59.97 | 53.47 | 58.31 | 51.99 | 57.92 |
| Whisper-Small | 18.72 | 15.23 | 36.34 | 14.91 | 13.74 | 15.75 | 14.91 | 18.52 | 50.97 | 42.59 | 58.66 | 52.34 | 45.10 | 48.39 | 44.07 | 48.87 |
| IndicW2V2 (FT) | 14.71 | 14.51 | 19.59 | 14.17 | 10.49 | 12.63 | 10.90 | 13.86 | 45.05 | 37.22 | 44.77 | 45.40 | 38.42 | 42.71 | 39.35 | 41.85 |
| SPRING-W2V2 (FT) | 10.28 | 12.50 | 12.02 | 12.38 | 7.63 | 8.84 | 7.03 | 10.10 | 32.60 | 29.59 | 30.51 | 38.67 | 28.21 | 30.91 | 28.69 | 31.31 |
| SPRING-Data2Vec-AQC (FT) | 8.98 | 11.99 | **10.78** | 12.36 | 7.21 | 7.57 | 7.02 | 9.42 | 28.99 | 27.66 | **27.79** | 38.40 | 26.26 | 27.22 | 27.66 | 29.14 |

**Test-set abbreviations: CV**: CommonVoice, **FL**: FLEURS, **GV**: GramVaani, **IT**: IndicTTS, **KB**: Kathbath, **KBN**: Kathbath_Noisy, **MUCS**: MUCS.

Overall, these results highlight the utility of RESPIN-S1.0 as a robust benchmarking resource. The diversity in performance across models, languages, and dialects underscores the need for future work on dialect-adaptive modeling and context-aware evaluation. The RESPIN-S1.0 dataset offers a controlled and consistent framework for such comparative analyzes and will continue to facilitate reproducible benchmarking of ASR systems aimed at capturing the linguistic diversity of Indian languages.

## I.1 ASR performance on public test sets

Table 21 summarizes the language coverage of each public test set, indicating that not all corpora include all five languages. Therefore, the average CER/WER values reported in Table 20 are computed over the available languages for each test set. Pretrained models generally achieve the best performance across most test sets, with SPRING-Data2Vec-AQC (PT) and SPRING-W2V2 (PT) yielding the lowest average CER and WER. However, RESPIN fine-tuned models perform comparably well, despite being trained on data from only two domains—*agriculture* and *banking*. This demonstrates the linguistic diversity and robustness of RESPIN, which enables effective transfer learning and generalization to unseen corpora. SPRING-Data2Vec-AQC (FT) shows particularly strong cross-corpus performance, underscoring RESPIN's value as a benchmark for multidialectal, low-resource ASR. Implementation details and Zipformer results will be made available at: https://github.com/labspire/respin_baselines.git.

# J Ethical Protocols and Data Governance

## J.1 Contributor Onboarding and Validation Structure

A multi-tier validation framework was established to ensure linguistic and ethical quality throughout text data collection. The team comprised **Language Consultants (LCs)**, **Language Resource Managers (LRMs)**, **Senior LRMs**, and **technical reviewers** including speech scientists and computational linguists. LCs conducted dialect-level validation, while LRMs and Senior LRMs—linguists with advanced academic or field experience—oversaw inter-dialect consistency and annotation quality. Final reviews by technical experts

Table 21: Language availability per public test set (✓= available). Averages in Table 20 are computed only over languages marked as available for each test set.

| Test set | bn | hi | kn | mr | te |
|---|---|---|---|---|---|
| CommonVoice | ✓ | ✓ | — | ✓ | — |
| FLEURS | ✓ | ✓ | ✓ | ✓ | ✓ |
| GramVaani | — | ✓ | — | — | — |
| IndicTTS | ✓ | ✓ | ✓ | ✓ | ✓ |
| Kathbath | ✓ | ✓ | ✓ | ✓ | ✓ |
| Kathbath_Noisy | ✓ | ✓ | ✓ | ✓ | ✓ |
| MUCS | — | ✓ | — | ✓ | ✓ |

**Language codes:** bn = Bengali, hi = Hindi, kn = Kannada, mr = Marathi, te = Telugu.

verified linguistic accuracy, dialectal fidelity, and corpus-wide consistency. Figure 25 illustrates the validation team hierarchy.

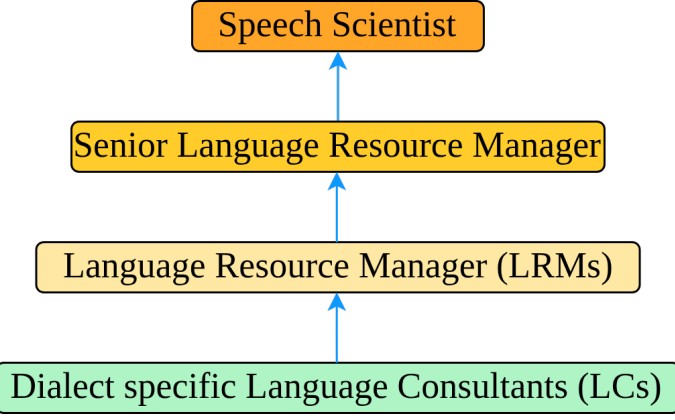

Figure 25: Validation team hierarchy used during data collection and quality control.

### J.2 Recruitment Frameworks and Data Collection Phases

**Phase 1: Domain Expert Recruitment and Collection.** Phase 1 targeted domain-specific question generation by **Domain Experts (DEs)** selected for their native fluency, geographic authenticity, and professional experience in agriculture or finance. Typical DEs included agricultural officers, field workers, and local farmers—often the end users of the envisioned ASR systems. After recruitment, DEs received verbal and written orientation via regional coordinators. They were trained to compose natural, contextually relevant questions in their dialect using appropriate local terminology (e.g., crop names, diseases, loan schemes).

Submissions were made through Google Forms distributed via messaging platforms, each covering a specific subdomain such as *weather*, *irrigation*, or *savings*. DEs were required to contribute seven unique questions per topic, avoiding repetition or translation artifacts. Compensation was tied to the number of accepted entries. Contributors received installation guides for regional keyboards, handwriting input tools, and short tutorial videos. Common issues—English mixing, transliteration errors, or fragments—were flagged for correction before final submission. Each DE typically had one to two weeks to complete assignments, followed by a seven-day technical validation before transfer to the LC-led review pipeline.

**Phase 2: Dialect Expert Recruitment for Text Expansion.** Phase 2 aimed to expand linguistic and domain coverage by collecting an additional 15,000 dialect- and domain-rich sentences per language. Online and offline drives recruited **Dialect Experts (DEs)** for two roles: *Sentence Composition* and *Sentence Translation*. Applicants completed pilot tasks assessing dialect knowledge, grammar, and writing fluency, followed by structured training on syntax, domain terminology, and submission protocols.

**Sentence Composers** generated original, grammatically accurate sentences (8–15 words) in their native dialects for agriculture and finance domains. Minimum requirements included undergraduate-level education, native dialect proficiency, and digital literacy (Google Docs and basic tools). They worked remotely and were encouraged to maximize lexical diversity while avoiding copied or machine-translated text.

**Sentence Translators** converted standard dialect sentences into regional variants, maintaining semantic equivalence and grammatical integrity. They demonstrated control over both formal and informal registers and were evaluated through pilot tasks before onboarding. Translators worked remotely using collaborative documents, submitting regular batches for validation.

## J.3  Validation and Review Workflow

All Phase 1 and Phase 2 submissions passed through a unified validation pipeline. Data were first standardized and grouped by language, dialect, and domain, then reviewed by LCs for correctness, dialectal fidelity, and semantic clarity. Each validated entry included a normalized sentence, English translation, and feature tag (e.g., crop, pest, finance), enabling structured downstream filtering. LC-reviewed files were escalated to LRMs and Senior LRMs for secondary review.

Automated scripts assisted validators by flagging anomalies such as spelling errors, punctuation inconsistencies, English insertions, or unsupported characters. These flags appeared as columns in shared Google Sheets, allowing validators to filter and resolve issues efficiently.

## J.4  Contributor Communication and Management

**Coordination and Communication.** Contributors working on under-represented dialects received templated instruction sheets specifying translation rules, formatting conventions, and turnaround expectations. Dedicated communication channels via email and WhatsApp ensured real-time clarification.

**Compensation and Incentives.** Payment was performance-based and transparent. For composers and translators, remuneration was proportional to the number of accepted entries, with adjustments for complexity or length. Validators were compensated by validated entry volume and review quality. Compensation slabs were communicated during onboarding.

**Recruitment and Outreach.** Outreach materials—including WhatsApp banners, email flyers, and community posters—were customized by region and dialect to reach both rural and semi-urban contributors effectively.

**Contributor Metadata and Demographics.** Contributor details such as district, dialect, role, and language were collected and anonymized for audit and demographic analysis. These records also informed dialectal balance across the corpus.

**Domain Preparation Resources.**  All contributors received curated resources, including domain sublists (e.g., irrigation, pest control, loan schemes), topic-specific vocabulary sets, and question-framing guidelines to support consistency in linguistic style.

**Pilot Task Evaluation.**  Before onboarding, each contributor completed pilot tasks reflecting actual assignment types. Evaluations measured grammar, dialect usage, and formatting adherence. Personalized feedback ensured quality alignment prior to full-scale contribution.

# K  Ethics and Participant Onboarding

## K.1  Demographic Statistics

We summarize high-level participant demographics to highlight geographic and sociolinguistic diversity. Table 22 shows the gender, age-group, and regional distributions derived from verified metadata. All languages include contributions from both male and female speakers spanning 18–60 years of age, with representation from 24 states and Union Territories.

Table 22: Aggregated participant demographics in the RESPIN-S1.0 corpus.

| Attribute | Coverage | Notes |
|---|---|---|
| Gender | 57% Male / 43% Female | Balanced across languages |
| Age Group | 18–25 (22%), 26–40 (54%), 41–60 (24%) | Mean  31 yrs |
| Regional Coverage | 24 States + UTs | Derived from postal codes |

## K.2  Compensation and Wage Compliance

Participants were paid a task-based honorarium ranging from  550 –  750 per completed task ( 557 utterances), corresponding to an **effective rate of  275 per hour**. This rate is **3–5× higher** than the regional minimum wages prescribed by the Government of India, ensuring fair remuneration across skill levels and locations. To verify compliance, we refer to the official notification issued by the *Office of the Chief Labour Commissioner (Central), Ministry of Labour and Employment*, effective April 1, 2025. Table 23 lists the published daily wage rates.

Table 23: Regional minimum daily wage rates (April 1 2025) — Government of India.

| Worker Category | Area A (Metro) | Area B (Urban) | Area C (Rural) |
|---|---|---|---|
| Unskilled | 514/day ( 64.25/hr) | 470/day ( 58.75/hr) | 465/day ( 58.13/hr) |
| Skilled | 610/day ( 76.25/hr) | 562/day ( 70.25/hr) | 515/day ( 64.38/hr) |
| Highly Skilled | 675/day ( 84.38/hr) | 628/day ( 78.50/hr) | 562/day ( 70.25/hr) |

Hourly equivalents assume 8-hour workdays. RESPIN's  275/hr rate is therefore substantially higher than the highest prescribed category in Area A, ensuring ethical compensation.

## K.3 Participant Instructions and Consent

The onboarding workflow combined automated and manual steps: (i) field coordinators provided **task instructions** and explained compensation, privacy, and consent requirements in local languages; (ii) participants viewed an onboarding video tutorial: `https://www.youtube.com/watch?v=7uZjAE3uRS0`; (iii) within the Bolo App, users completed a **digital consent form** and accepted a click-wrap privacy policy before recording; and (iv) only participants who digitally consented could proceed to tasks. Screenshots of the mobile interfaces illustrating onboarding and payments are shown in Figures 6 and 7. The consent form text is reproduced below.

**Excerpt from Digital Consent Form.**

> By proceeding, I confirm that I am participating voluntarily. I understand that my recordings may be used for research and model development and that my identity will not be disclosed. I consent to data storage and processing under the RESPIN project and may withdraw my participation at any time.

## K.4 Consent Withdrawal and Data Rights

Participants retained the right to revoke consent at any time. The Privacy Policy contained the following clause:

> **"C. Your Choices — Access to Personal Data or Information."** You may request to access, modify, or delete information held by us. You may withdraw consent by contacting `operations@navanatech.in`. Any withdrawal request will be honored and processed by the data team, though it may disable access to certain app features.

All such requests received via email were logged and resolved by the data-collection team within 72 hours.

## K.5 Ethics Approval and PII Safeguards

The RESPIN project received institutional ethics clearance from the *Indian Institute of Science (IISc) Bangalore.* No personally identifiable information (names, phone numbers, addresses, or photographs) was retained in the released corpus. Only anonymized IDs and coarse location codes were stored. All data were encrypted during transfer and hosted on secure IISc servers.

## K.6 Summary

RESPIN-S1.0 adhered to the principles of transparency, fair compensation, and informed consent. All ethical documentation, including the consent form, task instructions, and the wage-compliance table, is released alongside the dataset for reproducibility.

