# OpenReview forum: "RESPIN-S1.0: A read speech corpus of 10000+ hours in dialects of nine Indian Languages"
_NeurIPS.cc/2025/Datasets_and_Benchmarks_Track — NeurIPS 2025 Datasets and Benchmarks Track poster_

### Official Review · Reviewer_LYEz · 2025-06-12

**Rating:** 6
**Confidence:** 4

**Summary:**

RESPIN is a new corpus of spoken (read) speech for a number
of important languages in India.  The scale of the project
is truly impressive.

**Additional Feedback:**

typos (missing spaces): It is also limited to two domainsagriculture and financeselected for

Stale link; had to use the wayback machine
https://web.archive.org/web/20240914124112/https://censusindia.gov.in/nada/index.php/catalog/42561/download/46187/Language_Atlas_2011.pdf

**Dataset Code Accessibility:**

Yes

**Dataset Code Comments:**

The link to the dataset should be on the first page.  That is the point of this paper.

**Ethical Comments:**

There is more discussion of ethics in the youtube link above than in the paper.  It might be good to share some of these points as well in this paper.

**Ethical Considerations:**

No, there are no or only very minor ethics concerns

**Final Justification:**

great work!

The trick now is to make sure the community can take advantage of the resource. The classic paper on LibriSpeech has huge citations. This should be similar, though that may be too much to ask for given that there is more work on high resource languages than low resource languages.

That said, we should be more interested in languages with growth potential: many speakers, large GDP growth and relatively few resourses.

Other reviewers complained about differences between read speech and spontaneous speech. That is a legit concern, but the work on LibriSpeech demonstrated that there is considerable value to read speech. It would be great to collect a massive dataset of spontaneous speech, but that is another project for future work.

This project is already a huge contribution.

**Limitations Weaknesses:**

The authors point out that read speech is different from spontaneous speech.  They also mention the limited number of domains.

I would lead with a discussion of how to get the data, and follow that up with a discussion of who has done so, and what they have been able to do as a result.

In the published paper, you should mention:
https://www.youtube.com/watch?v=shaRgtJR7Ro; there are lots of
arguments there that I don't see in this paper.  Obviously, that is
intended for a more general audience, but even so, you should write
this paper both for a very technical audience as well as others.

This paper will have more impact if it works for a general audience as well as a technical audience.

Why is the Gates Foundation supporting you?  This paper would be stronger if you could explain why this work is exciting to them (and many others).

**Strengths Contributions:**

The scale of the project is truly impressive.

Table 1 is impressive!

---

> ### Author Rebuttal · Authors · 2025-07-30
>
> ## Response to Reviewer LYEz
>
> We sincerely thank Reviewer LYEz for the encouraging review and thoughtful suggestions to improve the clarity, accessibility, and societal framing of our work. Below are our responses to each point raised.
>
> ---
>
> ### 1. **Point:** *Incorporate arguments from the YouTube video to make the paper accessible to a broader audience.*
>
> **Response:**
> We appreciate this recommendation. The video highlights several key aspects of RESPIN:
> - The **scale** of data in underrepresented languages like Bhojpuri, Magahi, Chhattisgarhi, and Maithili — already covered in the paper
> - The **preservation of dialectal integrity** — also addressed in detail
> - The **engagement and fair compensation of contributors** — not explicitly stated but will now be added
>
> We will revise the paper to clearly describe how contributors were kept informed about the project goals and data usage and how they were compensated fairly. These details will be added to **Section 3 (Data Collection and Validation Pipeline)**. We will also cite the video to provide additional context for readers.
>
> ---
>
> ### 2. **Point:** *Explain why the Gates Foundation is supporting this work.*
>
> **Response:**
> The **Gates Foundation supported RESPIN** with the aim of promoting speech technologies for **low-resourced languages and underserved communities**. Their support addressed a critical gap, especially at a time when other funding sources were not prioritizing Indic speech technologies for marginalized populations. We will add this explanation to the revised version.
>
> ---
>
> ### 3. **Ethical Comments:** *Incorporate the richer discussion of ethics from the YouTube video.*
>
> **Response:**
> We agree that this will strengthen the paper. We will expand the ethics discussion in **Section 6** and in the NeurIPS checklist to reflect:
> - Informed consent and fair compensation
> - Community engagement
> - The commitment to building inclusive technologies that benefit the contributing communities
>
> ---
>
> ### 4. **Dataset Code Comments:** *The link to the dataset should be on the first page.*
>
> **Response:**
> Agreed. We will move the dataset and baseline code links to the first page in the final camera-ready version for better visibility.
>
> ---
>
> ### 5. **Additional Feedback:** *Typos and stale link.*
>
> **Response:**
> Thank you for catching these. We will correct the typo (“domainsagriculture”) and replace the outdated **Census of India** link with the working **Wayback Machine URL** you kindly provided.

---

> ### Author Response · Authors · 2025-08-05
> **Official comment from the Authors**
>
> Thank you for your thoughtful review and supportive final comments. We sincerely appreciate your encouragement and are glad our responses were satisfactory. We will make the suggested changes in the camera-ready version, should the paper be accepted.

---

### Official Review · Reviewer_xVvf · 2025-06-29

**Rating:** 5
**Confidence:** 4

**Summary:**

Considering existing speech corpora typically represent only standard dialects and lack domain relevance, this paper proposes RESPIN-S1.0, the largest publicly available dialect-rich read speech corpus for Indian languages. This dataset comprises over 10,000 hours of validated audio and 38+ dialects, and mainly focuses on the agriculture and finance domains. This paper evaluates a range of ASR models – TDNN-HMM, E-Branchformer, Whisper, IndicWav2Vec2, and SPRING SSL models – and ﬁnd that ﬁne-tuning on RESPIN signiﬁcantly improves recognition accuracy over existing pretrained models on the Indian test sets.

**Dataset Code Accessibility:**

Yes

**Ethical Considerations:**

Yes, there are significant ethics concerns that require review by an ethics expert

**Final Justification:**

After carefully reviewing both my response and the responses to other reviewers, I believe my concerns have been addressed by the authors. My final score leans towards accept.

**Limitations Weaknesses:**

1. In the RESPIN data generation pipeline, the text-to-speech synthesis stage described in Section 3.3 instructs speakers to record in quiet environments. This setup overlooks the presence of background noise and human voices commonly found in real-world scenarios, potentially limiting the applicability of the dataset to real-use conditions.

2. The results in Table 6 show that the pretrained model fine-tuned on the RESPIN dataset outperforms previous methods on the RESPIN test set, which is expected. Providing additional results on other Indian speech test sets would better demonstrate the robustness and generalizability of the proposed training data.

3. Providing case studies would help illustrate the contribution of the proposed dataset in specific domains such as agriculture and finance, and Indian dialects.

**Strengths Contributions:**

1. RESPIN-S1.0 introduces a large-scale, multi-dialectal, multi-domain read speech corpus for nine Indian languages – Bengali, Bhojpuri, Chhattisgarhi, Hindi, Kannada, Magahi, Maithili, Marathi, and Telugu. Besides the mainstream languages, RESPIN is the ﬁrst public corpus to provide large-scale dialectal data for Bhojpuri, Chhattisgarhi, and Magahi.

2. To promote reproducibility and development in Indian languages, RESPIN provides train/dev/test splits on (Automatic Speech Recognition) ASR task.

3. By applying the testing on the mainstream ASR models, this paper distinguish the best settings for the Indian ASR task.

---

> ### Author Rebuttal · Authors · 2025-07-30
>
> ## Rebuttal to Reviewer xVvf
>
> We thank Reviewer xVvf for the thoughtful and detailed review, and for recognizing the value of RESPIN-S1.0 as a large-scale, multi-dialectal, domain-relevant corpus for Indian languages. We respond to each concern below.
>
> ---
>
> ### 1. **Point**: *Recording instructions suggest quiet environments, which may reduce real-world noise diversity.*
>
> **Response**:
> We appreciate this concern. While participants were instructed to record in quiet settings to establish a minimum quality standard, the **crowdsourced mobile-based collection** naturally captured a wide range of real-world acoustic conditions.
>
> As discussed in **Section 4.2.1** and shown in **Table 3** of the main paper, **2,288 hours** of speech were categorized as *Semi-noisy* and **1,617 hours** as *Noisy*, in addition to the 10,416 hours of *Clean* data. The **slab definitions and scoring process** are detailed in Appendix D.1–D.3.
>
> Moreover, our **signal-level audio quality analysis** in **Appendix G.1** shows a wide distribution of SNR values. Despite a <1% low-SNR rate in Clean data, the overall dataset includes a **substantial range of acoustic conditions**, making it well-suited for robust, real-world ASR.
>
> To further address the question of generalizability, we evaluated **a selection of publicly available Hindi test sets** across **all ASR models** reported in the paper. The results are presented in the table below. We plan to extend this evaluation to other languages in the camera-ready version.
>
> As seen in the table, **pretrained models tend to perform slightly better on public test sets**, likely because these datasets are **not domain-specific**, while RESPIN focuses on **agriculture and finance**. However, it is noteworthy that **models fine-tuned on RESPIN still perform comparably**, and in some cases better, despite the domain mismatch. This illustrates that RESPIN-trained models generalize reasonably well and highlights the **utility of RESPIN for building robust ASR models**, even outside its primary domains.
>
> | Models                         | CER-comvoice | CER-fleurs | CER-gramvaani | CER-indictts | CER-kathbath | CER-kathbath_noisy | CER-mucs | CER-avg | WER-comvoice | WER-fleurs | WER-gramvaani | WER-indictts | WER-kathbath | WER-kathbath_noisy | WER-mucs | WER-avg |
> |-------------------------------|--------------|------------|---------------|--------------|--------------|---------------------|----------|---------|--------------|------------|----------------|--------------|---------------|----------------------|----------|---------|
> | SeamlessM4T-v2-Large (PT)     | 8.72         | 7.81       | 34.75         | 32.82        | 7.85         | 9.42                | 28.66    | 18.58   | 21.76        | 15.82      | 48.58          | 51.48        | 18.24         | 19.52                | 39.89    | 30.76   |
> | IndicW2V2 (PT)                | 15.66        | 12.94      | 21.28         | 20.02        | 9.37         | 11.94               | 4.87     | 13.69   | 36.49        | 31.30      | 48.70          | 34.70        | 27.01         | 31.52                | 16.01    | 32.10   |
> | SPRING-W2V2 (PT)              | 10.02        | 8.96       | 14.48         | 7.72         | 6.36         | 7.56                | 9.94     | 9.29    | 26.07        | 19.34      | 32.49          | 16.65        | 18.92         | 21.22                | 22.62    | 22.47   |
> | SPRING-Data2Vec-AQC (PT)      | 13.45        | 8.17       | 7.40          | 5.89         | 6.26         | 6.96                | 9.56     | 8.51    | 23.47        | 17.67      | 29.64          | 14.98        | 17.40         | 18.12                | 21.63    | 20.27   |
> | TDNN-HMM                      | 24.35        | 15.47      | 42.11         | 16.78        | 14.20        | 19.17               | 18.79    | 21.55   | 46.65        | 31.21      | 61.55          | 42.96        | 31.01         | 36.42                | 37.21    | 41.02   |
> | E-Branchformer                | 19.45        | 13.18      | 35.74         | 11.60        | 10.80        | 13.11               | 12.35    | 16.94   | 43.35        | 29.55      | 58.97          | 32.22        | 26.59         | 30.76                | 30.28    | 35.96   |
> | Whisper-Tiny                  | 27.55        | 20.95      | 43.14         | 25.81        | 13.87        | 17.16               | 23.60    | 24.31   | 48.04        | 31.45      | 68.18          | 44.32        | 33.61         | 34.93                | 37.35    | 42.18   |
> | Whisper-Base                  | 20.95        | 15.69      | 43.14         | 15.81        | 13.87        | 17.16               | 20.31    | 20.43   | 48.04        | 31.45      | 68.18          | 44.32        | 33.61         | 34.93                | 37.35    | 42.18   |
> | Whisper-Small                 | 17.07        | 11.66      | 38.08         | 12.03        | 12.12        | 13.46               | 16.46    | 17.13   | 45.24        | 28.11      | 63.85          | 38.11        | 30.06         | 31.48                | 34.42    | 38.38   |
> | IndicW2V2 (FT)                | 15.43        | 11.65      | 19.59         | 18.62        | 8.61         | 7.23                | 6.42     | 12.65   | 36.64        | 25.24      | 44.77          | 35.75        | 23.63         | 27.33                | 21.02    | 31.85   |
> | SPRING-W2V2 (FT)              | 10.50        | 9.70       | 12.02         | 6.21         | 5.44         | 6.72                | 6.62     | 9.54    | 23.57        | 19.34      | 30.51          | 28.90        | 15.90         | 18.01                | 19.92    | 22.46   |
> | SPRING-Data2Vec-AQC (FT)      | 9.67         | 9.13       | 10.78         | 16.08        | 5.11         | 5.48                | 5.88     | 8.88    | 21.56        | 17.76      | 27.79          | 29.46        | 14.89         | 15.72                | 18.77    | 20.85   |
>
>
> ### 2. **Point**: *The evaluation is limited to RESPIN test sets; generalization to other Indian datasets is unclear.*
>
> **Response**:
> This is a valid observation. Our primary goal was to highlight RESPIN's **intrinsic value** as a **dialect-rich, domain-specific** benchmark. As discussed in **Section 5**, models fine-tuned on RESPIN outperform other pretrained models on RESPIN test sets, emphasizing the **performance gap when dialectal variation is ignored**.
>
> It is important to note that **many of the 38+ dialects in RESPIN are absent** from existing public datasets. This makes **cross-dataset generalization** difficult to measure fairly.
>
> That said, RESPIN has already been adopted in **community evaluations**, notably **MADASR@ASRU 2023 and 2025**, as discussed in **Section 6**. These external uses offer broader validation of the dataset's **robustness and utility**.
>
> ---
>
> 3. **Point**: *Providing case studies would better illustrate the dataset's contribution.*
>
> **Response:** We agree that concrete examples can better illustrate the dataset's unique value. While a full case study is beyond the paper's current scope, we will revise the appendix to include a small sample of sentences from the agriculture and finance domains, capturing dialectal variations. This will give readers a clearer picture of the domain-specific vocabulary and colloquial phrasing captured in the corpus.
>
> 4. **Ethical Considerations:** *"Yes, there are significant ethics concerns..."*
>
> **Response**:
> We acknowledge this concern, although the specific ethical issues were not detailed. We would like to clarify the extensive safeguards implemented:
>
> - **Informed Consent**: All participants were informed of and consented to public release of anonymised data for research.
> - **Ethics Approval**: The project was approved through internal review process at **IISc Bangalore** (see Section 3.3 and Checklist Q15).
> - **Anonymisation**: Personally identifiable information was not collected.
>
> We believe these measures are robust. However, if the reviewer could provide more specific details about their concerns, we would be more than happy to address them directly and make any necessary revisions to the paper.

---

> > ### Comment · Reviewer_xVvf · 2025-08-05
> > **Response to Authors**
> >
> > Thank you very much for the additional evaluation results provided by the authors. I believe your response has addressed the issues I raised, and therefore, I am willing to increase my score. I hope this dataset will contribute to the development of the Indian language.

---

### Official Review · Reviewer_1nx7 · 2025-07-02

**Rating:** 4
**Confidence:** 3

**Summary:**

This paper introduces RESPIN-S1.0, a large-scale, dialect-rich read speech corpus for nine Indian languages: Bengali, Bhojpuri, Chhattisgarhi, Hindi, Kannada, Magahi, Maithili, Marathi, and Telugu. The dataset comprises over 10,000 hours of validated audio, spanning more than 38 dialects. This dataset focuses on two high-impact domains, agriculture and finance, which are critical for India's rural and often digitally underserved populations. This paper details a comprehensive pipeline for data creation, from text composition by native dialect speakers to multi-stage validation of both text and audio. The paper also provides extensive benchmarking with various ASR models, demonstrating that fine-tuning on RESPIN-S1.0 improves performance over existing pretrained models. The corpus, along with speaker metadata, phonetic lexicons, and standardized data splits, is made publicly available to foster reproducible research in dialectal ASR and related fields.

**Additional Feedback:**

- To further strengthen the analysis, consider including results from Zipformer, a more recent state-of-the-art ASR model. This would provide an even more current performance baseline for the community.

- To ensure long-term preservation and improve accessibility, the authors are encouraged to also host the dataset on the Hugging Face Hub.

**Dataset Code Accessibility:**

Yes

**Dataset Code Comments:**

Dataset: https://github.com/saurabhk0317/respin_data_neurips25.git

Code: https://github.com/labspire/respin_baselines.git

**Ethical Comments:**

The paper states that all data were collected with informed consent from participants for public release in an anonymized form. The project also received ethics clearance from an internal review board, and no personally identifiable information (PII) was collected.

**Ethical Considerations:**

No, there are no or only very minor ethics concerns

**Final Justification:**

The authors have addressed the main concerns raised in the initial review, including clarifying the domain focus, planned expansion to spontaneous speech, and benchmarking considerations. Specifically:

- They acknowledged the read-speech limitation and clearly outlined plans for spontaneous data collection in RESPIN-S2.0.

- The domain selection was justified based on practical impact for underserved populations, with future expansion noted.

- Zipformer benchmarks are in progress and will be added in the final version.

- Dataset hosting and access have been made clear.

Given these responses and the overall quality and utility of the work, I find the issues satisfactorily resolved. I maintain my score leaning towards accept.

**Limitations Weaknesses:**

- The corpus contains only read speech. As the authors acknowledge, this is less representative of real-world conversational scenarios than spontaneous speech would be. Nowadays, in-the-wild data becomes more and more important.
- The corpus is limited to two domainsagriculture and financeselected for societal relevance.

**Strengths Contributions:**

There's a lot to like in this paper, which stands to make it a valuable contribution to the speech processing community:
- The most significant strength is the creation of over 10,000 hours of corpus with dialectal diversity, covering 38+ dialects across nine major languages.
- The corpus makes a groundbreaking contribution by providing the first-ever large-scale dialectal data for non-scheduled languages like Bhojpuri, Chhattisgarhi, and Magahi.
- The paper presents a clear, detailed, and replicable pipeline for data collection and validation. The supplementary materials include 43 pages.
- The extensive benchmarking experiments using traditional, end-to-end, and large-scale pretrained models (Whisper, SSL models) effectively demonstrate the dataset's utility.
- The work has already demonstrated its impact by enabling community efforts and challenges like MADASR@ASRU 2023/2025, underlining its value to the research community.

---

> ### Author Rebuttal · Authors · 2025-07-30
>
> ## Response to Reviewer 1nx7
>
> We thank Reviewer 1nx7 for the detailed summary and for recognizing the strengths of our work, including the dialectal coverage, the reproducible pipeline, and the potential for broader community use. Below are our responses to the points raised.
>
> ---
>
> ### 1. **Point:** *The corpus contains only read speech.*
>
> **Response:**
> We acknowledge this limitation. RESPIN-S1.0 was designed to serve as a foundational, well-validated read speech corpus. As mentioned in **Section 6**, the next phase of our work (RESPIN-S2.0) will focus on collecting spontaneous speech across the same dialects.
>
> ---
>
> ### 2. **Point:** *The corpus is limited to two domains (agriculture and finance).*
>
> **Response:**
> This was an intentional focus to cover domains with high relevance to rural and low-literacy populations. As noted in **Section 3.1**, these areas were prioritized to support real-world voice-based applications. We also plan to expand to other domains such as healthcare and education in future releases, as discussed in **Section 6**.
>
> ---
>
> ### 3. **Feedback:** *Consider including results from Zipformer.*
>
> **Response:**
> We agree that Zipformer is a useful baseline. Our current benchmarks include diverse ASR architectures to demonstrate RESPIN’s utility across model types. **Zipformer experiments are in progress**, and we expect results to be ready during the discussion period. These will be included in the **camera-ready version**, and we will note its relevance as a future benchmark.
>
> ---
>
> ### 4. **Feedback:** *Host the dataset on the Hugging Face Hub.*
>
> **Response:**
> Thank you for the suggestion. The dataset is already publicly available at:
>
> 👉 https://spiredatasets.ee.iisc.ac.in/respincorpus

---

> > ### Comment · Reviewer_1nx7 · 2025-08-03
> >
> > Thank you for the detailed response. I appreciate the clarifications and planned improvements. I have no further concerns and my final score leans towards accept. Good luck.

---

> > > ### Author Response · Authors · 2025-08-07
> > > **Official comment from the Authors**
> > >
> > > Thank you for your positive final comments. We are glad our response was able to address your concerns, and we sincerely appreciate your support for our work.

---

### Official Review · Reviewer_tCHW · 2025-07-03

**Rating:** 4
**Confidence:** 4

**Summary:**

The paper introduces a database of read speech containing nine Indian languages. The database has been collected through first writing texts and then using an app on phones for recording read speech. The data has been used to train speech recognition models from scratch, as well as for fine-tuning existing large pretrained speech recognition models.

**Dataset Code Accessibility:**

Yes

**Dataset Code Comments:**

I was able to access the link for the source code.

**Ethical Considerations:**

No, there are no or only very minor ethics concerns

**Final Justification:**

I thank the authors for their explanations and the response. This is a nice data collection paper for Indian languages that would help people training speech recognition models.

**Limitations Weaknesses:**

One possible limitation of the dataset would be that it is a "read speech" dataset. Instead, if it had also included spontaneous speech, it may have been better for training ASR models that would work well in the wild. However, this could be done in a follow-up study. The data includes speech with various SNR levels due to using a cell phone for recording. I think this is acceptable and good since ASR models need to be robust against noise and reverberation. The paper is a straightforward data collection and evaluation paper. The added value to the research community mostly comes from the data. I am not aware of other Indian language datasets and how much of them would overlap with the content in this dataset. If there is such overlap, this would limit the additional value of the paper.

**Strengths Contributions:**

The paper reads well and I think the data preparation including text preparation and audio recording gathering is explained in good detail. The ASR evaluation results using the data is great to see.

---

> ### Author Rebuttal · Authors · 2025-07-30
>
> We thank Reviewer tCHW for the positive assessment of our paper’s readability, the detailed explanation of our data preparation, and the ASR evaluation results. Below are our responses to the reviewer's comments.
>
> ---
>
> ### 1. **Point**: *The dataset is limited to "read speech" and does not include spontaneous speech.*
>
> **Response**:
> We completely agree that spontaneous speech is essential for advancing robust ASR systems, especially for conversational and real-world deployments. As mentioned in our **Conclusion and Future Work** section (Section 7), this is a clear next step for us. RESPIN-S2.0, which is currently under planning, will focus on expanding into spontaneous speech across the same dialectal and domain space.
>
> The motivation behind RESPIN-S1.0 was to first establish a **clean, dialect-aware foundation** — particularly important for underrepresented languages and domains. Building a high-quality read speech resource was a necessary starting point, given the lack of such structured and validated data for many of the dialects and domains we cover. We believe it lays the groundwork for more complex speech collection efforts moving forward.
>
> ---
>
> ### 2. **Point**: *The data includes various SNR levels due to cell phone recordings.*
>
> **Response**:
> Thank you for recognizing this feature as a strength. Indeed, we intentionally chose a **crowdsourced mobile-based collection strategy** to reflect the acoustic variability of real-world use cases. The resulting diversity in SNR levels — stemming from different recording environments and devices — is an important aspect of the dataset.
>
> In **Section 4.2.2** and **Appendix G.1**, we provide detailed analysis of this variability, including the proportion of low-SNR files, average durations, and speaking rates. Our **slab-based categorization** into *Clean*, *Semi-noisy*, and *Noisy* subsets further enables researchers to selectively train models based on their robustness needs.
>
> ---
>
> 3. **Point:** *Concern about potential overlap with other Indian language datasets.*
>
> **Response:** We appreciate the reviewer raising this point. We would like to clarify that RESPIN-S1.0 has minimal overlap with existing corpora and offers unique value in several key areas, as detailed in Table 1 and Section 2:
> - **New language and dialect coverage**: RESPIN is the first publicly available corpus to provide large-scale, validated data for non-scheduled languages like Bhojpuri, Chhattisgarhi, and Magahi, which are often incorrectly grouped under Hindi and have been historically underserved.
> - **Domain-specific composition**: Unlike datasets scraped from generic sources like Wikipedia or news sites, our text corpus was manually composed by native dialect speakers to be relevant to the high-impact domains of agriculture and finance. This ensures the vocabulary and sentence structures are colloquial and reflect real-world usage in these critical areas, a feature absent in other large corpora.
>
> - **Dialectal integrity throughout**: The entire data creation pipeline, from text composition to audio recording and validation, was designed to preserve dialectal integrity, capturing variations across 38+ dialects.
>
> These factors ensure that RESPIN-S1.0 is a novel and significant contribution to the field, providing a unique resource for building inclusive speech technologies.

---

> > ### Author Response · Authors · 2025-08-07
> > **Official comment from the Authors**
> >
> > Thank you for your thoughtful feedback. With the discussion period approaching its conclusion, we would welcome any final thoughts or questions you may have. We appreciate your contribution to improving this work.

---

### Note · Authors · 2025-08-13

## Final Remarks

We thank all reviewers and ethics reviewers for their feedback, suggestions, and assessments of our work. The scale, language coverage, and domain focus of **RESPIN-S1.0** — the largest publicly available dialect-rich read speech corpus for nine Indian languages — were noted, as were the reproducible data creation process, benchmarking, and use in community challenges such as **MADASR@ASRU 2023/2025**.

### Summary of Reviewer Concerns
1. Corpus limited to read speech
2. Focus on agriculture and finance domains
3. Potential noise and overlap considerations
4. Evaluation beyond RESPIN test sets
5. Further elaboration of ethical safeguards

### Responses and Planned Revisions

1. **Plan for Including Spontaneous Speech**
   As noted in the paper, **RESPIN-S2.0** will extend to spontaneous speech across the same dialects and add domains such as healthcare and education.

2. **Rationale for Domain Selection and Future Expansion**
   Agriculture and finance were chosen for their relevance to underrepresented groups, providing domain-specific vocabulary not found in general corpora. More domains will be added in future releases.

3. **Acoustic Diversity and Dataset Uniqueness**
   Mobile-based crowdsourcing resulted in varied SNR conditions. We provide noise analysis and categorisation into clean, semi-noisy, and noisy subsets. The dataset has minimal overlap with existing corpora while covering dialects absent from other public resources.

4. **Evaluation on External Public Test Sets**
   We have added results on multiple public Hindi test sets, showing RESPIN-trained models perform well despite domain differences. We also evaluated on available public test sets for other languages; these results will appear in the camera-ready version.

5. **Ethics, Participant Safeguards, and Transparency**
   The ethics section will expand to include IRB approval, informed consent, anonymisation, fair compensation, demographic diversity, participant onboarding, and consent withdrawal. Additional details from outreach materials will be added.

### Additional Planned Updates
- Adding Zipformer results
- Highlighting societal and funding motivations (e.g., Gates Foundation support)
- Placing dataset and code links more prominently

These updates address reviewer concerns, improve clarity, and ensure **RESPIN-S1.0** remains a practical and accessible resource for research and development in Indian language speech technologies.

---

### Decision · Program_Chairs · 2025-09-18

**Decision:**

Accept (poster)

**Comment:**

Summary

This paper introduces RESPIN-S1.0, a large-scale, dialect-rich read speech corpus for nine Indian languages: Bengali, Bhojpuri, Chhattisgarhi, Hindi, Kannada, Magahi, Maithili, Marathi, and Telugu. The dataset comprises over 10,000 hours of validated audio, spanning more than 38 dialects. This dataset focuses on two high-impact domains, agriculture and finance, which are critical for India's rural and often digitally underserved populations. This paper details a comprehensive pipeline for dataset creation, from text composition by native dialect speakers via collection using a phone app, to multi-stage validation of both text and audio. The paper also provides extensive benchmarking with various ASR models (incl training from scratch), demonstrating that fine-tuning on RESPIN-S1.0 improves performance of existing pre-trained models on in-domain and other data. The corpus, along with speaker metadata, phonetic lexicons, and standardized data splits, is made publicly available (and sub-sets have already been used for various purposes).

Strengths

- The paper is well written and the data preparation including text preparation and audio recording gathering is explained in good detail, the pipeline seems fully documented and scales well, further increasing the utility of the paper.
- The most significant strength is the creation of over 10,000 hours of corpus with dialectal diversity, covering 38+ dialects across nine major languages incl Bhojpuri, Chhattisgarhi, and Magahi. The dataset includes proposed splits.
- The extensive benchmarking experiments using traditional, end-to-end, and large-scale pretrained models (Whisper, SSL models) effectively demonstrate the dataset's utility.
- The work has already demonstrated its impact by enabling community efforts and challenges like MADASR@ASRU 2023/2025, underlining its value to the research community.

Weaknesses

- One possible limitation of the dataset would be that it is a "read speech" dataset, with a focus on "clean" environments.
- The corpus is limited to two domains: agriculture and finance for societal relevance.
- Yet more analyses and cross-domain results could be provided.

Rationale

This is a substantial effort which makes both a practical impact (releasing a dataset), and an intellectual one (creating and implementing a data collection approach). The paper is well written and receives strong ratings from all reviewers.

Discussion

Reviewers and authors clarified various aspects of the paper, authors added more details & results, incl announcing the creation of RESPIN-S2.0, which will include spontaneous speech.